# WHO IS THE STRONGEST ENEMY? TOWARDS OPTIMAL AND EFFICIENT EVASION ATTACKS IN DEEP RL

**Yanchao Sun**[1]  **Ruijie Zheng**[2]  **Yongyuan Liang**[3]  **Furong Huang**[4]

[1,2,4] University of Maryland, College Park  [3] Sun Yat-sen University

[1,2,4]{ycs,rzheng12,furongh}@umd.edu  [3]liangyy58@mail2.sysu.edu.cn

## ABSTRACT

Evaluating the worst-case performance of a reinforcement learning (RL) agent under the strongest/optimal adversarial perturbations on state observations (within some constraints) is crucial for understanding the robustness of RL agents. However, finding the optimal adversary is challenging, in terms of both whether we can find the optimal attack and how efficiently we can find it. Existing works on adversarial RL either use heuristics-based methods that may not find the strongest adversary, or directly train an RL-based adversary by treating the agent as a part of the environment, which can find the optimal adversary but may become intractable in a large state space. This paper introduces a novel attacking method to find the optimal attacks through collaboration between a designed function named "actor" and an RL-based learner named "director". The actor crafts state perturbations for a given policy perturbation direction, and the director learns to propose the best policy perturbation directions. Our proposed algorithm, PA-AD, is theoretically optimal and significantly more efficient than prior RL-based works in environments with large state spaces. Empirical results show that our proposed PA-AD universally outperforms state-of-the-art attacking methods in various Atari and MuJoCo environments. By applying PA-AD to adversarial training, we achieve state-of-the-art empirical robustness in multiple tasks under strong adversaries.

## 1 INTRODUCTION

Deep Reinforcement Learning (DRL) has achieved incredible success in many applications. However, recent works (Huang et al., 2017; Pattanaik et al., 2018) reveal that a well-trained RL agent may be vulnerable to test-time *evasion attacks*, making it risky to deploy RL models in high-stakes applications. As in most related works, we consider a *state adversary* which adds imperceptible noise to the observations of an agent such that its cumulative reward is reduced during test time.

In order to understand the vulnerability of an RL agent and to improve its certified robustness, it is important to evaluate the worst-case performance of the agent under any adversarial attacks with certain constraints. In other words, it is crucial to find the strongest/optimal adversary that can minimize the cumulative reward gained by the agent with fixed constraints, as motivated in a recent paper by Zhang et al. (2021). Therefore, we focus on the following question:

**Given an arbitrary attack radius (budget) $\epsilon$ for each step of the deployment, what is the worst-case performance of an agent under the strongest adversary?**

Finding the strongest adversary in RL is challenging. Many existing attacks (Huang et al., 2017; Pattanaik et al., 2018) are based on heuristics, crafting adversarial states at every step independently, although steps are interrelated in contrast to image classification tasks. These heuristic methods can often effectively reduce the agent's reward, but are not guaranteed to achieve the strongest attack under a given budget. This type of attack is "myopic" since it does not plan for the future. Figure 1 shows an intuitive example, where myopic adversaries only prevent the agent from selecting the best action in the current step, but the strongest adversary can strategically "lead" the agent to a trap, which is the worst event for the agent.

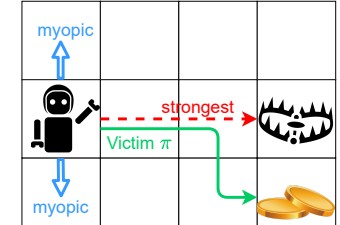

**Figure 1:** An example that a myopic adversary is not the strongest.

Achieving computational efficiency arises as another challenge in practice, even if the strongest adversary can be found in theory. A recent work (Zhang et al., 2020a) points out that learning the optimal state adversary is equivalent to learning an optimal policy in a new Markov Decision Process (MDP). A follow-up work (Zhang et al., 2021) shows that the learned adversary significantly outperforms prior adversaries in MuJoCo games. However, the state space and the action space of the new MDP are both as large as the state space in the original environment, which can be high-dimensional in practice. For example, video games and autonomous driving systems use images as observations. In these tasks, learning the state adversary directly becomes computationally intractable.

To overcome the above two challenges, **we propose a novel attack method called Policy Adversarial Actor Director (PA-AD)**, where we design a "director" and an "actor" that collaboratively finds the optimal state perturbations. In PA-AD, a director learns an MDP named *Policy Adversary MDP (PAMDP)*, and an actor is embedded in the dynamics of PAMDP. At each step, the director proposes a perturbing direction in the policy space, and the actor crafts a perturbation in the state space to lead the victim policy towards the proposed direction. Through a trail-and-error process, the director can find the optimal way to cooperate with the actor and attack the victim policy. Theoretical analysis shows that the optimal policy in PAMDP induces an optimal state adversary. The size of PAMDP is generally smaller than the adversarial MDP defined by Zhang et al. (2021) and thus is easier to be learned efficiently using off-the-shelf RL algorithms. With our proposed *director-actor collaborative mechanism*, PA-AD outperforms state-of-the-art attacking methods on various types of environments, and improves the robustness of many DRL agents by adversarial training.

**Summary of Contributions**
**(1)** We establish a theoretical understanding of the optimality of evasion attacks from the perspective of policy perturbations, allowing a more efficient implementation of optimal attacks.
**(2)** We introduce a Policy Adversary MDP (PAMDP) model, whose optimal policy induces the optimal state adversary under any attacking budget $\epsilon$.
**(3)** We propose a novel attack method, PA-AD, which efficiently searches for the optimal adversary in the PAMDP. PA-AD is a general method that works on stochastic and deterministic victim policies, vectorized and pixel state spaces, as well as discrete and continuous action spaces.
**(4)** Empirical study shows that PA-AD universally outperforms previous attacking methods in various environments, including Atari games and MuJoCo tasks. PA-AD achieves impressive attacking performance in many environments using very small attack budgets,
**(5)** Combining our strong attack PA-AD with adversarial training, we significantly improve the robustness of RL agents, and achieve the *state-of-the-art robustness in many tasks.*

## 2 PRELIMINARIES AND NOTATIONS

**The Victim RL Agent** In RL, an agent interacts with an environment modeled by a Markov Decision Process (MDP) denoted as a tuple $\mathcal{M} = \langle \mathcal{S}, \mathcal{A}, P, R, \gamma \rangle$, where $\mathcal{S}$ is a state space with cardinality $|\mathcal{S}|$, $\mathcal{A}$ is an action space with cardinality $|\mathcal{A}|$, $P : \mathcal{S} \times \mathcal{A} \to \Delta(\mathcal{S})$ is the transition function [1], $R : \mathcal{S} \times \mathcal{A} \to \mathbb{R}$ is the reward function, and $\gamma \in (0, 1)$ is the discount factor. In this paper, we consider a setting where the state space is much larger than the action space, which arises in a wide variety of environments. For notation simplicity, our theoretical analysis focuses on a finite MDP, but our algorithm applies to continuous state spaces and continuous action spaces, as verified in experiments. The agent takes actions according to its *policy*, $\pi : \mathcal{S} \to \Delta(\mathcal{A})$. We suppose the victim uses a fixed policy $\pi$ with a function approximator (e.g. a neural network) during test time. We denote the *space of all policies* as $\Pi$, which is a Cartesian product of $|\mathcal{S}|$ simplices. The *value* of a policy $\pi \in \Pi$ for state $s \in \mathcal{S}$ is defined as $V^{\pi}(s) = \mathbb{E}_{\pi,P}[\sum_{t=0}^{\infty} \gamma^t R(s_t, a_t)|s_0 = s]$.

**Evasion Attacker** Evasion attacks are test-time attacks that aim to reduce the expected total reward gained by the agent/victim. As in most literature (Huang et al., 2017; Pattanaik et al., 2018; Zhang et al., 2020a), we assume the attacker knows the victim policy $\pi$ (white-box attack). However, the attacker does not know the environment dynamics, nor does it have the ability to change the environment directly. The attacker can observe the interactions between the victim agent and the environment, including states, actions and rewards. We focus on a typical *state adversary* (Huang et al., 2017; Zhang et al., 2020a), which perturbs the state observations returned by the environment before the agent observes them. Note that the underlying states in the environment are not changed.

---

[1] $\Delta(X)$ denotes the the space of probability distributions over $X$.

Formally, we model a state adversary by a function $h$ which perturbs state $s \in \mathcal{S}$ into $\tilde{s} := h(s)$, so that the input to the agent's policy is $\tilde{s}$ instead of $s$. In practice, the adversarial perturbation is usually under certain constraints. In this paper, we consider the common $\ell_p$ threat model (Goodfellow et al., 2015): $\tilde{s}$ should be in $\mathcal{B}_\epsilon(s)$, where $\mathcal{B}_\epsilon(s)$ denotes an $\ell_p$ norm ball centered at $s$ with radius $\epsilon \geq 0$, a constant called the *budget* of the adversary for every step. With the budget constraint, we define the *admissible state adversary* and the *admissible adversary set* as below.

**Definition 1** (**Set of Admissible State Adversaries** $H_\epsilon$). *A state adversary $h$ is said to be admissible if $\forall s \in \mathcal{S}$, we have $h(s) \in \mathcal{B}_\epsilon(s)$. The set of all admissible state adversaries is denoted by $H_\epsilon$.*

Then the goal of the attacker is to find an adversary $h^*$ in $H_\epsilon$ that maximally reduces the cumulative reward of the agent. In this work, we propose a novel method to learn the optimal state adversary through the identification of an optimal *policy perturbation* defined and motivated in the next section.

## 3 UNDERSTANDING OPTIMAL ADVERSARY VIA POLICY PERTURBATIONS

In this section, we first motivate our idea of interpreting evasion attacks as perturbations of policies, then discuss how to efficiently find the optimal state adversary via the optimal policy perturbation.

**Evasion Attacks Are Perturbations of Policies**  Although existing literature usually considers state-attacks and action-attacks separately, we point out that evasion attacks, either applied to states or actions, are essentially equivalent to perturbing the agent's policy $\pi$ into another policy $\pi_h$ in the policy space $\Pi$. For instance, as shown in Figure 2, if the adversary $h$ alters state $s$ into state $\tilde{s}$, the victim selects an action $\tilde{a}$ based on $\pi(\cdot|\tilde{s})$. This is equivalent to directly perturbing $\pi(\cdot|s)$ to $\pi_h(\cdot|s) := \pi(\cdot|\tilde{s})$. (See Appendix A for more detailed analysis including action adversaries.)

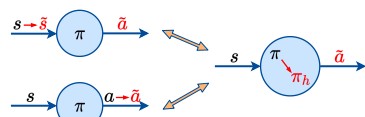

**Figure 2:** Equivalence between evasion attacks and policy perturbations.

In this paper, we aim to find the optimal state adversary through the identification of the "optimal policy perturbation", which has the following **merits**. **(1)** $\pi_h(\cdot|s)$ usually lies in a lower dimensional space than $h(s)$ for an arbitrary state $s \in \mathcal{S}$. For example, in Atari games, the action space is discrete and small (e.g. $|\mathcal{A}| = 18$), while a state is a high-dimensional image. Then the state perturbation $h(s)$ is an image, while $\pi_h(\cdot|s)$ is a vector of size $|\mathcal{A}|$. **(2)** It is easier to characterize the optimality of a policy perturbation than a state perturbation. How a state perturbation changes the value of a victim policy depends on both the victim policy network and the environment dynamics. In contrast, how a policy perturbation changes the victim value only depends on the environment. Our Theorem 4 in Section 3 and Theorem 12 in Appendix B both provide insights about how $V^\pi$ changes as $\pi$ changes continuously. **(3)** Policy perturbation captures the essence of evasion attacks, and unifies state and action attacks. Although this paper focuses on state-space adversaries, the learned "optimal policy perturbation" can also be used to conduct action-space attacks against the same victim.

**Characterizing the Optimal Policy Adversary**  As depicted in Figure 3, the policy perturbation serves as a bridge connecting the perturbations in the state space and the value space. Our goal is to find the optimal state adversary by identifying the optimal "policy adversary". We first define an Admissible Adversarial Policy Set (Adv-policy-set) $\mathcal{B}_\epsilon^H(\pi) \subset \Pi$ as the set of policies perturbed from $\pi$ by all admissible state adversaries $h \in H_\epsilon$. In other words, when a state adversary perturbs states within an $\ell_p$ norm ball $\mathcal{B}_\epsilon(\cdot)$, the victim policy is perturbed within $\mathcal{B}_\epsilon^H(\pi)$.

**Definition 2** (**Admissible Adversarial Policy Set (Adv-policy-set)** $\mathcal{B}_\epsilon^H(\pi)$). *For an MDP $\mathcal{M}$, a fixed victim policy $\pi$, we define the admissible adversarial policy set (Adv-policy-set) w.r.t. $\pi$, denoted by $\mathcal{B}_\epsilon^H(\pi)$, as the set of policies that are perturbed from $\pi$ by all admissible adversaries, i.e.,*

$$\mathcal{B}_\epsilon^H(\pi) := \{\pi_h \in \Pi : \exists h \in H_\epsilon \; s.t \; \forall s, \pi_h(\cdot|s) = \pi(\cdot|h(s))\}. \tag{1}$$

**Remarks**  (1) $\mathcal{B}_\epsilon^H(\pi)$ is a subset of the policy space $\Pi$ and it surrounds the victim $\pi$, as shown in Figure 3(middle). In the same MDP, $\mathcal{B}_\epsilon^H(\pi)$ varies for different victim $\pi$ or different attack budget $\epsilon$. (2) In Appendix B, we characterize the topological properties of $\mathcal{B}_\epsilon^H(\pi)$. We show that for a continuous function $\pi$ (e.g., neural network), $\mathcal{B}_\epsilon^H(\pi)$ is connected and compact, and the value functions generated by all policies in the Adv-policy-set $\mathcal{B}_\epsilon^H(\pi)$ form a polytope (Figure 3(right)), following the polytope theorem by Dadashi et al. (2019).

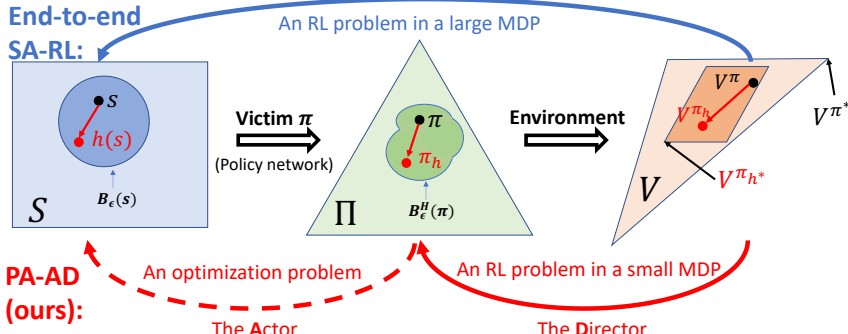

**Figure 3:** A state adversary $h$ perturbs $s$ into $h(s) \in \mathcal{B}_\epsilon(s)$ in the state space; hence, the victim's policy $\pi$ is perturbed into $\pi_h$ within the Adv-policy-set $\mathcal{B}_\epsilon^H(\pi)$; as a result, the expected total reward the victim can gain becomes $V^{\pi_h}$ instead of $V^\pi$. A prior work SA-RL (Zhang et al., 2021) directly uses an RL agent to learn the best state adversary $h^*$, which works for MDPs with small state spaces, but suffers from high complexity in larger MDPs. In contrast, we find the optimal state adversary $h^*$ efficiently through identifying the optimal policy adversary $\pi_{h^*}$. Our proposed attack method called PA-AD contains an RL-based "director" which learns to propose policy perturbation $\pi_h$ in the policy space, and a non-RL "actor", which targets at the proposed $\pi_h$ and computes adversarial states in the state space. Through this collaboration, the director can learn the optimal policy adversary $\pi_{h^*}$ using RL methods, such that the actor executes $h^*$ as justified in Theorem 7.

Given that the Adv-policy-set $\mathcal{B}_\epsilon^H(\pi)$ contains all the possible policies the victim may execute under admissible state perturbations, we can characterize the optimality of a state adversary through the lens of policy perturbations. Recall that the attacker's goal is to find a state adversary $h^* \in H_\epsilon$ that minimizes the victim's expected total reward. From the perspective of policy perturbation, the attacker's goal is to perturb the victim's policy to another policy $\pi_{h^*} \in \mathcal{B}_\epsilon^H(\pi)$ with the lowest value. Therefore, we can define the optimal state adversary and the optimal policy adversary as below.

**Definition 3** (**Optimal State Adversary** $h^*$ **and Optimal Policy Adversary** $\pi_{h^*}$). *For an MDP $\mathcal{M}$, a fixed policy $\pi$, and an admissible adversary set $H_\epsilon$ with attacking budget $\epsilon$,*
*(1) an **optimal state adversary** $h^*$ satisfies $h^* \in \operatorname{argmin}_{h \in H_\epsilon} V^{\pi_h}(s), \forall s \in \mathcal{S}$, which leads to*
*(2) an **optimal policy adversary** $\pi_{h^*}$ satisfies $\pi_{h^*} \in \operatorname{argmin}_{\pi_h \in \mathcal{B}_\epsilon^H(\pi)} V^{\pi_h}(s), \forall s \in \mathcal{S}$.*
*Recall that $\pi_h$ is the perturbed policy caused by adversary $h$, i.e., $\pi_h(\cdot|s) = \pi(\cdot|h(s)), \forall s \in \mathcal{S}$.*

Definition 3 implies an equivalent relationship between the optimal state adversary and the optimal policy adversary: an optimal state adversary leads to an optimal policy adversary, and any state adversary that leads to an optimal policy adversary is optimal. Theorem 19 in Appendix D.1 shows that there always exists an optimal policy adversary for a fixed victim $\pi$, and learning the optimal policy adversary is an RL problem. (A similar result have been shown by Zhang et al. (2020a) for the optimal state adversary, while we focus on the policy perturbation.)

Due to the equivalence, if one finds an optimal policy adversary $\pi_{h^*}$, then the optimal state adversary can be found by executing targeted attacks with target policy $\pi_{h^*}$. However, directly finding the optimal policy adversary in the Adv-policy-set $\mathcal{B}_\epsilon^H(\pi)$ is challenging since $\mathcal{B}_\epsilon^H(\pi)$ is generated by all admissible state adversaries in $H_\epsilon$ and is hard to compute. To address this challenge, we first get insights from theoretical characterizations of the Adv-policy-set $\mathcal{B}_\epsilon^H(\pi)$. Theorem 4 below shows that the "outermost boundary" of $\mathcal{B}_\epsilon^H(\pi)$ always contains an optimal policy adversary. Intuitively, a policy $\pi'$ is in the outermost boundary of $\mathcal{B}_\epsilon^H(\pi)$ if and only if no policy in $\mathcal{B}_\epsilon^H(\pi)$ is farer away from $\pi$ than $\pi'$ in the direction $\pi' - \pi$. Therefore, if an adversary can perturb a policy along a direction, it should push the policy as far away as possible in this direction under the budget constraints. Then, the adversary is guaranteed to find an optimal policy adversary after trying all the perturbing directions. In contrast, such a guarantee does not exist for state adversaries, justifying the benefits of considering policy adversaries. Our proposed algorithm in Section 4 applies this idea to find the optimal attack: *an RL-based director searches for the optimal perturbing direction, and an actor is responsible for pushing the policy to the outermost boundary of $\mathcal{B}_\epsilon^H(\pi)$ with a given direction.*

**Theorem 4.** *For an MDP $\mathcal{M}$, a fixed policy $\pi$, and an admissible adversary set $H_\epsilon$, define the **outermost boundary** of the admissible adversarial policy set $\mathcal{B}_\epsilon^H(\pi)$ w.r.t $\pi$ as*

$$\partial_\pi \mathcal{B}_\epsilon^H(\pi) := \{\pi' \in \mathcal{B}_\epsilon^H(\pi) : \forall s \in \mathcal{S}, \theta > 0, \nexists \hat{\pi} \in \mathcal{B}_\epsilon^H(\pi) \text{ s.t. } \hat{\pi}(\cdot|s) = \pi'(\cdot|s) + \theta(\pi'(\cdot|s) - \pi(\cdot|s))\}. \quad (2)$$

*Then there exists a policy $\tilde{\pi} \in \partial_\pi \mathcal{B}_\epsilon^H(\pi)$, such that $\tilde{\pi}$ is the optimal policy adversary w.r.t. $\pi$.*

Theorem 4 is proven in Appendix B.3, and we visualize the outermost boundary in Appendix B.5.

## 4    PA-AD: OPTIMAL AND EFFICIENT EVASION ATTACK

In this section, we first formally define the optimality of an attack algorithm and discuss some existing attack methods. Then, based on the theoretical insights in Section 3, we introduce our algorithm, *Policy Adversarial Actor Director (PA-AD)* that has an optimal formulation and is efficient to use.

Although many attack methods for RL agents have been proposed (Huang et al., 2017; Pattanaik et al., 2018; Zhang et al., 2020a), it is not yet well-understood how to characterize the strength and the optimality of an attack method. Therefore, we propose to formulate the optimality of an attack algorithm, which answers the question "whether the attack objective finds the strongest adversary".

**Definition 5** (Optimal Formulation of Attacking Algorithm)**.** *An attacking algorithm* Algo *is said to have an optimal formulation iff for any MDP* $\mathcal{M}$*, policy* $\pi$ *and admissible adversary set* $H_\epsilon$ *under attacking budget* $\epsilon$*, the set of optimal solutions to its objective,* $H_\epsilon^{\mathsf{Algo}}$*, is a subset of the optimal adversaries against* $\pi$*, i.e.,* $H_\epsilon^{\mathsf{Algo}} \subseteq H_\epsilon^* := \{h^* | h^* \in \arg\min_{h \in H_\epsilon} V^{\pi_h}(s), \forall s \in \mathcal{S}\}$*.*

Many heuristic-based attacks, although are empirically effective and efficient, do not meet the requirements of optimal formulation. In Appendix D.3, we categorize existing heuristic attack methods into four types, and theoretically prove that there exist scenarios where these heuristic methods may not find the strongest adversary. A recent paper (Zhang et al., 2021) proposes to learn the optimal state adversary using RL methods, which we will refer to as *SA-RL* in our paper for simplicity. SA-RL can be viewed as an "end-to-end" RL attacker, as it directly learns the optimal state adversary such that the value of the victim policy is minimized. The formulation of SA-RL satisfies Definition 5 and thus is optimal. However, SA-RL learns an MDP whose state space and action space are both the same as the original state space. If the original state space is high-dimensional (e.g. images), learning a good policy in the adversary's MDP may become computationally intractable, as empirically shown in Section 6.

Can we address the optimal attacking problem in an efficient manner? SA-RL treats the victim and the environment together as a black box and directly learns a state adversary. But if the victim policy is known to the attacker (e.g. in adversarial training), we can exploit the victim model and simplify the attacking problem while maintaining the optimality. Therefore, we propose a novel algorithm, *Policy Adversarial Actor Director (PA-AD)*, that has optimal formulation and is generally more efficient than SA-RL. PA-AD decouples the whole attacking process into two simpler components: policy perturbation and state perturbation, solved by a "director" and an "actor" through collaboration. The director learns the optimal policy perturbing direction with RL methods, while the actor crafts adversarial states at every step such that the victim policy is perturbed towards the given direction. Compared to the black-box SA-RL, PA-AD is a white-box attack, but works for a broader range of environments more efficiently. Note that PA-AD can be used to conduct black-box attack based on the transferability of adversarial attacks (Huang et al., 2017), although it is out of the scope of this paper. Appendix F.2 provides a comprehensive comparison between PA-AD and SA-RL in terms of complexity, optimality, assumptions and applicable scenarios.

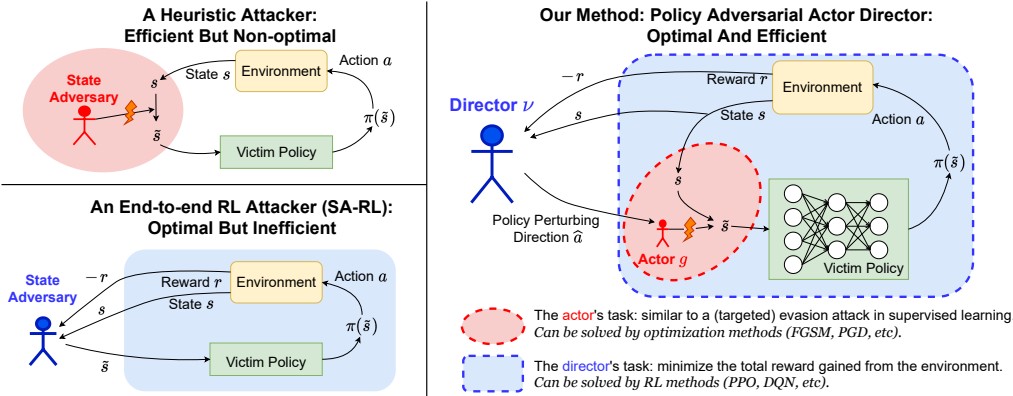

**Figure 4:** An overview of PA-AD compared with a heuristic attacker and an end-to-end RL attacker. Heuristic attacks are efficient, but may not find the optimal adversary as they do not learn from the environment dynamics. An end-to-end RL attacker directly learns a policy to generate state perturbations, but is inefficient in large-state-space environments. In contrast, our **PA-AD** solves the attack problem with a combination of an RL-based director and a non-RL actor, so that PA-AD achieves both optimality and efficiency.

Formally, for a given victim policy $\pi$, our proposed PA-AD algorithm solves a *Policy Adversary MDP (PAMDP)* defined in Definition 6. An actor denoted by $g$ is embedded in the dynamics of the PAMDP, and a director searches for an optimal policy $\nu^*$ in the PAMDP.

**Definition 6** (**Policy Adversary MDP (PAMDP)** $\widehat{\mathcal{M}}$). *Given an MDP $\mathcal{M} = \langle \mathcal{S}, \mathcal{A}, P, R, \gamma \rangle$, a fixed **stochastic** victim policy $\pi$, an attack budget $\epsilon \geq 0$, we define a Policy Adversarial MDP $\widehat{\mathcal{M}} = \langle \mathcal{S}, \widehat{\mathcal{A}}, \widehat{P}, \widehat{R}, \gamma \rangle$, where the action space is $\widehat{\mathcal{A}} := \{d \in [-1,1]^{|\mathcal{A}|}, \sum_{i=1}^{|\mathcal{A}|} d_i = 0\}$, and $\forall s, s' \in \mathcal{S}, \forall \widehat{a} \in \widehat{\mathcal{A}}$,*

$$\widehat{P}(s'|s,\widehat{a}) = \sum_{a \in \mathcal{A}} \pi(a|g(\widehat{a},s))P(s'|s,a), \quad \widehat{R}(s,\widehat{a}) = -\sum_{a \in \mathcal{A}} \pi(a|g(\widehat{a},s))R(s,a),$$

*where $g$ is the actor function defined as*

$$g(\widehat{a},s) = \mathrm{argmax}_{\tilde{s} \in B_\epsilon(s)} \|\pi(\tilde{s}) - \pi(s)\| \text{ subject to } \left(\pi(\tilde{s}) - \pi(s)\right)^T \widehat{a} = \|\pi(\tilde{s}) - \pi(s)\| \|\widehat{a}\|. \quad (G)$$

*If the victim policy is **deterministic**, i.e., $\pi_D := \mathrm{argmax}_a \pi(a|s)$, (subscript $_D$ stands for deterministic), the action space of PAMDP is $\widehat{\mathcal{A}}_D := \mathcal{A}$, and the actor function $g_D$ is*

$$g_D(\widehat{a},s) = \mathrm{argmax}_{\tilde{s} \in \mathcal{B}_\epsilon(s)} \left(\pi(\widehat{a}|\tilde{s}) - \max_{a \in \mathcal{A}, a \neq \widehat{a}} \pi(a|\tilde{s})\right). \quad (G_D)$$

*Detailed definition of the deterministic-victim version of PAMDP is in Appendix C.1.*

A key to PA-AD is the director-actor collaboration mechanism. The input to director policy $\nu$ is the current state $s$ in the original environment, while its output $\widehat{a}$ is a signal to the actor denoting "which direction to perturb the victim policy into". $\widehat{\mathcal{A}}$ is designed to contain all "perturbing directions" in the policy space. That is, $\forall \widehat{a} \in \widehat{\mathcal{A}}$, there exists a constant $\theta_0 \geq 0$ such that $\forall \theta \leq \theta_0, \pi(\cdot|s) + \theta \frac{\widehat{a}}{\|\widehat{a}\|}$ belongs to the simplex $\Delta(A)$. The actor $g$ takes in the state $s$ and director's direction $\widehat{a}$ and then computes a state perturbation within the attack budget. Therefore, the director and the actor together induce a state adversary: $h(s) := g(\nu(s), s), \forall s \in \mathcal{S}$. The definition of PAMDP is slightly different for a stochastic victim policy and a deterministic victim policy, as described below.

*For a stochastic victim $\pi$,* the director's action $\widehat{a} \in \widehat{\mathcal{A}}$ is designed to be a unit vector lying in the policy simplex, denoting the perturbing direction in the policy space. The actor, once receiving the perturbing direction $\widehat{a}$, will "push" the policy as far as possible by perturbing $s$ to $g(\widehat{a},s) \in \mathcal{B}_\epsilon(s)$, as characterized by the optimization problem ($G$). In this way, the policy perturbation resulted by the director and the actor is always in the outermost boundary of $\mathcal{B}_\epsilon^H(\pi)$ w.r.t. the victim $\pi$, where the optimal policy perturbation can be found according to Theorem 4.

*For a deterministic victim $\pi_D$,* the director's action $\widehat{a} \in \widehat{\mathcal{A}}_D$ can be viewed as a target action in the original action space, and the actor conducts targeted attacks to let the victim execute $\widehat{a}$, by forcing the logit corresponding to the target action to be larger than the logits of other actions.

In both the stochastic-victim and deterministic-victim case, PA-AD has an optimal formulation as stated in Theorem 7 (proven in Appendix D.2).

**Theorem 7** (**Optimality of PA-AD**). *For any MDP $\mathcal{M}$, any fixed victim policy $\pi$, and any attack budget $\epsilon \geq 0$, an optimal policy $\nu^*$ in $\widehat{\mathcal{M}}$ induces an optimal state adversary against $\pi$ in $\mathcal{M}$. That is, the formulation of PA-AD is optimal, i.e., $H^{PA\text{-}AD} \subseteq H_\epsilon^*$.*

---

**Algorithm 1:** Policy Adversarial Actor Director (PA-AD)

---

1 **Input:** Initialization of director's policy $\nu$; victim policy $\pi$; budget $\epsilon$; start state $s_0$
2 **for** $t = 0, 1, 2, ...$ **do**
3      *Director* samples a policy perturbing direction $\widehat{a}_t \sim \nu(\cdot|s_t)$
4      *Actor* perturbs $s_t$ to $\tilde{s}_t = g_D(\widehat{a}_t, s_t)$ if *Victim* is deterministic, otherwise to $\tilde{s}_t = g(\widehat{a}_t, s_t)$
5      *Victim* takes action $a_t \sim \pi(\cdot|\tilde{s}_t)$, proceeds to $s_{t+1}$, receives $r_t$
6      *Director* saves $(s_t, \widehat{a}_t, -r_t, s_{t+1})$ to its buffer
7      *Director* updates its policy $\nu$ using any RL algorithm

---

**Efficiency of PA-AD** As commonly known, the sample complexity and computational cost of learning an MDP usually grow with the cardinalities of its state space and action space. Both SA-RL and PA-AD have state space $\mathcal{S}$, the state space of the original MDP. But the action space of SA-RL is also $\mathcal{S}$, while our PA-AD has action space $\mathbb{R}^{|\mathcal{A}|}$ for stochastic victim policies, or $\mathcal{A}$ for deterministic victim policies. In most DRL applications, the state space (e.g., images) is much larger than the action space, then PA-AD is generally more efficient than SA-RL as it learns a smaller MDP.

The attacking procedure is illustrated in Algorithm 1. At step $t$, the director observes a state $s_t$, and proposes a policy perturbation $\widehat{a}_t$, then the actor searches for a state perturbation to meet the policy perturbation. Afterwards, the victim acts with the perturbed state $\tilde{s}_t$, then the director updates its policy based on the opposite value of the victim's reward. Note that the actor solves a constrained optimization problem, $(G_D)$ or $(G)$. Problem $(G_D)$ is similar to a targeted attack in supervised learning, while the stochastic version $(G)$ can be approximately solved with a Lagrangian relaxation. In Appendix C.2, we provide our implementation details for solving the actor's optimization, which empirically achieves state-of-the-art attack performance as verified in Section 6.

**Extending to Continuous Action Space**    Our PA-AD can be extended to environments with continuous action spaces, where the actor minimizes the distance between the policy action and the target action, i.e., $\mathrm{argmin}_{s' \in B_\epsilon(s)} \| \pi(s') - \widehat{a} \|$. More details and formal definitions of the variant of PA-AD in continuous action space are provided in Appendix C.3. In Section 6, we show experimental results in MuJoCo tasks, which have continuous action spaces.

## 5   RELATED WORK

**Heuristic-based Evasion Attacks on States**    There are many works considering evasion attacks on the state observations in RL. Huang et al. (2017) first propose to use FGSM (Goodfellow et al., 2015) to craft adversarial states such that the probability that the agent selects the "best" action is minimized. The same objective is also used in a recent work by Korkmaz (2020), which adopts a Nesterov momentum-based optimization method to further improve the attack performance. Pattanaik et al. (2018) propose to lead the agent to select the "worst" action based on the victim's Q function and use gradient descent to craft state perturbations. Zhang et al. (2020a) define the concept of a state-adversarial MDP (SAMDP) and propose two attack methods: Robust SARSA and Maximal Action Difference. The above heuristic-based methods are shown to be effective in many environments, although might not find the optimal adversaries, as proven in Appendix D.3.

**RL-based Evasion Attacks on States**    As discussed in Section 4, SA-RL (Zhang et al., 2021) uses an end-to-end RL formulation to learn the optimal state adversary, which achieves state-of-the-art attacking performance in MuJoCo tasks. For a pixel state space, an end-to-end RL attacker may not work as shown by our experiment in Atari games (Section 6). Russo & Proutiere (2021) propose to use feature extraction to convert the pixel state space to a small state space and then learn an end-to-end RL attacker. But such feature extractions require expert knowledge and can be hard to obtain in many real-world applications. In contrast, our PA-AD works for both pixel and vector state spaces and does not require expert knowledge.

**Other Works Related to Adversarial RL**    There are many other papers studying adversarial RL from different perspectives, including limited-steps attacking (Lin et al., 2017; Kos & Song, 2017), multi-agent scenarios (Gleave et al., 2020), limited access to data (Inkawhich et al., 2020), and etc. Adversarial action attacks (Xiao et al., 2019; Tan et al., 2020; Tessler et al., 2019; Lee et al., 2021) are developed separately from state attacks; although we mainly consider state adversaries, our PA-AD can be extended to action attacks as formulated in Appendix A. Poisoning (Behzadan & Munir, 2017; Huang & Zhu, 2019; Sun et al., 2021; Zhang et al., 2020b; Rakhsha et al., 2020) is another type of adversarial attacks that manipulates the training data, different from evasion attacks that deprave a well-trained policy. Training a robust agent is the focus of many recent works (Pinto et al., 2017; Fischer et al., 2019; Lütjens et al., 2020; Oikarinen et al., 2020; Zhang et al., 2020a; 2021). Although our main goal is to find a strong attacker, we also show by experiments that our proposed attack method significantly improves the robustness of RL agents by adversarial training.

## 6   EXPERIMENTS

In this section, we show that PA-AD produces stronger evasion attacks than state-of-the-art attack algorithms on various OpenAI Gym environments, including Atari and MuJoCo tasks. Also, our experiment justifies that PA-AD can evaluate and improve the robustness of RL agents.

**Baselines and Performance Metric**    We compare our proposed attack algorithm with existing evasion attack methods, including *MinBest* (Huang et al., 2017) which minimizes the probability that the agent chooses the "best" action, *MinBest +Momentum* (Korkmaz, 2020) which uses Nesterov momentum to improve the performance of MinBest, *MinQ* (Pattanaik et al., 2018) which leads

| | Environment | Natural Reward | $\epsilon$ | Random | MinBest | MinBest + Momentum | MinQ | MaxDiff | SA-RL | PA-AD (ours) |
|---|---|---|---|---|---|---|---|---|---|---|
| **DQN** | **Boxing** | 96 ± 4 | 0.001 | 95 ± 4 | 53 ± 16 | 52 ± 18 | 88 ± 7 | 95 ± 5 | 94 ± 6 | **19 ± 11** |
| | **Pong** | 21 ± 0 | 0.0002 | 21 ± 0 | −10 ± 4 | −14 ± 2 | 14 ± 3 | 15 ± 4 | 20 ± 1 | **−21 ± 0** |
| | **RoadRunner** | 46278 ± 4447 | 0.0005 | 44725 ± 6614 | 17012 ± 6243 | 15823 ± 5252 | 5765 ± 12331 | 36074 ± 6544 | 43615 ± 7183 | **0 ± 0** |
| | **Freeway** | 34 ± 1 | 0.0003 | 34 ± 1 | 12 ± 1 | 12 ± 1 | 15 ± 2 | 22 ± 3 | 34 ± 1 | **9 ± 1** |
| | **Seaquest** | 10650 ± 2716 | 0.0005 | 8177 ± 2962 | 3820 ± 1947 | 2337 ± 862 | 6468 ± 2493 | 5718 ± 1884 | 8152 ± 3113 | **2304 ± 838** |
| | **Alien** | 1623 ± 252 | 0.00075 | 1650 ± 381 | 819 ± 486 | 775 ± 648 | 938 ± 446 | 869 ± 279 | 1693 ± 439 | **256 ± 210** |
| | **Tutankham** | 227 ± 29 | 0.00075 | 221 ± 65 | 30 ± 13 | 26 ± 16 | 88 ± 74 | 130 ± 48 | 202 ± 65 | **0 ± 0** |
| **A2C** | **Breakout** | 356 ± 79 | 0.0005 | 355 ± 79 | 86 ± 104 | 74 ± 95 | N/A | 304 ± 111 | 353 ± 79 | **44 ± 62** |
| | **Seaquest** | 1752 ± 70 | 0.005 | 1752 ± 73 | 356 ± 153 | 179 ± 83 | N/A | 46 ± 52 | 1752 ± 71 | **4 ± 13** |
| | **Pong** | 20 ± 1 | 0.0005 | 20 ± 1 | −4 ± 8 | −11 ± 7 | N/A | 18 ± 3 | 20 ± 1 | **−13 ± 6** |
| | **Alien** | 1615 ± 601 | 0.001 | 1629 ± 592 | 1062 ± 610 | 940 ± 565 | N/A | 1482 ± 633 | 1661 ± 625 | **507 ± 278** |
| | **Tutankham** | 258 ± 53 | 0.001 | 260 ± 54 | 139 ± 26 | 134 ± 28 | N/A | 196 ± 34 | 260 ± 54 | **71 ± 47** |
| | **RoadRunner** | 34367 ± 6355 | 0.002 | 35851 ± 6675 | 9198 ± 3814 | 5410 ± 3058 | N/A | 31856 ± 7125 | 36550 ± 6848 | **2773 ± 3468** |

**Table 1:** Average episode rewards ± standard deviation of vanilla DQN and A2C agents under different evasion attack methods in Atari environments. Results are averaged over 1000 episodes. Note that RS works for continuous action spaces, thus is not included. MinQ is not applicable to A2C which does not have a Q network. In each row, we bold the strongest (best) attack performance over all attacking methods.

the agent to select actions with the lowest action values based on the agent's Q network, *Robust SARSA (RS)* (Zhang et al., 2020a) which performs the MinQ attack with a learned stable Q network, *MaxDiff* (Zhang et al., 2020a) which maximizes the KL-divergence between the original victim policy and the perturbed policy, as well as *SA-RL* (Zhang et al., 2021) which directly learns the state adversary with RL methods. We consider state attacks with $\ell_\infty$ norm as in most literature (Zhang et al., 2020a; 2021). Appendix E.1 provides hyperparameter settings and implementation details.

**PA-AD Finds the Strongest Adversaries in Atari Games**     We first evaluate the performance of PA-AD against well-trained DQN (Mnih et al., 2015) and A2C (Mnih et al., 2016) victim agents on Atari games with pixel state spaces. The observed pixel values are normalized to the range of [0, 1]. SA-RL and PA-AD adversaries are learned using the ACKTR algorithm (Wu et al., 2017) with the same number of steps. (Appendix E.1 shows hyperparameter settings.) Table 1 presents the experiment results, where PA-AD significantly outperforms all baselines against both DQN and A2C victims. In contrast, SA-RL does not converge to a good adversary in the tested Atari games with the same number of training steps as PA-AD, implying the importance of sample efficiency. Surprisingly, using a relatively small attack budget $\epsilon$, PA-AD leads the agent to the **lowest possible reward** in many environments such as Pong, RoadRunner and Tutankham, whereas other attackers may require larger attack budget to achieve the same attack strength. Therefore, we point out that *vanilla RL agents are extremely vulnerable to carefully learned adversarial attacks.* Even if an RL agent works well under naive attacks, a carefully learned adversary can let an agent totally fail with the same attack budget, which stresses the importance of evaluating and improving the robustness of RL agents using the strongest adversaries. Our further investigation in Appendix F.3 shows that RL models can be generally more vulnerable than supervised classifiers, due to the different loss and architecture designs. In Appendix E.2.1, we show more experiments with various selections of the budget $\epsilon$, where one can see *PA-AD reduces the average reward more than all baselines over varying $\epsilon$'s in various environments.*

**PA-AD Finds the Strongest Adversaries MuJoCo Tasks**     We further evaluate PA-AD on Mu-JoCo games, where both state spaces and action spaces are continuous. We use the same setting with Zhang et al. (2021), where both the victim and the adversary are trained with PPO (Schulman et al., 2017). During test time, the victim executes a deterministic policy, and we use the deterministic

| Environment | State Dimension | Natural Reward | $\epsilon$ | Random | MaxDiff | RS | SA-RL | PA-AD (ours) |
|---|---|---|---|---|---|---|---|---|
| **Hopper** | 11 | 3167 ± 542 | 0.075 | 2101 ± 793 | 1410 ± 655 | 794 ± 238 | 636 ± 9 | **160 ± 136** |
| **Walker** | 17 | 4472 ± 635 | 0.05 | 3007 ± 1200 | 2869 ± 1271 | 1336 ± 654 | 1086 ± 516 | **804 ± 130** |
| **HalfCheetah** | 17 | 7117 ± 98 | 0.15 | 5486 ± 1378 | 1836 ± 866 | 489 ± 758 | **−660 ± 218** | −356 ± 307 |
| **Ant** | 111 | 5687 ± 758 | 0.15 | 5261 ± 1005 | 1759 ± 828 | 268 ± 227 | −872 ± 436 | **−2580 ± 872** |

**Table 2:** Average episode rewards ± standard deviation of vanilla PPO agent under different evasion attack methods in MuJoCo environments. Results are averaged over 50 episodes. Note that MinBest and MinQ do not fit this setting, since MinBest works for discrete action spaces, and MinQ requires the agent's Q network.

version of PA-AD with a continuous action space, as discussed in Section 4 and Appendix C.3. We use the same attack budget $\epsilon$ as in Zhang et al. (2021) for all MuJoCo environments. Results in Table 2 show that PA-AD reduces the reward much more than heuristic methods, and also outperforms SA-RL in most cases. In Ant, our PA-AD achieves much stronger attacks than SA-RL, since PA-AD is more efficient than SA-RL when the state space is large. Admittedly, PA-AD requires additional knowledge of the victim model, while SA-RL works in a black-box setting. Therefore, SA-RL is more applicable to black-box scenarios with a relatively small state space, whereas PA-AD is more applicable when the attacker has access to the victim (e.g. in adversarial training as shown in Table 3). Appendix E.2.3 provides more empirical comparison between SA-RL and PA-AD, which shows that *PA-AD converges faster, takes less running time, and is less sensitive to hyperparameters than SA-RL* by a proper exploitation of the victim model.

| Environment | Model | Natural Reward | Random | MaxDiff | RS | SA-RL | PA-AD (ours) | Average reward across attacks |
|---|---|---|---|---|---|---|---|---|
| **Hopper** (state-dim: 11) $\epsilon$: 0.075 | SA-PPO | $3705 \pm 2$ | $2710 \pm 801$ | $2652 \pm 835$ | $1130 \pm 42$ | $1076 \pm 791$ | $\mathbf{856 \pm 21}$ | 1684.8 |
| | ATLA-PPO | $3291 \pm 600$ | $3165 \pm 576$ | $2814 \pm 725$ | $2244 \pm 618$ | $1772 \pm 802$ | $\mathbf{1232 \pm 350}$ | 2245.4 |
| | **PA-ATLA-PPO (ours)** | $3449 \pm 237$ | $3325 \pm 239$ | $3145 \pm 546$ | $3002 \pm 129$ | $\mathbf{1529 \pm 284}$ | $2521 \pm 325$ | 2704.4 |
| **Walker** (state-dim: 17) $\epsilon$: 0.05 | SA-PPO | $4487 \pm 61$ | $4867 \pm 39$ | $3668 \pm 1789$ | $3808 \pm 138$ | $2908 \pm 1136$ | $\mathbf{1042 \pm 153}$ | 3258.6 |
| | ATLA-PPO | $3842 \pm 475$ | $3927 \pm 368$ | $3836 \pm 492$ | $3239 \pm 894$ | $3663 \pm 707$ | $\mathbf{1224 \pm 770}$ | 3177.8 |
| | **PA-ATLA-PPO (ours)** | $4178 \pm 529$ | $4129 \pm 78$ | $4024 \pm 572$ | $3966 \pm 307$ | $3450 \pm 478$ | $\mathbf{2248 \pm 131}$ | 3563.4 |
| **Halfcheetah** (state-dim: 17) $\epsilon$: 0.15 | SA-PPO | $3632 \pm 20$ | $3619 \pm 18$ | $3624 \pm 23$ | $3283 \pm 20$ | $3028 \pm 23$ | $\mathbf{2512 \pm 16}$ | 3213.2 |
| | ATLA-PPO | $6157 \pm 852$ | $6164 \pm 603$ | $5790 \pm 174$ | $4806 \pm 603$ | $5058 \pm 718$ | $\mathbf{2576 \pm 1548}$ | 4878.8 |
| | **PA-ATLA-PPO (ours)** | $6289 \pm 342$ | $6215 \pm 346$ | $5961 \pm 53$ | $5226 \pm 114$ | $4872 \pm 79$ | $\mathbf{3840 \pm 673}$ | 5222.8 |
| **Ant** (state-dim: 111) $\epsilon$: 0.15 | SA-PPO | $4292 \pm 384$ | $4986 \pm 452$ | $4662 \pm 522$ | $3412 \pm 1755$ | $2511 \pm 1117$ | $\mathbf{-1296 \pm 923}$ | 2855.0 |
| | ATLA-PPO | $5359 \pm 153$ | $5366 \pm 104$ | $5240 \pm 170$ | $4136 \pm 149$ | $3765 \pm 101$ | $\mathbf{220 \pm 338}$ | 3745.4 |
| | **PA-ATLA-PPO (ours)** | $5469 \pm 106$ | $5496 \pm 158$ | $5328 \pm 196$ | $4124 \pm 291$ | $3694 \pm 188$ | $\mathbf{2986 \pm 864}$ | 4325.6 |

**Table 3:** Average episode rewards $\pm$ standard deviation of robustly trained PPO agents under different attack methods. Results are averaged over 50 episodes. In each row corresponding to a robust agent, we bold the strongest attack. The gray cells are the most robust agents with the highest average rewards across attacks. Our PA-AD achieves the strongest attack against robust models, and our PA-ATLA-PPO achieves the most robust performance under multiple attacks. The attack budget $\epsilon$'s are the same as in Zhang et al. (2021).

**Training and Evaluating Robust Agents** A natural application of PA-AD is to evaluate the robustness of a known model, or to improve the robustness of an agent via adversarial training, where the attacker has white-box access to the victim. Inspired by ATLA (Zhang et al., 2021) which alternately trains an agent and an SA-RL attacker, we propose PA-ATLA, which alternately trains an agent and a PA-AD attacker. In Table 3, we evaluate the performance of PA-ATLA for a PPO agent (namely PA-ATLA-PPO) in MuJoCo tasks, compared with state-of-the-art robust training methods, SA-PPO (Zhang et al., 2020a) and ATLA-PPO (Zhang et al., 2021) [2]. From the table, we make the following observations. **(1)** *Our PA-AD attacker can significantly reduce the reward of previous "robust" agents.* Take the Ant environment as an example, although SA-PPO and ATLA-PPO agents gain 2k+ and 3k+ rewards respectively under SA-RL, the previously strongest attack, our PA-AD still reduces their rewards to about -1.3k and 200+ with the same attack budget. Therefore, we emphasize the importance of understanding the worst-case performance of RL agents, even robustly-trained agents. **(2)** *Our PA-ATLA-PPO robust agents gain noticeably higher average rewards across attacks than other robust agents*, especially under the strongest PA-AD attack. Under the SA-RL attack, PA-ATLA-PPO achieves comparable performance with ATLA-PPO, although ATLA-PPO agents are trained to be robust against SA-RL. Due to the efficiency of PA-AD, PA-ATLA-PPO requires fewer training steps than ATLA-PPO, as justified in Appendix E.2.4. The results of attacking and training robust models in Atari games are in Appendix E.2.5 and E.2.6, where PA-ATLA improves the robustness of Atari agents against strong attacks with $\epsilon$ as large as $3/255$.

## 7 CONCLUSION

In this paper, we propose an attack algorithm called PA-AD for RL problems, which achieves optimal attacks in theory and significantly outperforms prior attack methods in experiments. PA-AD can be used to evaluate and improve the robustness of RL agents before deployment. A potential future direction is to use our formulation for robustifying agents under both state and action attacks.

---

[2] We use **ATLA-PPO(LSTM)+SA Reg**, the most robust method reported by Zhang et al. (2021).

## ACKNOWLEDGMENTS

This work is supported by National Science Foundation IIS-1850220 CRII Award 030742-00001 and DOD-DARPA-Defense Advanced Research Projects Agency Guaranteeing AI Robustness against Deception (GARD), and Adobe, Capital One and JP Morgan faculty fellowships.

## ETHICS STATEMENT

Despite the rapid advancement of interactive AI and ML systems using RL agents, the learning agent could fail catastrophically in the presence of adversarial attacks, exposing a serious vulnerability in current RL systems such as autonomous driving systems, market-making systems, and security monitoring systems. Therefore, there is an urgent need to understand the vulnerability of an RL model, otherwise, it may be risky to deploy a trained agent in real-life applications, where the observations of a sensor usually contain unavoidable noise.

Although the study of a strong attack method may be maliciously exploited to attack some RL systems, it is more important for the owners and users of RL systems to get aware of the vulnerability of their RL agents under the strongest possible adversary. As the old saying goes, "if you know yourself and your enemy, you'll never lose a battle". In this work, we propose an optimal and efficient algorithm for evasion attacks in Deep RL (DRL), which can significantly influence the performance of a well-trained DRL agent, by adding small perturbations to the state observations of the agent. Our proposed method can automatically measure the vulnerability of an RL agent, and discover the "flaw" in a model that might be maliciously attacked. We also show in experiments that our attack method can be applied to improve the robustness of an RL agent via robust training. Since our proposed attack method achieves state-of-the-art performance, the RL agent trained under our proposed attacker could be able to "defend" against any other adversarial attacks with the same constraints. Therefore, our work has the potential to help combat the threat to high-stakes systems.

A limitation of PA-AD is that it requires the "attacker" to know the victim's policy, i.e., PA-AD is a white-box attack. If the attacker does not have full access to the victim, PA-AD can still be used based on the transferability of adversarial attacks (Huang et al., 2017), although the optimality guarantee does not hold in this case. However, this limitation only restricts the ability of the malicious attackers. In contrast, PA-AD should be used when one wants to evaluate the worst-case performance of one's own RL agent, or to improve the robustness of an agent under any attacks, since PA-AD produces strong attacks efficiently. In these cases, PA-AD does have white-box access to the agent. Therefore, ***PA-AD is more beneficial to defenders than attackers.***

## REPRODUCIBILITY STATEMENT

**For theoretical results**, we provide all detailed technical proofs and lemmas in Appendix. In Appendix A, we analyze the equivalence between evasion attacks and policy perturbations. In Appendix B, we theoretically prove some topological properties of the proposed Adv-policy-set, and derive Theorem 4 that the outermost boundary of $\mathcal{B}_\epsilon^H(\pi)$ always contains an optimal policy perturbation. In Appendix D, we systematically characterize the optimality of many existing attack methods. We theoretically show (1) the existence of an optimal adversary, (2) the optimality of our proposed PA-AD, and (3) the optimality of many heuristic attacks, following our Definition 5 in Section 4.

**For experimental results**, the detailed algorithm description in various types of environments is provided in Appendix C. In Appendix E, we illustrate the implementation details, environment settings, hyperparameter settings of our experiments. Additional experimental results show the performance of our algorithm from multiple aspects, including hyperparameter sensitivity, learning efficiency, etc. In addition, in Appendix F, we provide some detailed discussion on the algorithm design, as well as a comprehensive comparison between our method and prior works.

**The source code and running instructions** for both Atari and MuJoCo experiments are in our supplementary materials. We also provide trained victim and attacker models so that one can directly test their performance using a test script we provide.

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

# Appendix: Who Is the Strongest Enemy? Towards Optimal and Efficient Evasion Attacks in Deep RL

## A  RELATIONSHIP BETWEEN EVASION ATTACKS AND POLICY PERTURBATIONS.

As mentioned in Section 2, all evasion attacks can be regarded as perturbations in the policy space. To be more specific, we consider the following 3 cases, where we assume the victim uses policy $\pi$.

*Case 1 (attack on states)*: define the state adversary as function $h$ such that $\forall s \in \mathcal{S}$

$$h(s) = \tilde{s} \in \mathcal{B}_\epsilon(s) := \{s' \in \mathcal{S} : \|s' - s\| \le \epsilon\}.$$

(For simplicity, we consider the attacks within a $\epsilon$-radius norm ball.)
In this case, for all $s \in \mathcal{S}$, the victim samples action from $\pi_h(\cdot|s) = \pi(\cdot|h(s)) = \pi(\tilde{s})$, which is equivalent to the victim executing a perturbed policy $\pi_h \in \Pi$.

*Case 2 (attack on actions for a deterministic $\pi$)*: define the action adversary as function $h^{(\mathcal{A})} : \mathcal{S} \times \mathcal{A} \to \mathcal{A}$, and $\forall s \in \mathcal{S}, a \in \mathcal{A}$

$$h^{(\mathcal{A})}(a|s) = \tilde{a} \in \mathcal{B}_\epsilon(a) := \{a' \in \mathcal{A} : \|a' - a\| \le \epsilon\}.$$

In this case, there exists a policy $\pi_{h^{(\mathcal{A})}}$ such that $\pi_{h^{(\mathcal{A})}}(s) = h^{(\mathcal{A})}(a|s) = \tilde{a}$, which is equivalent to the victim executing policy $\pi_{h^{(\mathcal{A})}} \in \Pi$.

*Case 3 (attack on actions for a stochastic $\pi$)*: define the action adversary as function $h^{(\mathcal{A})} : \mathcal{S} \times \mathcal{A} \to \mathcal{A}$, and $\forall s \in \mathcal{S}, a \in \mathcal{A}$

$$h^{(\mathcal{A})}(a|s) = \tilde{a} \text{ such that } \{\|\pi(\cdot|s) - Pr(\cdot|s)\| \le \epsilon\},$$

where $Pr(\tilde{a}|s)$ denotes the probability that the action is perturbed into $\tilde{a}$.
In this case, there exists a policy $\pi_{h^{(\mathcal{A})}}$ such that $\pi_{h^{(\mathcal{A})}}(s) = Pr(\cdot|s)$, which is equivalent to the victim executing policy $\pi_{h^{(\mathcal{A})}} \in \Pi$.

Most existing evasion RL works (Huang et al., 2017; Pattanaik et al., 2018; Zhang et al., 2020a; 2021) focus on state attacks, while there are also some works (Tessler et al., 2019; Tan et al., 2020) studying action attacks. For example, Tessler et al. (Tessler et al., 2019) consider Case 2 and Case 3 above and train an agent that is robust to action perturbations.

These prior works study either state attacks or action attacks, considering them in two different scenarios. However, the ultimate goal of robust RL is to train an RL agent that is robust to any threat models. Otherwise, an agent that is robust against state attacks may still be ruined by an action attacker. We take a step further to this ultimate goal by proposing a framework, policy attack, that unifies observation attacks and action attacks.

Although the focus of this paper is on state attacks, we would like to point out that our proposed method can also deal with action attacks (the director proposes a policy perturbation direction, and an actor perturbs the action accordingly). It is also an exciting direction to explore hybrid attacks (multiple actors conducting states perturbations and action perturbations altogether, directed by a single director.) Our policy perturbation framework can also be easily incorporated in robust training procedures, as an agent that is robust to policy perturbations is simultaneously robust to both state attacks and action attacks.

## B  TOPOLOGICAL PROPERTIES OF THE ADMISSIBLE ADVERSARIAL POLICY SET

As discussed in Section 3, finding the optimal state adversary in the admissible adversary set $H_\epsilon$ can be converted to a problem of finding the optimal policy adversary in the Adv-policy-set $\mathcal{B}_\epsilon^H(\pi)$. In this section, we characterize the topological properties of $\mathcal{B}_\epsilon^H(\pi)$, and identify how the value function changes as the policy changes within $\mathcal{B}_\epsilon^H(\pi)$.

In Section B.1, we show that under the settings we consider, $\mathcal{B}_\epsilon^H(\pi)$ is a connected and compact subset of $\Pi$. Then, Section B.2, we define some additional concepts and re-formulate the notations.

In Section B.3, we prove Theorem 4 in Section 3 that the outermost boundary of $\mathcal{B}_\epsilon^H(\pi)$ always contains an optimal policy perturbation. In Section B.4, we prove that the value functions of policies in $\mathcal{B}_\epsilon^H(\pi)$ (or more generally, any connected and compact subset of $\Pi$) form a polytope. Section B.6 shows an example of the polytope result with a 2-state MDP, and Section B.5 shows examples of the outermost boundary defined in Theorem 4.

## B.1 THE SHAPE OF ADV-POLICY-SET $\mathcal{B}_\epsilon^H(\pi)$

It is important to note that $\mathcal{B}_\epsilon^H(\pi)$ is generally connected and compact as stated in the following lemma.

**Lemma 8** ($\mathcal{B}_\epsilon^H(\pi)$ **is connected and compact**). *Given an MDP $\mathcal{M}$, a policy $\pi$ that is a continuous mapping, and admissible adversary set $H_\epsilon := \{h : h(s) \in \mathcal{B}_\epsilon(s), \forall s \in \mathcal{S}\}$ (where $\epsilon > 0$ is a constant), the admissible adversarial policy set $\mathcal{B}_\epsilon^H(\pi)$ is a connected and compact subset of $\Pi$.*

*Proof of Lemma 8.* For an arbitrary state $s \in \mathcal{S}$, an admissible adversary $h \in H_\epsilon$ perturbs it within an $\ell_p$ norm ball $\mathcal{B}_\epsilon(s)$, which is connected and compact. Since $\pi$ is a continuous mapping, we know $\pi(s)$ is compact and connected.

Therefore, $\mathcal{B}_\epsilon^H(\pi)$ as a Cartesian product of a finite number of compact and connected sets, is compact and connected. □

## B.2 ADDITIONAL NOTATIONS AND DEFINITIONS FOR PROOFS

We first formally define some concepts and notations.

For a stationary and stochastic policy $\pi : \mathcal{S} \to \Delta(\mathcal{A})$, we can define the state-to-state transition function as

$$P^\pi(s'|s) := \sum_{a \in \mathcal{A}} \pi(a|s) P(s'|s, a), \forall s, s' \in \mathcal{S},$$

and the state reward function as

$$R^\pi(s) := \sum_{a \in \mathcal{A}} \pi(a|s) R(s, a), \forall s \in \mathcal{S}.$$

Then the value of $\pi$, denoted as $V^\pi$, can be computed via the Bellman equation

$$V^\pi = R^\pi + \gamma P^\pi V^\pi = (I - \gamma P^\pi)^{-1} R^\pi.$$

We further use $\Pi_{s_i}$ to denote the projection of $\Pi$ into the simplex of the $i$-th state, i.e., the space of action distributions at state $s_i$.

Let $f_v : \Pi \to \mathbb{R}^{|\mathcal{S}|}$ be a mapping that maps policies to their corresponding value functions. Let $\mathcal{V} = f_v(\Pi)$ be the space of all value functions.

Dadashi et al. (Dadashi et al., 2019) show that the image of $f_v$ applied to the space of policies, i.e., $f_v(\Pi)$, form a (possibly non-convex) polytope as defined below.

**Definition 9** (**(Possibly non-convex) polytope**). *A is called a **convex polytope** iff there are $k \in \mathbb{N}$ points $x_1, x_2, \cdots, x_k \in \mathbb{R}^n$ such that $A = Conv(x_1, \cdots, x_k)$. Furthermore, a **(possibly non-convex) polytope** is defined as a finite union of convex polytopes.*

And a more general concept is (possibly non-convex) polyhedron, which might not be bounded.

**Definition 10** (**(Possibly non-convex) polyhedron**). *A is called a **convex polyhedron** iff it is the intersection of $k \in \mathbb{N}$ half-spaces $\hat{B}_1, \hat{B}_2, \cdots, \hat{B}_k$, i.e., $A = \cap_{i=1}^k \hat{B}_i$. Furthermore, a **(possibly non-convex) polyhedron** is defined as a finite union of convex polyhedra.*

In addition, let $Y_{s_1, \cdots, s_k}^\pi$ be the set of policies that agree with $\pi$ on states $s_1, \cdots, s_k$. Dadashi et al. (Dadashi et al., 2019) also prove that the values of policies that agree on all but one state $s$, i.e., $f_v(Y_{\mathcal{S} \setminus \{s\}}^\pi)$, form a line segment, which can be bracketed by two policies that are deterministic on $s$. Our Lemma 14 extends this line segment result to our setting where policies are restricted in a subset of policies.

### B.3   PROOF OF THEOREM 4: BOUNDARY CONTAINS OPTIMAL POLICY PERTURBATIONS

Lemma 4 in Dadashi et al. (2019) shows that policies agreeing on all but one state have certain monotone relations. We restate this result in Lemma 11 below.

**Lemma 11** (Monotone Policy Interpolation). *For any $\pi_0, \pi_1 \in Y_{\mathcal{S} \setminus \{s\}}^{\pi}$ that agree with $\pi$ on all states except for $s \in \mathcal{S}$, define a function $l : [0, 1] \to \mathcal{V}$ as*

$$l(\alpha) = f_v(\alpha \pi_1 + (1 - \alpha)\pi_0).$$

*Then we have*
*(1) $l(0) \succcurlyeq l(1)$ or $l(1) \succcurlyeq l(0)$ ($\succcurlyeq$ stands for element-wise greater than or equal to);*
*(2) If $l(0) = l(1)$, then $l(\alpha) = l(0), \forall \alpha \in [0, 1]$;*
*(3) If $l(0) \neq l(1)$, then there is a strictly monotonic rational function $\rho : [0, 1] \to \mathbb{R}$, such that $l(\alpha) = \rho(\alpha)l(1) + (1 - \rho(\alpha))l(0)$.*

More intuitively, Lemma 11 suggests that the value of $\pi^{\alpha} := \alpha \pi_1 + (1 - \alpha)\pi_0$ changes (strictly) monotonically with $\alpha$, unless the values of $\pi_0, \pi_1$ and $\pi_{\alpha}$ are all equal. With this result, we can proceed to prove Theorem 4.

*Proof of Theorem 4.* We will prove the theorem by contradiction.

Suppose there is a policy $\hat{\pi} \in \mathcal{B}_{\epsilon}^H(\pi)$ such that $\hat{\pi} \notin \partial_{\pi} \mathcal{B}_{\epsilon}^H(\pi)$ and $f_v(\hat{\pi}) = V^{\hat{\pi}} < V^{\tilde{\pi}}, \forall \tilde{\pi} \in \mathcal{B}_{\epsilon}^H(\pi)$, i.e., there is no optimal policy adversary on the outermost boundary of $\mathcal{B}_{\epsilon}^H(\pi)$.

Then according to the definition of $\partial_{\pi} \mathcal{B}_{\epsilon}^H(\pi)$, there exists at least one state $s \in \mathcal{S}$ such that we can find another policy $\pi' \in \mathcal{B}_{\epsilon}^H(\pi)$ agreeing with $\hat{\pi}$ on all states except for $s$, where $\pi'(s)$ satisfies

$$\hat{\pi}(\cdot|s) = \alpha \pi(\cdot|s) + (1 - \alpha)\pi'(\cdot|s)$$

for some scalar $\alpha \in (0, 1)$.

Then by Lemma 11, either of the following happens:

(1) $f_v(\pi) \succ f_v(\hat{\pi}) \succ f_v(\pi')$.
(2) $f_v(\pi) = f_v(\hat{\pi}) = f_v(\pi')$;

Note that $f_v(\hat{\pi}) \succ f_v(\pi)$ is impossible because we have assumed $\hat{\pi}$ has the lowest value over all policies in $\mathcal{B}_{\epsilon}^H(\pi)$ including $\pi$.

If (1) is true, then $\pi'$ is a better policy adversary than $\hat{\pi}$ in $\mathcal{B}_{\epsilon}^H(\pi)$, which contradicts with the assumption.

If (2) is true, then $\pi'$ is another optimal policy adversary. By recursively applying the above process to $\pi'$, we can finally find an optimal policy adversary on the outermost boundary of $\mathcal{B}_{\epsilon}^H(\pi)$, which also contradicts with our assumption.

In summary, there is always an optimal policy adversary lying on the outermost boundary of $\mathcal{B}_{\epsilon}^H(\pi)$.

$\square$

### B.4   PROOF OF THEOREM 12: VALUES OF POLICIES IN ADMISSIBLE ADVERSARIAL POLICY SET FORM A POLYTOPE

We first present a theorem that describes the "shape" of the value functions generated by all admissible adversaries (admissible adversarial policies).

**Theorem 12** (**Policy Perturbation Polytope**). *For a finite MDP $\mathcal{M}$, consider a policy $\pi$ and an Adv-policy-set $\mathcal{B}_{\epsilon}^H(\pi)$. The space of values (a subspace of $\mathbb{R}^{|\mathcal{S}|}$) of all policies in $\mathcal{B}_{\epsilon}^H(\pi)$, denoted by $\mathcal{V}^{\mathcal{B}_{\epsilon}^H(\pi)}$, is a (possibly non-convex) polytope.*

In the remaining of this section, we prove a more general version of Theorem 12 as below.

**Theorem 13** (**Policy Subset Polytope**). *For a finite MDP $\mathcal{M}$, consider a connected and compact subset of $\Pi$, denoted as $\mathcal{T}$. The space of values (a subspace of $\mathbb{R}^{|\mathcal{S}|}$) of all policies in $\mathcal{T}$, denoted by $\mathcal{V}^{\mathcal{T}}$, is a (possibly non-convex) polytope.*

According to Lemma 8, $\mathcal{B}_\epsilon^H(\pi)$ is a connected and compact subset of $\Pi$, thus Theorem 12 is a special case of Theorem 13.

**Additional Notations**  To prove Theorem 13, we further define a variant of $Y_{s_1,\cdots,s_k}^\pi$ as $\mathcal{T}_{s_1,\cdots,s_k}^\pi$, which is the set of policies that are in $\mathcal{T}$ and agree with $\pi$ on states $s_1,\cdots,s_k$, i.e.,

$$\mathcal{T}_{s_1,\cdots,s_k}^\pi := \{\pi' \in \mathcal{T} : \pi'(s_i) = \pi(s_i), \forall i = 1,\cdots,k\}.$$

Note that different from $\mathcal{B}_\epsilon^H(\pi)$, $\mathcal{T}$ is no longer restricted under an admissible adversary set and can be any connected and compact subset of $\Pi$.

The following lemma shows that the values of policies in $\mathcal{T}$ that agree on all but one state form a line segment.

**Lemma 14.** *For a policy $\pi \in \mathcal{T}$ and an arbitrary state $s \in \mathcal{S}$, there are two policies in $\partial_\pi \mathcal{T}_{\mathcal{S}\setminus\{s\}}^\pi$, namely $\pi_s^-, \pi_s^+$, such that $\forall \pi' \in \mathcal{T}_{\mathcal{S}\setminus\{s\}}^\pi$,*

$$f_v(\pi_s^-) \preccurlyeq f_v(\pi') \preccurlyeq f_v(\pi_s^+), \tag{3}$$

*where $\preccurlyeq$ denotes element-wise less than or equal to (if $a \preccurlyeq b$, then $a_i \leq b_i$ for all index $i$). Moreover, the image of $f_v$ restricted to $\mathcal{T}_{\mathcal{S}\setminus\{s\}}^\pi$ is a line segment.*

*Proof of Lemma 14.* Lemma 5 in Dadashi et al. (2019) has shown that $f_v$ is infinitely differentiable on $\Pi$, hence we know $f_v(\mathcal{T}_{\mathcal{S}\setminus\{s\}}^\pi)$ is compact and connected. According to Lemma 4 in Dadashi et al. (2019), for any two policies $\pi_1, \pi_2 \in Y_{\mathcal{S}\setminus\{s\}}^\pi$, either $f_v(\pi_1) \preccurlyeq f_v(\pi_2)$, or $f_v(\pi_2) \preccurlyeq f_v(\pi_1)$ (there exists a total order). The same property applies to $\mathcal{T}_{\mathcal{S}\setminus\{s\}}^\pi$ since $\mathcal{T}_{\mathcal{S}\setminus\{s\}}^\pi$ is a subset of $Y_{\mathcal{S}\setminus\{s\}}^\pi$.

Therefore, there exists $\pi_s^-$ and $\pi_s^+$ that achieve the minimum and maximum over all policies in $\mathcal{T}_{\mathcal{S}\setminus\{s\}}^\pi$. Next we show $\pi_s^-$ and $\pi_s^+$ can be found on the outermost boundary of $\mathcal{T}_{\mathcal{S}\setminus\{s\}}^\pi$.

Assume $\pi_s^+ \notin \partial_\pi \mathcal{T}_{\mathcal{S}\setminus\{s\}}^\pi$, and for all $\tilde{\pi} \in \mathcal{T}_{\mathcal{S}\setminus\{s\}}^\pi$, $f_v(\tilde{\pi}) \prec f_v(\pi_s^+)$. Then we can find another policy $\pi' \in \partial_\pi \mathcal{T}_{\mathcal{S}\setminus\{s\}}^\pi$ such that $\pi_s^+ = \alpha\pi + (1-\alpha)\pi'$ for some scalar $\alpha \in (0,1)$. Then according to Lemma 11, $f_v(\pi') \succcurlyeq f_v(\pi_s^+)$, contradicting with the assumption. Therefore, one should be able to find a policy on the outermost boundary of $\mathcal{T}_{\mathcal{S}\setminus\{s\}}^\pi$ whose value dominates all other policies. And similarly, we can also find $\pi_s^-$ on $\partial_\pi \mathcal{T}_{\mathcal{S}\setminus\{s\}}^\pi$.

Furthermore, $f_v(\mathcal{T}_{\mathcal{S}\setminus\{s\}}^\pi)$ is a subset of $f_v(Y_{\mathcal{S}\setminus\{s\}}^\pi)$ since $\mathcal{T}_{\mathcal{S}\setminus\{s\}}^\pi$ is a subset of $Y_{\mathcal{S}\setminus\{s\}}^\pi$. Given that $f_v(Y_{\mathcal{S}\setminus\{s\}}^\pi)$ is a line segment, and $f_v(\mathcal{T}_{\mathcal{S}\setminus\{s\}}^\pi)$ is connected, we can conclude that $f_v(\mathcal{T}_{\mathcal{S}\setminus\{s\}}^\pi)$ is also a line segment.

$\square$

Next, the following lemma shows that $\pi_s^+$ and $\pi_s^-$ and their linear combinations can generate values that cover the set $f_v(\mathcal{T}_{\mathcal{S}\setminus\{s\}}^\pi)$.

**Lemma 15.** *For a policy $\pi \in \mathcal{T}$, an arbitrary state $s \in \mathcal{S}$, and $\pi_s^+, \pi_s^-$ defined in Lemma 14, the following three sets are equivalent:*
*(1) $f_v(\mathcal{T}_{\mathcal{S}\setminus\{s\}}^\pi)$;*
*(2) $f_v\big(closure(\mathcal{T}_{\mathcal{S}\setminus\{s\}}^\pi)\big)$, where $closure(\cdot)$ is the convex closure of a set;*
*(3) $\{f_v(\alpha\pi_s^+ + (1-\alpha)\pi_s^-)|\alpha \in [0,1]\}$;*
*(4) $\{\alpha f_v(\pi_s^+) + (1-\alpha)f_v(\pi_s^-)|\alpha \in [0,1]\}$;*

*Proof of Lemma 15.* We show the equivalence by showing $(1) \subseteq (4) \subseteq (3) \subseteq (2) \subseteq (1)$ as below.

**$(2) \subseteq (1)$:** For any $\pi_1, \pi_2 \in \mathcal{T}_{\mathcal{S}\setminus\{s\}}^\pi$, without loss of generality, suppose $f_v(\pi_1) \preccurlyeq f_v(\pi_2)$. According to Lemma 11, for any $\alpha \in [0,1]$, $f_v(\pi_1) \preccurlyeq \alpha\pi_1 + (1-\alpha)\pi_2 \preccurlyeq f_v(\pi_2)$. Therefore, any convex combinations of policies in $\mathcal{T}_{\mathcal{S}\setminus\{s\}}^\pi$ has value that is in the range of $f_v(\mathcal{T}_{\mathcal{S}\setminus\{s\}}^\pi)$. So the values of policies in the convex closure of $\mathcal{T}_{\mathcal{S}\setminus\{s\}}^\pi$ do not exceed $f_v(\mathcal{T}_{\mathcal{S}\setminus\{s\}}^\pi)$, i.e., $(2) \subseteq (1)$.

**(3) $\subseteq$ (2):** Based on the definition, $\alpha\pi_s^+ + (1-\alpha)\pi_s^- \in closure(\mathcal{T}_{\mathcal{S}\setminus\{s\}}^\pi)$, so (3) $\subseteq$ (2).

**(4) $\subseteq$ (3):** According to Lemma 11, there exists a strictly monotonic rational function $\rho : [0,1] \to \mathbb{R}$, such that

$$l(\alpha) = f_v(\alpha\pi_s^+ + (1-\alpha)\pi_s^-) = \rho(\alpha)f_v(\pi_s^+) + (1-\rho(\alpha))f_v(\pi_s^-).$$

Therefore, due to intermediate value theorem, for $\alpha \in [0,1]$, $\rho(\alpha)$ takes all values from 0 to 1. So (4) = (3).

**(1) $\subseteq$ (4):** Lemma 14 shows that $f_v(\mathcal{T}_{\mathcal{S}\setminus\{s\}}^\pi)$ is a line segment bracketed by $f_v(\pi_s^+)$ and $f_v(\pi_s^-)$. Therefore, for any $\pi' \in \mathcal{T}_{\mathcal{S}\setminus\{s\}}^\pi$, its value is a convex combination of $f_v(\pi_s^+)$ and $f_v(\pi_s^-)$.

$\square$

Next, we show that the relative boundary of the value space constrained to $\mathcal{T}_{s_1,\cdots,s_k}^\pi$ is covered by policies that dominate or are dominated in at least one state. The **relative interior** of set $A$ in $B$ is defined as the set of points in $A$ that have a relative neighborhood in $A \cap B$, denoted as $\text{relint}_B A$. The **relative boundary** of set $A$ in $B$, denoted as $\partial_B A$, is defined as the set of points in $A$ that are not in the relative interior of $A$, i.e., $\partial_B A = A\setminus\text{relint}_B A$. When there is no ambiguity, we omit the subscript of $\partial$ to simplify notations.

In addition, we introduce another notation $F_{s_1,\cdots,s_k}^\pi := V^\pi + span(C_{k+1}^\pi, \cdots, C_{|\mathcal{S}|}^\pi)$, where $C_i^\pi$ stands for the $i$-th column of the matrix $(I - \gamma P^\pi)^{-1}$. Note that $F_{s_1,\cdots,s_k}^\pi$ is the same with $H_{s_1,\cdots,s_k}^\pi$ in Dadashi et al. Dadashi et al. (2019), and we change $H$ to $F$ in order to distinguish from the admissible adversary set $H_\epsilon$ defined in our paper.

**Lemma 16.** *For a policy $\pi \in \mathcal{T}$, $k \leq |\mathcal{S}|$, and a set of policies $\mathcal{T}_{s_1,\cdots,s_k}^\pi$ that agree with $\pi$ on $s_1, \cdots, s_k$ (perturb $\pi$ only at $s_{k+1}, \cdots, s_{|\mathcal{S}|}$), define $\mathcal{V}^t := f_v(\mathcal{T}_{s_1,\cdots,s_k}^\pi)$. Define two sets of policies $X_s^+ := \{\pi' \in \mathcal{T}_{s_1,\cdots,s_k}^\pi : \pi'(\cdot|s) = \pi_s^+(\cdot|s)\}$, and $X_s^- := \{\pi' \in \mathcal{T}_{s_1,\cdots,s_k}^\pi : \pi'(\cdot|s) = \pi_s^-(\cdot|s)\}$. We have that the relative boundary of $\mathcal{V}^t$ in $F_{s_1,\cdots,s_k}^\pi$ is included in the value functions spanned by policies in $\mathcal{T}_{s_1,\cdots,s_k}^\pi \cap (X_{s_j}^+ \cup X_{s_j}^-)$ for at least one $s \notin \{s_1, \cdots, s_k\}$, i.e.,*

$$\partial\mathcal{V}^t \subset \bigcup_{j=k+1}^{|\mathcal{S}|} f_v(\mathcal{T}_{s_1,\cdots,s_k}^\pi \cap (X_{s_j}^+ \cup X_{s_j}^-))$$

*Proof of Lemma 16.* We first prove the following claim:

**Claim 1:** For a policy $\pi_0 \in \mathcal{T}_{s_1,\cdots,s_k}^\pi$, if $\forall j \in \{k+1, \cdots, |\mathcal{S}|\}$, $\nexists\pi' \in closure(\mathcal{T}_{s_1,\cdots,s_k}^\pi) \cap (X_{s_j}^+ \cup X_{s_j}^-)$ such that $f_v(\pi') = f_v(\pi_0)$, then $f_v(\pi_0)$ has a relative neighborhood in $\mathcal{V}^t \cap F_{s_1,\cdots,s_k}^\pi$.

First, based on Lemma 14 and Lemma 15, we can construct a policy $\hat{\pi} \in closure(\mathcal{T}_{s_1,\cdots,s_k}^\pi)$ such that $f_v(\hat{\pi}) = f_v(\pi_0)$ through the following steps:

---

**Algorithm 2:** Constructing $\hat{\pi}$

---
1  Set $\pi^k = \pi_0$
2  **for** $j = k+1, \cdots, |\mathcal{S}|$ **do**
3     Find $\pi_{s_j}^+, \pi_{s_j}^- \in \mathcal{T}_{\mathcal{S}\setminus\{s_j\}}^{\pi_{j-1}}$
4     Find $\pi^j = \hat{\alpha}_j\pi_{s_j}^+ + (1-\hat{\alpha}_j)\pi_{s_j}^-$ such that $f_v(\pi_j) = f_v(\pi_{j-1})$
5  **Return** $\hat{\pi} = \pi^{|\mathcal{S}|}$

---

Denote the concatenation of $\alpha_j$'s as a vector $\hat{\alpha} := [\hat{\alpha}_{k+1}, \cdots, \hat{\alpha}_{|\mathcal{S}|}]$.

According to the assumption that $\forall j \in \{k+1, \cdots, |\mathcal{S}|\}$, $\nexists\pi' \in closure(\mathcal{T}_{s_1,\cdots,s_k}^\pi) \cap (X_{s_j}^+ \cup X_{s_j}^-)$ such that $f_v(\pi') = f_v(\pi_0)$, we have $\hat{\alpha}_j \notin \{0,1\}, \forall j = k+1, \cdots, |\mathcal{S}|$. Then, define a function $\phi : (0,1)^{|\mathcal{S}|-k} \to \mathcal{V}^t$ such that

$$\phi(\alpha) = f_v(\pi_\alpha), \text{ where } \begin{cases} \pi_\alpha(\cdot|s_j) = \alpha\pi_{s_j}^+ + (1-\alpha)\pi_{s_j}^- & \text{if } j \in \{k+1, \cdots, |\mathcal{S}|\} \\ \pi_\alpha(\cdot|s_j) = \hat{\pi}(\cdot|s_j) & \text{otherwise} \end{cases}$$

Then we have that

1. $\phi$ is continuously differentiable.

2. $\phi(\hat{\alpha}) = f_v(\hat{\pi})$.

3. $\frac{\partial \phi}{\partial \alpha_j}$ is non-zero at $\hat{\alpha}$ (because of Lemma 11 (3)).

4. $\frac{\partial \phi}{\partial \alpha_j}$ is along the $i$-the column of $(I - \gamma P^{\hat{\pi}})^{-1}$ (see Lemma 3 in Dadashi et al. Dadashi et al. (2019)).

Therefore, by the inverse theorem function, there is a neighborhood of $\phi(\alpha) = f_v(\hat{\pi})$ in the image space.

Now we have proved Claim 1. As a result, for any policy $\pi_0 \in \mathcal{T}^\pi_{s_1,\cdots,s_k}$, if $f_v(\pi_0)$ is in the relative boundary of $\mathcal{V}^t$ in $F^\pi_{s_1,\cdots,s_k}$, then $\exists j \in \{k+1,\cdots,|\mathcal{S}|\}, \pi' \in closure(\mathcal{T}^\pi_{s_1,\cdots,s_k}) \cap (X^+_{s_j} \cup X^-_{s_j})$ such that $f_v(\pi') = f_v(\pi_0)$. Based on Lemma 15, we can also find $\pi'' \in \mathcal{T}^\pi_{s_1,\cdots,s_k} \cap (X^+_{s_j} \cup X^-_{s_j})$ such that $f_v(\pi'') = f_v(\pi_0)$. So Lemma 16 holds.

$\square$

Now, we are finally ready to prove Theorem 13.

*Proof of Theorem 13.* We will show that $\forall \{s_1, \cdots, s_k\} \subseteq \mathcal{S}$, the value $\mathcal{V}^t = f_v(\mathcal{T}^\pi_{s_1,\cdots,s_k})$ is a polytope.

We prove the above claim by induction on the cardinality of the number of states $k$. In the base case where $k = |\mathcal{S}|$, $\mathcal{V}^t = \{f_v(\pi)\}$ is a polytope.

Suppose the claim holds for $k+1$, then we show it also holds for $k$, i.e., for a policy $\pi \in \Pi$, the value of $\mathcal{T}^\pi_{s_1,\cdots,s_k} \subseteq Y^\pi_{s_1,\cdots,s_k} \subseteq \Pi$ for a polytope.

According to Lemma 16, we have

$$\partial \mathcal{V}^t \subset \bigcup_{j=k+1}^{|\mathcal{S}|} f_v(\mathcal{T}^\pi_{s_1,\cdots,s_k} \cap (X^+_{s_j} \cup X^-_{s_j})) = \bigcup_{j=k+1}^{|\mathcal{S}|} \mathcal{V}^t \cap (F^+_{s_j} \cup F^-_{s_j})$$

where $\partial \mathcal{V}^t$ denotes the relative boundary of $\mathcal{V}^t$ in $F^\pi_{s_1,\cdots,s_k}$; $F^+_{s_j}$ and $F^-_{s_j}$ are two affine hyperplanes of $F^\pi_{s_1,\cdots,s_k}$, standing for the value space of policies that agree with $\pi^+_{s_j}$ and $\pi^-_{s_j}$ in state $s_j$ respectively.

Then we can get

1. $\mathcal{V}^t = f_v(\mathcal{T}^\pi_{s_1,\cdots,s_k})$ is closed as $\mathcal{T}^\pi_{s_1,\cdots,s_k}$ is compact and $f_v$ is continuous.

2. $\partial \mathcal{V}^t \subset \bigcup_{j=k+1}^{|\mathcal{S}|} (F^+_{s_j} \cup F^-_{s_j})$, a finite number of affine hyperplanes in $F^\pi_{s_1,\cdots,s_k}$.

3. $\mathcal{V}^t \cap F^+_{s_j}$ (or $\mathcal{V}^t \cap F^-_{s_j}$) is a polyhedron by induction assumption.

Hence, based on Proposition 1 by Dadashi et al. Dadashi et al. (2019), we get $\mathcal{V}^t$ is a polyhedron. Since $\mathcal{V}^t \subseteq \mathcal{V}$ is bounded, we can further conclude that $\mathcal{V}^t$ is a polytope.

Therefore, for an arbitrary connected and compact set of policies $\mathcal{T} \subseteq \Pi$, let $\pi \in \mathcal{T}$ be an arbitrary policy in $\mathcal{T}$, then $f_v(\mathcal{T}) = f_v(\mathcal{T}^\pi_\emptyset)$ is a polytope.

$\square$

## B.5 EXAMPLES OF THE OUTERMOST BOUNDARY

See Figure 5 for examples of the outermost boundary for different $\mathcal{B}^H_\epsilon(\pi)$'s.

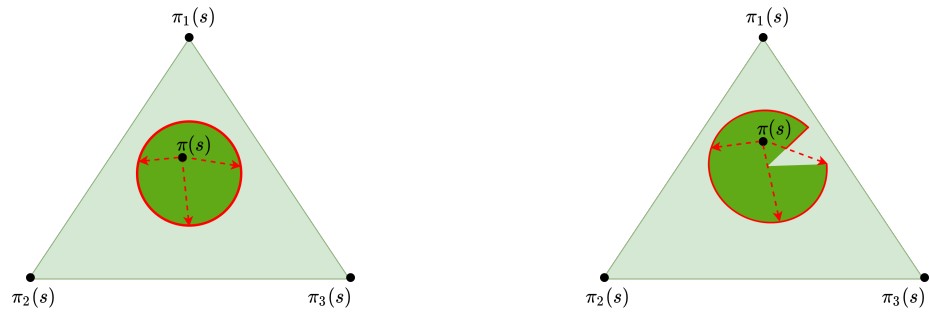

**Figure 5:** Two examples of the outermost boundary with $|\mathcal{A}| = 3$ actions at one single state $s$. The large triangle denotes the distributions over the action space at state $s$, i.e., $\Pi_s$; $\pi_1, \pi_2$ and $\pi_3$ are three policies that deterministically choose $a_1, a_2$ and $a_3$ respectively. $\pi$ is the victim policy, the dark green area is the $\mathcal{B}_\epsilon^H(\pi)_s : \mathcal{B}_\epsilon^H(\pi) \cap \Pi_s$. The red solid curve depicts the outermost boundary of $\mathcal{B}_\epsilon^H(\pi)_s$. Note that a policy is in the outermost boundary of $\mathcal{B}_\epsilon^H(\pi)$ iff it is in the outermost boundary of $\mathcal{B}_\epsilon^H(\pi)_s$ for all $s \in \mathcal{S}$.

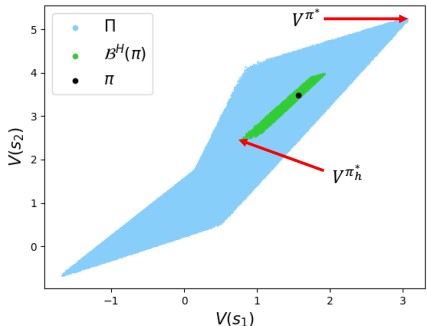

**Figure 6:** Value space of an example MDP. The values of the whole policy space $\Pi$ form a polytope (blue) as suggested by Dadashi et al. (2019). The values of all perturbed policies with $H_\epsilon$ also form a polytope (green) as suggested by Theorem 12.

### B.6 An Example of The Policy Perturbation Polytope

An example is given by Figure 6, where we define an MDP with 2 states and 3 actions. We train an DQN agent with one-hot encodings of the states, and then randomly perturb the states within an $\ell_\infty$ ball with $\epsilon = 0.8$. By sampling 5M random policies, and 100K random perturbations, we visualize the value space of approximately the whole policy space $\Pi$ and the admissible adversarial policy set $\mathcal{B}_\epsilon^H(\pi)$, both of which are polytopes (boundaries are flat). A learning agent searches for the optimal policy $\pi^*$ whose value is the upper right vertex of the larger blue polytope, while the attacker attempts to find an optimal adversary $h^*$, which perturbs a given clean policy $\pi$ to the worst perturbed policy $\pi_{h^*}$ whose value is the lower left vertex of the smaller green polytope. This also justifies the fact that learning an optimal adversary is as difficult as learning an optimal policy in an RL problem.

The example MDP $\mathcal{M}_{\text{ex}}$:

$$|\mathcal{A}| = 3, \gamma = 0.8$$
$$\hat{r} = [-0.1, -1., 0.1, 0.4, 1.5, 0.1]$$
$$\hat{P} = [[0.9, 0.1], [0.2, 0.8], [0.7, 0.3], [0.05, 0.95], [0.25, 0.75], [0.3, 0.7]]$$

The base/clean policy $\pi$:

$$\pi(a_1|s_1) = 0.215, \pi(a_2|s_1) = 0.429, \pi(a_3|s_1) = 0.356$$
$$\pi(a_1|s_2) = 0.271, \pi(a_2|s_2) = 0.592, \pi(a_3|s_2) = 0.137$$

## C  EXTENTIONS AND ADDITIONAL DETAILS OF OUR ALGORITHM

### C.1  ATTACKING A DETERMINISTIC VICTIM POLICY

For a deterministic victim $\pi_D = \text{argmax}_a \pi(a|s)$, we define Deterministic Policy Adversary MDP (D-PAMDP) as below, where a subscript $_D$ is added to all components to distinguish them from their stochastic counterparts. In D-PAMDP, the director proposes a target action $\widehat{a}_D \in \mathcal{A}(=: \widehat{\mathcal{A}}_D)$, and the actor tries its best to let the victim output this target action.

**Definition 17** (**Deterministic Policy Adversary MDP (D-PAMDP)**). *Given an MDP $\mathcal{M} = \langle \mathcal{S}, \mathcal{A}, P, R, \gamma \rangle$, a fixed and deterministic victim policy $\pi_D$, we define a Deterministic Policy Adversarial MDP $\widehat{\mathcal{M}}_D = \langle \mathcal{S}, \widehat{\mathcal{A}}_D, \widehat{P}_D, \widehat{R}_D, \gamma \rangle$, where the action space is $\widehat{\mathcal{A}}_D = \widehat{\mathcal{A}}_D$, and $\forall s, s' \in \mathcal{S}, \forall \widehat{a} \in \mathcal{A}$,*

$$\widehat{P}_D(s'|s, \widehat{a}) = P(s'|s, \pi_D(g(\widehat{a}, s))), \quad \widehat{R}_D(s, \widehat{a}) = -R(s, \pi_D(g(\widehat{a}, s))).$$

*The actor function $g$ is defined as*

$$g_D(\widehat{a}, s) = \text{argmax}_{\tilde{s} \in \mathcal{B}_\epsilon(s)} \big( \pi(\widehat{a}|\tilde{s}) - \max_{a \in \mathcal{A}, a \neq \widehat{a}} \pi(a|\tilde{s}) \big) \qquad (G_D)$$

The optimal policy of D-PAMDP is an optimal adversary against $\pi_D$ as proved in Appendix D.2.2

### C.2  IMPLEMENTATION DETAILS OF PA-AD

To address the actor function $g$ (or $g_D$) defined in ($G$) and ($G_D$), we let the actor maximize objectives $J_D$ and $J$ within the $\mathcal{B}_\epsilon(\cdot)$ ball around the original state, for a deterministic victim and a stochastic victim, respectively. Below we explicitly define $J_D$ and $J$.

**Actor Objective for Deterministic Victim**    For the deterministic variant of PA-AD, the actor function ($G_D$) is simple and can be directly solved to identify the optimal adversary. Concretely, we define the following objective

$$J_D(\tilde{s}; \widehat{a}, s) := \pi(\widehat{a}|\tilde{s}) - \max_{a \in \mathcal{A}, a \neq \widehat{a}} \pi(a|\tilde{s}), \qquad (J_D)$$

which can be realized with the multi-class classification hinge loss. In practice, a relaxed cross-entropy objective can also be used to maximize $\pi(\widehat{a}|\tilde{s})$.

**Actor Objective for Stochastic Victim**    Different from the deterministic-victim case, the actor function for a stochastic victim defined in ($G$) requires solving a more complex optimization problem with a non-convex constraint set, which in practice can be relaxed to ($J$) (a Lagrangian relaxation) to efficiently get an approximation of the optimal adversary.

$$\text{argmax}_{\tilde{s} \in \mathcal{B}_\epsilon(s)} J(\tilde{s}; \widehat{a}, s) := \|\pi(\cdot|\tilde{s}) - \pi(\cdot|s)\| + \lambda \times \text{CosineSim}\big( \pi(\cdot|\tilde{s}) - \pi(\cdot|s), \widehat{a} \big) \qquad (J)$$

where CosineSim in the second refers to the cosine similarity function; the first term measures how far away the policy is perturbed from the victim policy; $\lambda$ is a hyper-parameter controlling the trade-off between the two terms. Experimental results show that our PA-AD is not sensitive to the value of $\lambda$. In our reported results in Section 6, we set $\lambda$ as 1. Appendix E.2.2 shows the evaluation of our algorithm using varying $\lambda$'s.

The procedure of learning the optimal adversary is depicted in Algorithm 3, where we simply use the Fast Gradient Sign Method (FGSM) (Goodfellow et al., 2015) to approximately solve the actor's objective, although more advanced solvers such as Projected Gradient Decent (PGD) can be applied to further improve the performance. Experiment results in Section 6 verify that the above FGSM-based implementation achieves state-of-the-art attack performance.

**What is the Influence of the Relaxation in ($J$)?**  First, it is important that the relaxation is only needed for a stochastic victim. For a deterministic victim, which is often the case in practice, the actor solves the original unrelaxed objective.

Second, as we will discuss in the next paragraph, the optimality of both SA-RL and PA-AD is regarding the formulation. That is, SA-RL and PA-AD formulate the optimal attack problem as an MDP whose optimal policy is the optimal adversary. However, in a large-scale task, deep RL algorithms themselves usually do not converge to the globally optimal policy and exploration becomes the main challenge. Thus, when the adversary's MDP is large, the suboptimality caused by the RL solver due to exploration difficulties could be much more severe than the suboptimality caused by the relaxation of the formulation. The comparison between SA-RL and PA-AD in our experiments

---

**Algorithm 3:** Policy Adversarial Actor Director (PA-AD) with FGSM

---

1  **Input:** Initialization of director's policy $\nu$; victim policy $\pi$; budget $\epsilon$; start state $s_0$
2  **for** $t = 0, 1, 2, ...$ **do**
3      *Director* samples a policy perturbing direction $\widehat{a}_t \sim \nu(\cdot|s_t)$
4      **if** *Victim is deterministic* **then**
5          # for a deterministic victim, $J_D$ is defined in Equation ($J_D$)
6          *Actor* computes the gradient of its objective $\nabla_\delta J_D(s_t + \delta; \widehat{a}_t, s_t)$
7      **else**
8          # for a stochastic victim, $J$ is defined in Equation ($J$)
9          *Actor* computes the gradient of its objective $\nabla_\delta J(s_t + \delta; \widehat{a}_t, s_t)$
10     *Actor* sets $\tilde{s}_t = s_t + \epsilon \cdot \text{sign}(\delta)$
11     *Victim* takes action $a_t \sim \pi(\cdot|\tilde{s}_t)$, proceeds to $s_{t+1}$, receives $r_t$
12     *Director* saves $(s_t, \widehat{a}_t, -r_t, s_{t+1})$ to its buffer
13     *Director* updates its policy $\nu$ using any RL algorithm

---

can justify that the size of the adversary MDP has a larger impact than the relaxation of the problem on the final solution found by the attackers.

Third, in Appendix F.1, we empirically show that with the relaxed objective, PA-AD can still find the optimal attacker in 3 example environments.

**Optimality in Formulation v.s. Approximated Optimality in Practice**  PA-AD has an optimal formulation, as the optimal solution to its objective (the optimal policy in PAMDP) is always an optimal adversary (Theorem 7). Similarly, the previous attack method SA-RL has an optimal solution since the optimal policy in the adversary's MDP is also an optimal adversary. However, in practice where the environments are in a large scale and the number of samples is finite, the optimal policy is not guaranteed to be found by either PA-AD and SA-RL with deep RL algorithms. Therefore, for practical consideration, our goal is to search for a good solution or approximate the optimal solution using optimization techniques (e.g. actor-critic learning, one-step FGSM attack, Lagrangian relaxation for the stochastic-victim attack). In experiments (Section 6), we show that our implementation universally finds stronger attackers than prior methods, which verifies the effectiveness of both our theoretical framework and our practical implementation.

### C.3  Variants For Environments with Continuous Action Spaces

Although the analysis in the main paper focuses on an MDP whose action space is discrete, our algorithm also extends to a continuous action space as justified in our experiments.

#### C.3.1  For A Deterministic Victim

In this case, we can still use the formulation D-PAMDP, but a slightly different actor function

$$g_D(\widehat{a}, s) = \text{argmin}_{\tilde{s} \in \mathcal{B}_\epsilon(s)} \|\pi_D(\tilde{s}) - \widehat{a}\|. \tag{$G_{CD}$}$$

#### C.3.2  For A Stochastic Victim

Different from a stochastic victim in a discrete action space whose actions are sampled from a categorical distribution, a stochastic victim in a continuous action space usually follows a parametrized probability distribution with a certain family of distributions, usually Gaussian distributions. In this case, the formulation of PAMDP in Definition 6 is impractical. However, since the mean of a Gaussian distribution has the largest probability to be selected, one can still use the formulation in ($G_{CD}$), while replacing $\pi_D(\tilde{s})$ with the mean of the output distribution. Then, the director and the actor can collaboratively let the victim output a Gaussian distribution whose mean is the target action. If higher accuracy is needed, we can use another variant of PAMDP, named Continuous Policy Adversary MDP (C-PAMDP) that can also control the variance of the Gaussian distribution.

**Definition 18 (Continuous Policy Adversary MDP (C-PAMDP)).** *Given an MDP $\mathcal{M} = \langle \mathcal{S}, \mathcal{A}, P, R, \gamma \rangle$ where $\mathcal{A}$ is continuous, a fixed and stochastic victim policy $\pi$, we define a Continuous Policy Adversarial MDP $\widehat{\mathcal{M}}_C = \langle \mathcal{S}, \widehat{\mathcal{A}}_C, \widehat{P}_C, \widehat{R}_C, \gamma \rangle$, where the action space is $\widehat{\mathcal{A}}_D = \mathcal{A}$,*

and $\forall s, s' \in \mathcal{S}, \forall \widehat{a} \in \mathcal{A}$,

$$\widehat{P}(s'|s,\widehat{a}) = \int_{\mathcal{A}} \pi(a|g(\widehat{a},s))P(s'|s,a)\,da, \quad \widehat{R}(s,\widehat{a}) = -\int_{\mathcal{A}} \pi(a|g(\widehat{a},s))R(s,a)da.$$

*The actor function g is defined as*

$$g(\widehat{a},s) = \mathrm{argmin}_{\tilde{s} \in \mathcal{B}_{\epsilon}(s)} \mathsf{KL}(\pi(\cdot|\tilde{s})||\mathcal{N}(\widehat{a}, \sigma^2 I_{|\mathcal{A}|})). \tag{$G_C$}$$

*where $\sigma$ is a hyper-parameter, and $\mathcal{N}$ denotes a multivariate Gaussian distribution.*

In short, Equation ($G_C$) encourages the victim to output a distribution that is similar to the target distribution. The hyperparameter $\sigma$ controls the standard deviation of the target distribution. One can set $\sigma$ to be small in order to let the victim execute the target action $\widehat{a}$ with higher probabilities.

## D  CHARACTERIZE OPTIMALITY OF EVASION ATTACKS

In this section, we provide a detailed characterization for the optimality of evasion attacks from the perspective of policy perturbation, following Definition 5 in Section 4. Section D.1 establishes the existence of the optimal policy adversary which is defined in Section 3. Section D.2 then provides a proof for Theorem 7 that the formulation of PA-AD is optimal. We also analyze the optimality of heuristic attacks in Section D.3.

### D.1  EXISTENCE OF AN OPTIMAL POLICY ADVERSARY

**Theorem 19** (Existence of An Optimal Policy Adversary). *Given an MDP $\mathcal{M} = \langle \mathcal{S}, \mathcal{A}, P, R, \gamma \rangle$, and a fixed stationary policy $\pi$ on $\mathcal{M}$, let $H_\epsilon$ be a non-empty set of admissible state adversaries and $\mathcal{B}_\epsilon^H(\pi)$ be the corresponding Adv-policy-set, then there exists an optimal policy adversary $\pi_{h^*} \in \mathcal{B}_\epsilon^H(\pi)$ such that $\pi_{h^*} \in \mathrm{argmin}_{\pi_h \in \mathcal{B}_\epsilon^H(\pi)} V_{\mathcal{M}}^{\pi_h}(s), \forall s \in \mathcal{S}$.*

*Proof.* We prove Theorem 19 by constructing a new MDP corresponding to the original MDP $\mathcal{M}$ and the victim $\pi$.

**Definition 20** (**Policy Perturbation MDP**). *For a given MDP $\mathcal{M}$, a fixed stochastic victim policy $\pi$, and an admissible state adversary set $H_\epsilon$, define a policy perturbation MDP as $\mathcal{M}_P = \langle \mathcal{S}, \mathcal{A}_P, P_P, R_P, \gamma \rangle$, where $\mathcal{A}_P = \Delta(\mathcal{A})$, and $\forall s \in \mathcal{S}, a_P \in \mathcal{A}_P$,*

$$R_P(s, a_P) := \begin{cases} -\sum_{a \in \mathcal{A}} a_P(a|s)R(s,a) & \textit{if } \exists h \in H_\epsilon \textit{ s.t. } a_P(\cdot|s) = \pi(\cdot|h(s)) \\ -\infty & \textit{otherwise} \end{cases} \tag{4}$$

$$P_P(s'|s, a_P) := \sum_{a \in \mathcal{A}} a_P(a|s)P(s'|s,a) \tag{5}$$

Then we can prove Theorem 19 by proving the following lemma.

**Lemma 21.** *The optimal policy in $\mathcal{M}_P$ is an optimal policy adversary for $\pi$ in $\mathcal{M}$.*

Let $N_P$ denote the set of deterministic policies in $\mathcal{M}_P$. According to the traditional MDP theory (Puterman, 1994), there exists a deterministic policy that is optimal in $\mathcal{M}_P$. Note that $H_\epsilon$ is non-empty, so there exists at least one policy in $\mathcal{M}_P$ with value $\geq -\infty$, and then the optimal policy should have value $\geq -\infty$. Denote this optimal and deterministic policy as $\nu_P^* \in N_P$. Then we write the Bellman equation of $\nu_P^*$, i.e.,

$$\begin{aligned} V_P^{\nu_P^*}(s) &= \max_{\nu_P \in N_P} R_P(s, \nu_P(s)) + \gamma \sum_{s' \in \mathcal{S}} P_P(s'|s, \nu_P(s))V_P^{\nu_P}(s') \\ &= \max_{\nu_P \in N_P} \left[ -\sum_{a \in \mathcal{A}} \nu_P(a|s)R(s,a) + \gamma \sum_{s' \in \mathcal{S}} \sum_{a \in \mathcal{A}} \nu_P(a|s)P(s'|s,a)V_P^{\nu_P}(s') \right] \\ &= \max_{\nu_P \in N_P} \sum_{a \in \mathcal{A}} \nu_P(a|s) \left[ -R(s,a) + \gamma \sum_{s' \in \mathcal{S}} P(s'|s,a)V_P^{\nu_P}(s') \right] \end{aligned} \tag{6}$$

Note that $\nu_P^*(s)$ is a distribution on action space, $\nu_P^*(a|s)$ is the probability of $a$ given by distribution $\nu^*(s)$.

Multiply both sides of Equation (6) by $-1$, and we obtain

$$-V_P^{\nu_P^*}(s) = \min_{\nu \in N_P} \sum_{a \in \mathcal{A}} \nu_P(s)(a|s) \left[ R(s,a) + \gamma \sum_{s' \in \mathcal{S}} P(s'|s,a)\big(-V_P^{\nu_P}(s')\big) \right] \qquad (7)$$

In the original MDP $\mathcal{M}$, an optimal policy adversary (if exists) $\pi_{h^*}$ for $\pi$ should satisfy

$$V^{\pi_{h^*}}(s) = \min_{\pi_h \in \mathcal{B}_\epsilon^H(\pi)} \sum_{a \in \mathcal{A}} \pi_h(a|s) \left[ R(s,a) + \gamma \sum_{s' \in \mathcal{S}} P(s'|s,a)V^{\pi_h}(s') \right] \qquad (8)$$

By comparing Equation (7) and Equation (8) we get the conclusion that $\nu_P^*$ is an optimal policy adversary for $\pi$ in $\mathcal{M}$.

$\qquad \square$

## D.2 PROOF OF THEOREM 7: OPTIMALITY OF OUR PA-AD

In this section, we provide theoretical proof of the optimality of our proposed evasion RL algorithm PA-AD.

### D.2.1 OPTIMALITY OF PA-AD FOR A STOCHASTIC VICTIM

We first build a connection between the PAMDP $\widehat{\mathcal{M}}$ defined in Definition 6 (Section 4) and the policy perturbation MDP defined in Definition 20 (Appendix D.1).

A deterministic policy $\nu$ in the PAMDP $\widehat{\mathcal{M}}$ can **induce** a policy $\hat{\nu}_P$ in $\mathcal{M}_P$ in the following way: $\widehat{\nu}_P(s) = \pi(\cdot|g(\nu(s),s)), \forall s \in \mathcal{S}$. More importantly, the values of $\nu$ and $\widehat{\nu}_P$ in $\widehat{\mathcal{M}}$ and $\mathcal{M}_P$ are equal because of the formulations of the two MDPs, i.e., $\widehat{V}^\nu = V_P^{\widehat{\nu}_P}$, where $\widehat{V}$ and $V_P$ denote the value functions in $\widehat{\mathcal{M}}$ and $V_P$ respectively.

Proposition 22 below builds the connection of the optimality between the policies in these two MDPs.

**Proposition 22.** *An optimal policy in $\widehat{\mathcal{M}}$ induces an optimal policy in $\mathcal{M}_P$.*

*Proof of Proposition 22.* Let $\nu^*$ be an deterministic optimal policy in $\widehat{\mathcal{M}}$, and it induces a policy in $\mathcal{M}_P$, namely $\widehat{\nu}_P$.

Let us assume $\widehat{\nu}_P$ is not an optimal policy in $\mathcal{M}_P$, hence there exists a policy $\nu_P^*$ in $\mathcal{M}_P$ s.t. $V_P^{\nu_P^*}(s) > V_P^{\widehat{\nu}_P}(s)$ for at least one $s \in \mathcal{S}$. And based on Theorem 4, we are able to find such a $\nu_P^*$ whose corresponding policy perturbation is on the outermost boundary of $\mathcal{B}(\pi)$, i.e., $\nu^* \in \partial_\pi \mathcal{B}_\epsilon^H(\pi)$.

Then we can construct a policy $\nu'$ in $\widehat{\mathcal{M}}$ such that $\nu'(s) = \nu_P^*(s) - \pi(s), \forall s \in \mathcal{S}$. And based on Equation (G), $\pi(\cdot|g(\nu'(s),s))$ is in $\partial_\pi \mathcal{B}(\pi(s))$ for all $s \in \mathcal{S}$. According to the definition of $\partial_\pi$, if two policy perturbations perturb $\pi$ in the same direction and are both on the outermost boundary, then they are equal. Thus, we can conclude that $\pi(g(\nu'(s),s)) = \nu_P^*(s), \forall s \in \mathcal{S}$. Then we obtain $\widehat{V}^{\nu'}(s) = V_P^{\nu_P^*}(s), \forall s \in \mathcal{S}$.

Now we have conditions:
(1) $\widehat{V}^{\nu^*}(s) = V_P^{\widehat{\nu}_P}(s), \forall s \in \mathcal{S}$;
(2) $V_P^{\nu_P^*}(s) > V_P^{\widehat{\nu}_P}(s)$ for at least one $s \in \mathcal{S}$;
(3) $\exists \nu'$ such that $\widehat{V}^{\nu'}(s) = V_P^{\nu_P^*}(s), \forall s \in \mathcal{S}$.

From (1), (2) and (3), we can conclude that $\widehat{V}^{\nu'}(s) > \widehat{V}^{\nu^*}(s)$ for at least one $s \in \mathcal{S}$, which conflicts with the assumption that $\nu^*$ is optimal in $\widehat{\mathcal{M}}$. Therefore, Proposition 22 is proven.

□

Proposition 22 and Lemma 21 together justifies that the optimal policy of $\widehat{\mathcal{M}}$, namely $\nu^*$, induces an optimal policy adversary for $\pi$ in the original $\mathcal{M}$. Then, if the director learns the optimal policy in $\widehat{\mathcal{M}}$, then it collaborates with the actor and generates the optimal state adversary $h^*$ by $h^*(s) = g(\nu^*(s), s), \forall s \in \mathcal{S}$.

### D.2.2 OPTIMALITY OF OUR PA-AD FOR A DETERMINISTIC VICTIM

In this section, we show that the optimal policy in D-PAMDP (the deterministic variant of PAMDP defined in Appendix C.1) also induces an optimal policy adversary in the original environment.

Let $\pi_D$ be a deterministic policy reduced from a stochastic policy $\pi$, i.e.,
$$\pi_D(s) := \text{argmax}_{a \in \mathcal{A}} \pi(a|s), \forall s \in \mathcal{S}.$$
Note that in this case, the Adv-policy-set $\mathcal{B}_\epsilon^H(\pi)$ is not connected as it contains only deterministic policies. Therefore, we re-formulate the policy perturbation MDP introduced in Appendix D.1 with a deterministic victim as below:

**Definition 23** (**Deterministic Policy Perturbation MDP**). *For a given MDP $\mathcal{M}$, a fixed deterministic victim policy $\pi$, and an admissible adversary set $H_\epsilon$, define a deterministic policy perturbation MDP as $\mathcal{M}_{DP} = \langle \mathcal{S}, \mathcal{A}_{DP}, P_{DP}, R_{DP}, \gamma \rangle$, where $\mathcal{A}_{DP} = \mathcal{A}$, and $\forall s \in \mathcal{S}, a_{DP} \in \mathcal{A}_{DP}$,*

$$R_{DP}(s, a_{DP}) := \{ \begin{array}{ll} -R(s, a_{DP}) & \text{if } \exists h \in H_\epsilon \text{ s.t. } a_{DP}(s) = \pi_D(h(s)) \\ -\infty & \text{otherwise} \end{array} \tag{9}$$

$$P_{DP}(s'|s, a_{DP}) := P(s, a_{DP}) \tag{10}$$

$\mathcal{M}_{DP}$ can be viewed as a special case of $\mathcal{M}_P$ where only deterministic policies have $\geq -\infty$ values. Therefore Theorem 19 and Lemma 21 also hold for deterministic victims.

Next we will show that an optimal policy in $\widehat{\mathcal{M}}_D$ induces an optimal policy in $\mathcal{M}_{DP}$.

**Proposition 24.** *An optimal policy in $\widehat{\mathcal{M}}_D$ induces an optimal policy in $\mathcal{M}_{DP}$.*

*Proof of Proposition 24.* We will prove Proposition 24 by contradiction. Let $\nu^*$ be an optimal policy in $\widehat{\mathcal{M}}_D$, and it induces a policy in $\mathcal{M}_{DP}$, namely $\widehat{\nu}_{DP}$.

Let us assume $\widehat{\nu}_{DP}$ is not an optimal policy in $\mathcal{M}_{DP}$, hence there exists a deterministic policy $\nu^*_{DP}$ in $\mathcal{M}_{DP}$ s.t. $V_{DP}^{\nu^*_{DP}}(s) > V_{DP}^{\widehat{\nu}_{DP}}(s)$ for at least one $s \in \mathcal{S}$. Without loss of generality, suppose $V_{DP}^{\nu^*_{DP}}(s_0) > V_{DP}^{\widehat{\nu}_{DP}}(s_0)$.

Next we construct another policy $\nu'$ in $\widehat{\mathcal{M}}_D$ by setting $\nu'(s) = \nu^*_{DP}(s), \forall s \in \mathcal{S}$. Given that $\nu^*_{DP}$ is deterministic, $\nu'$ is also a deterministic policy. So we use $\nu^*_{DP}(s)$ and $\nu'(s)$ to denote the action selected by $\nu^*_{DP}$ and $\nu'$ respectively at state $s$.

For an arbitrary state $s_i$, let $a_i := \nu^*_{DP}(s_i)$. Since $\nu^*_{DP}$ is the optimal policy in $\mathcal{M}_{DP}$, we get that there exists a state adversary $h \in H_\epsilon$ such that $\pi_D(h(s_i)) = a_i$, or equivalently, there exists a state $\tilde{s}_i \in \mathcal{B}_\epsilon(s_i)$ such that $\text{argmax}_{a \in \mathcal{A}} \pi(\tilde{s}_i) = a_i$. Then, the solution to the actor's optimization problem ($G_D$) given direction $a_i$ and state $s_i$, denoted as $\tilde{s}^*$, satisfies

$$\tilde{s}^* = \text{argmax}_{s' \in B_\epsilon(s)} \left( \pi(\widehat{a}|s') - \text{argmax}_{a \in \mathcal{A}, a \neq \widehat{a}} \pi(a|s') \right) \tag{11}$$

and we can get

$$\pi(\widehat{a}|\tilde{s}^*) - \text{argmax}_{a \in \mathcal{A}, a \neq \widehat{a}} \pi(a|\tilde{s}^*) \geq \pi(\widehat{a}|\tilde{s}_i) - \text{argmax}_{a \in \mathcal{A}, a \neq \widehat{a}} \pi(a|\tilde{s}_i) > 0 \tag{12}$$

Given that $\text{argmax}_{a \in \mathcal{A}} \pi(a_i|\tilde{s}_i) = a_i$, we obtain $\text{argmax}_{a \in \mathcal{A}} \pi(a_i|\tilde{s}^*) = a_i$, and hence $\pi_D(g_D(a_i, s_i)) = a_i$. Since this relation holds for an arbitrary state $s$, we can get

$$\pi_D(g_D(\nu'(s), s)) = \pi_D(g_D(\nu'(s), s)) = \nu'(s), \forall s \in \mathcal{S} \tag{13}$$

Also, we have $\forall s \in \mathcal{S}$

$$\widehat{V}_D^{\nu'}(s) = \widehat{R}_D(s, \nu'(s)) + \sum_{s' \in \mathcal{S}} \widehat{P}_D(s'|s, \nu'(s))\widehat{V}_D^{\nu'}(s') \tag{14}$$

$$V_{DP}^{\nu_{DP}^*}(s) = R_{DP}(s, \nu_{DP}^*(s)) + \sum_{s' \in \mathcal{S}} P_{DP}(s'|s, \nu_{DP}^*(s))V_{DP}^{\nu_{DP}^*}((s')) \tag{15}$$

Therefore, $\widehat{V}_D^{\nu'}(s) = V_{DP}^{\nu_{DP}^*}(s), \forall s \in \mathcal{S}$.

Then we have

$$\widehat{V}_D^{\nu'}(s_0) \le \widehat{V}^{\nu^*}(s_0) = V_{DP}^{\widehat{\nu}_{DP}}(s_0) < V_{DP}^{\nu_{DP}^*}(s_0) = \widehat{V}_D^{\nu'}(s_0) \tag{16}$$

which gives $\widehat{V}_D^{\nu'}(s_0) < \widehat{V}_D^{\nu'}(s_0)$, so there is a contradiction.

$\square$

Combining the results of Proposition 24 and Lemma 21 , for a deterministic victim, the optimal policy in D-PAMDP gives an optimal adversary for the victim.

### D.3 OPTIMALITY OF HEURISTIC-BASED ATTACKS

There are many existing methods of finding adversarial state perturbations for a fixed RL policy, most of which are solving some optimization problems defined by heuristics. Although these methods are empirically shown to be effective in many environments, it is not clear how strong these adversaries are in general. In this section, we carefully summarize and categorize existing heuristic attack methods into 4 types, and then characterize their optimality in theory.

#### D.3.1 *TYPE I* - MINIMIZE THE BEST (MINBEST)

A common idea of evasion attacks in supervised learning is to reduce the probability that the learner selects the "correct answer" Goodfellow et al. (2015). Prior works Huang et al. (2017); Kos & Song (2017); Korkmaz (2020) apply a similar idea to craft adversarial attacks in RL, where the objective is to minimize the probability of selecting the "best" action, i.e.,

$$h^{\text{MinBest}} \in \operatorname{argmin}_{h \in H_\epsilon} \pi_h(a^+|s), \forall s \in \mathcal{S} \tag{I}$$

where $a^+$ is the "best" action to select at state $s$. Huang et al. Huang et al. (2017) define $a^+$ as $\operatorname{argmax}_{a \in \mathcal{A}} Q^\pi(s, a)$ for DQN, or $\operatorname{argmax}_{a \in \mathcal{A}} \pi(a|s)$ for TRPO and A3C with a stochastic $\pi$. Since the agent's policy $\pi$ is usually well-trained in the original MDP, $a^+$ can be viewed as (approximately) the action taken by an optimal deterministic policy $\pi^*(s)$.

**Lemma 25 (Optimality of MinBest).** *Denote the set of optimal solutions to objective* (I) *as* $H^{MinBest}$. *There exist an MDP* $\mathcal{M}$ *and an agent policy* $\pi$, *such that* $H^{MinBest}$ *does not contain an optimal adversary* $h^*$, *i.e.,* $H^{MinBest} \cap H_\epsilon^* = \emptyset$.

*Proof of Lemma 25.* We prove this lemma by constructing the following MDP such that for any victim policy, there exists a reward configuration in which MinBest attacker is not optimal.

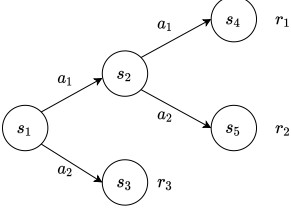

**Figure 7:** A simple MDP where MinBest Attacker cannot find the optimal adversary for a given victim policy.

Here, let $r_1 = r(s_4|s_2, a_1), r_2 = r(s_5|s_2, a_2), r_3 = r(s_3|s_1, a_2)$. Assuming all the other rewards are zero, transition dynamics are deterministic, and states $s_3, s_4, s_5$ are the terminal states. For the

sake of simplicity, we also assume that the discount factor here $\gamma = 1$.

Now given a policy $\pi$ such that $\pi(a_1|s_1) = \beta_1$ and $\pi(a_1|s_2) = \beta_2$ ($\beta_1, \beta_2 \in [0, 1]$), we could find $r_1, r_2, r_3$ such that the following constraints hold:

$$r_1 > r_2 \iff Q^\pi(s_1, a_1) > Q^\pi(s_1, a_2) \tag{17}$$

$$\beta_2 r_1 + (1 - \beta_2)r_2 > r_3 \iff Q^\pi(s_2, a_1) > Q^\pi(s_2, a_2) \tag{18}$$

$$r_3 > (\beta_2 - \epsilon_2)r_2 + (1 - \beta_2 + \epsilon_2)r_2 \iff r_3 > Q^\pi(s_1, a_1) - \epsilon_2(r_1 - r_2) \tag{19}$$

Now we consider the Adv-policy-set

$$\mathcal{B}_\epsilon^H(\pi) = \left\{ \pi' \in \Pi \,\middle|\, \|\pi'(\cdot|s_1) - \pi(\cdot|s_1)\| < \epsilon_1, \|\pi'(\cdot|s_2) - \pi(\cdot|s_2)\| < \epsilon_2 \right\}.$$

Under these three linear constraints, the policy given by MinBest attacker satisfies that $\pi_{h^{\mathrm{MinBest}}}(a_1|s_1) = \beta_1 - \epsilon_1$, and $\pi_{h^{\mathrm{MinBest}}}(a_1|s_2) = \beta_2 - \epsilon_2$. On the other hand, we can find another admissible policy adversary $\pi_{h^*}(a_1|s_1) = \beta_1 + \epsilon_1$, and $\pi_{h^*}(a_1|s_2) = \beta_2 - \epsilon_2$. Now we show that $V^{\pi_{h^*}}(s_1) < V^{\pi_{h^{\mathrm{MinBest}}}}(s_1)$, and thus MinBest attacker is not optimal.

$$V^{\pi_{h^{\mathrm{MinBest}}}}(s_1) = (\beta_1 - \epsilon_1)\Big[(\beta_2 - \epsilon_2)r_1 + (1 - \beta_2 + \epsilon_2)r_2\Big] + (1 - \beta_1 + \epsilon_1)r_3 \tag{20}$$

$$= (\beta_1 - \epsilon_1)(\beta_2 - \epsilon_2)r_1 + (\beta_1 - \epsilon_1)(1 - \beta_2 + \epsilon_2)r_2 + (1 - \beta_1 + \epsilon_1)r_3 \tag{21}$$

$$V^{\pi_{h^*}}(s_1) = (\beta_1 + \epsilon_1)\Big[(\beta_2 - \epsilon_2)r_1 + (1 - \beta_2 + \epsilon_2)r_2\Big] + (1 - \beta_1 - \epsilon_1)r_3 \tag{22}$$

$$= (\beta_1 + \epsilon_1)(\beta_2 - \epsilon_2)r_1 + (\beta_1 + \epsilon_1)(1 - \beta_2 + \epsilon_2)r_2 + (1 - \beta_1 - \epsilon_1)r_3 \tag{23}$$

Therefore,

$$V^{\pi_{h^*}}(s_1) - V^{\pi_{h^{\mathrm{MinBest}}}}(s_1) = 2\epsilon_1(\beta_2 - \epsilon_2)r_2 + 2\epsilon_1(1 - \beta_2 + \epsilon_2)r_2 - 2\epsilon_1 r_3 \tag{24}$$

$$= 2\epsilon_1\Big[(\beta_2 - \epsilon_2)r_2 + (1 - \beta_2 + \epsilon_2)r_2 - r_3\Big] \tag{25}$$

$$< 0 \quad \text{Because of the constraint (19)} \tag{26}$$

$$\square$$

### D.3.2 *Type II* - Maximize The Worst (MaxWorst)

Pattanaik et al. (Pattanaik et al., 2018) point out that only preventing the agent from selecting the best action does not necessarily result in a low total reward. Instead, Pattanaik et al. (Pattanaik et al., 2018) propose another objective function which maximizes the probability of selecting the worst action, i.e.,

$$h^{\mathrm{MaxWorst}} \in \mathrm{argmax}_{h \in H_\epsilon} \pi_h(a^-|s), \forall s \in \mathcal{S} \tag{II}$$

where $a^-$ refers to the "worst" action at state $s$. Pattanaik et al.(Pattanaik et al., 2018) define the "worst" action as the actions with the lowest Q value, which could be ambiguous, since the Q function is policy-dependent. If a worst policy $\pi^- \in \mathrm{argmin}_\pi V^\pi(s), \forall s \in \mathcal{S}$ is available, one can use $a^- = \mathrm{argmin} Q^{\pi^-}(s, a)$. However, in practice, the attacker usually only has access to the agent's current policy $\pi$, so it can also choose $a^- = \mathrm{argmin} Q^\pi(s, a)$. Note that these two selections are different, as the agent's policy $\pi$ is usually far away from the worst policy.

**Lemma 26 (Optimality of MaxWorst).** *Denote the set of optimal solutions to objective* (II) *as* $H^{MaxWorst}$, *which include both versions of MaxWorst attacker formulations as we discussed above. Then there exist an MDP $\mathcal{M}$ and an agent policy $\pi$, such that $H^{MaxWorst}$ contains a non-optimal adversary $h^*$, i.e., $H^{MaxWorst} \not\subset H_\epsilon^*$.*

*Proof of Lemma 26.*

**Case I: Using current policy to compute the target action**

We prove this lemma by constructing the MDP in Figure 8 such that for any victim policy, there exists a reward configuration in which MaxWorst attacker is not optimal.

Here, let $r_1 = r(s_{11}|s_1, a_1), r_2 = r(s_{12}|s_1, a_2), r_3 = r(s_{21}|s_2, a_1), r_4 = r(s_{22}|s_2, a_2)$. Assuming all the other rewards are zero, transition dynamics are deterministic, and states $s_{11}, s_{12}, s_{21}, s_{22}$ are the terminal states. For the sake of simplicity, we also assume that the discount factor here $\gamma = 1$.

Now given a policy $\pi$ such that $\pi(a_1|s_0) = \beta_0, \pi(a_1|s_1) = \beta_1$, and $\pi(a_2|s_2) = \beta_2$ ($\beta_0, \beta_1, \beta_2 \in [0, 1]$), consider the Adv-policy-set

$$\mathcal{B}_\epsilon^H(\pi) = \left\{ \pi' \in \Pi \,\middle|\, \|\pi'(\cdot|s_1) - \pi(\cdot|s_1)\| < \epsilon_0, \|\pi'(\cdot|s_1) - \pi(\cdot|s_1)\| < \epsilon_1, \|\pi'(\cdot|s_2) - \pi(\cdot|s_2)\| < \epsilon_2, \right\}.$$

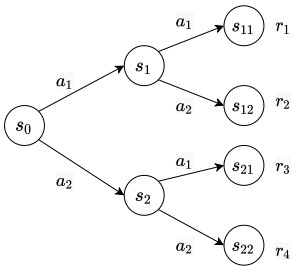

**Figure 8:** A simple MDP where the first version of MaxWorst Attacker cannot find the optimal adversary for a given victim policy.

We could find $r_1, r_2, r_3, r_4$ such that the following linear constraints hold:

$$\beta_1 r_1 + (1 - \beta_1)r_2 > \beta_2 r_3 + (1 - \beta_2)r_4 \iff Q^\pi(s_0, a_1) > Q^\pi(s_0, a_2) \tag{27}$$

$$r_1 > r_2 \iff Q^\pi(s_1, a_1) > Q^\pi(s_1, a_2) \tag{28}$$

$$r_3 > r_4 \iff Q^\pi(s_2, a_1) > Q^\pi(s_2, a_2) \tag{29}$$

$$(\beta_1 - \epsilon_1)r_1 + (1 - \beta_1 + \epsilon_1)r_2 < (\beta_2 - \epsilon_2)r_3 + (1 - \beta_2 + \epsilon_2)r_4 \tag{30}$$

Now, given these constraints, the perturbed policy given by MaxWorst attaker satisfies $\pi_{h\text{MaxWorst}}(a_1|s_0) = \beta_0 - \epsilon_0$, $\pi_{h\text{MaxWorst}}(a_1|s_1) = \beta_1 - \epsilon_1$, and $\pi_{h\text{MaxWorst}}(a_1|s_2) = \beta_2 - \epsilon_2$. However, consider another perturbed policy $\pi_{h^*}$ in Adv-policy-set such that $\pi_{h^*}(a_1|s_0) = \beta_0 + \epsilon_0$, $\pi_{h^*}(a_1|s_1) = \beta_1 - \epsilon_1$, and $\pi_{h^*}(a_1|s_2) = \beta_2 - \epsilon_2$. We will prove that $V^{\pi_{h^*}}(s_1) < V^{\pi_{h\text{MaxWorst}}}(s_1)$, and thus MaxWorst attacker is not optimal.
On the one hand,

$$V^{\pi_{h\text{MaxWorst}}}(s_1) = (\beta_0 - \epsilon_0)\Big[(\beta_1 - \epsilon_1)r_1 + (1 - \beta_1 + \epsilon_1)r_2\Big] + (1 - \beta_0 + \epsilon_0)\Big[(\beta_2 - \epsilon_2)r_3 + (1 - \beta_2 + \epsilon_2)r_4\Big] \tag{31}$$

$$= (\beta_0 - \epsilon_0)(\beta_1 - \epsilon_1)r_1 + (\beta_0 - \epsilon_0)(1 - \beta_1 + \epsilon_1)r_2$$
$$+ (1 - \beta_0 + \epsilon_0)(\beta_2 - \epsilon_2)r_3 + (1 - \beta_0 + \epsilon_0)(1 - \beta_2 + \epsilon_2)r_4 \tag{32}$$

On the other hand,

$$V^{\pi_{h^*}}(s_1) = (\beta_0 + \epsilon_0)\Big[(\beta_1 - \epsilon_1)r_1 + (1 - \beta_1 + \epsilon_1)r_2\Big] + (1 - \beta_0 - \epsilon_0)\Big[(\beta_2 - \epsilon_2)r_3 + (1 - \beta_2 + \epsilon_2)r_4\Big] \tag{33}$$

$$= (\beta_0 + \epsilon_0)(\beta_1 - \epsilon_1)r_1 + (\beta_0 + \epsilon_0)(1 - \beta_1 + \epsilon_1)r_2$$
$$+ (1 - \beta_0 - \epsilon_0)(\beta_2 - \epsilon_2)r_3 + (1 - \beta_0 - \epsilon_0)(1 - \beta_2 + \epsilon_2)r_4 \tag{34}$$

Therefore,

$$V^{\pi_{h^*}}(s_1) - V^{\pi_{h\text{MaxWorst}}}(s_1) = 2\epsilon_0(\beta_1 - \epsilon_1)r_1 + 2\epsilon_0(1 - \beta_1 + \epsilon_1)r_2$$
$$- 2\epsilon_0(\beta_2 - \epsilon_2)r_3 - 2\epsilon_0(1 - \beta_2 + \epsilon_2)r_4 \tag{35}$$

$$< 0 \quad \text{Because of the constraint (30)} \tag{36}$$

**Case II: Using worst policy to compute the target action**

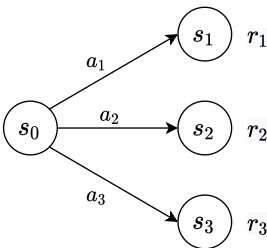

**Figure 9:** A simple MDP where the second version of MaxWorst Attacker cannot find the optimal adversary for a given victim policy.

Same as before, we construct a MDP where $H^{\text{MaxWorst}}$ contains a non-optimal adversary. Let $r_1 = r(s_1|s_0, a_1), r_2 = r(s_2|s_0, a_2), r_3 = r(s_3|s_0, a_3)$. Assuming all the other rewards are zero, transition dynamics are deterministic, and states $s_1, s_2, s_3$ are the terminal states. For the sake of simplicity, we also assume that the discount factor here $\gamma = 1$.

Let $pi$ be the given policy such that $\pi(a_1|s_0) = \beta_1$ and $\pi(a_2|s_0) = \beta_2$. Now without loss of generality, we assume $r_1 > r_2 > r_3$ (∗). Then the worst policy $\pi'$ satisfies that $\pi'(a_3|s_0) = 1$. Consider the Adv-policy-set $\mathcal{B}_\epsilon^H(\pi) = \left\{ \pi' \in \Pi \mid \|\pi'(\cdot|s_0) - \pi(\cdot|s_0)\|_1 < \epsilon \right\}$. Then $H^{\text{MaxWorst}} = \left\{ \pi' \in \Pi \mid \pi'(a_3|s_0) = (1 - \beta_1 - \beta_2) + \epsilon \right\}$.

Now consider two policies $\pi_{h^1}, \pi_{h^2} \in H^{\text{MaxWorst}}$, where $\pi_{h^1}(a_1|s_0) = \beta_1$, $\pi_{h^1}(a_2|s_0) = \beta_2 - \epsilon$, $\pi_{h^2}(a_1|s_0) = \beta_1 - \epsilon$, $\pi_{h^2}(a_2|s_0) = \beta_2$. Then $V^{\pi_{h^1}}(s_0) - V^{\pi_{h^2}}(s_0) = \epsilon(r_1 - r_2) > 0$. Therefore, $\pi_{h^1} \in H^{\text{MaxWorst}}$ but it's not optimal.

$\square$

### D.3.3 *TYPE III* - MINIMIZE Q VALUE (MINQ).

Another idea of attacking Pattanaik et al. (2018); Zhang et al. (2020a) is to craft perturbations such that the agent selects actions with minimized Q values at every step, i.e.,

$$h^{\text{MinQ}} \in \text{argmin}_{h \in H_\epsilon} \sum_{a \in \mathcal{A}} \pi_h(a|s) \hat{Q}^\pi(s, a), \forall s \in \mathcal{S} \tag{III}$$

where $\hat{Q}$ is the approximated Q function of the agent's original policy. For example, Pattanaik et al.Pattanaik et al. (2018) directly use the agent's Q network (of policy $\pi$), while the Robust SARSA (RS) attack proposed by Zhang et al.Zhang et al. (2020a) learns a more stable Q network for the agent's policy $\pi$. Note that in practice, this type of attack is usually applied to deterministic agents (e.g., DQN, DDPG, etc), then the objective becomes $\text{argmin}_{h \in H_\epsilon} \hat{Q}^\pi(s, \pi_h(s)), \forall s \in \mathcal{S}$ Pattanaik et al. (2018); Zhang et al. (2020a); Oikarinen et al. (2020). In this case, the MinQ attack is equivalent to the MaxWorst attack with the current policy as the target.

**Lemma 27** (**Optimality of MinQ**). *Denote the set of optimal solutions to objective* (III) *as $H^{MinQ}$, which include both versions of MinQ attacker formulations as we discussed above. Then there exist an MDP $\mathcal{M}$ and an agent policy $\pi$, such that $H^{MinQ}$ contains a non-optimal adversary $h^*$, i.e., $H^{MinQ} \not\subset H_\epsilon^*$.*

*Proof of Lemma 26.*
**Case I: For a deterministic victim**
In the deterministic case

$$h^{\text{MinQ}} \in \text{argmin}_{h \in H_\epsilon} \hat{Q}^\pi(s, \pi_h(s)) = \text{argmax}_{h \in H_\epsilon} \pi_h(\text{argmin}_a \hat{Q}^\pi(s, a)|s), \forall s \in \mathcal{S} \tag{III$_D$}$$

In this case, the objective is equivalent to objective (II), thus Lemma 27 holds.

**Case II: For a stochastic victim**
In this case, we consider the MDP in Figure 8 and condition (27) to (30). Then the MinQ objective gives $\pi_{h^{\text{MinQ}}}(a_1|s_0) = \beta_0 - \epsilon_0$, $\pi_{h^{\text{MinQ}}}(a_1|s_1) = \beta_1 - \epsilon_1$, and $\pi_{h^{\text{MinQ}}}(a_1|s_2) = \beta_2 - \epsilon_2$.

According to the proof of the first case of Lemma 26, $\pi_{h^{\text{MinQ}}} = \pi_{h^{\text{MaxWorst}}}$ is not an optimal adversary. Thus Lemma 27 holds.

$\square$

### D.3.4 *TYPE IV* - MAXIMIZE DIFFERENCE (MAXDIFF).

The MAD attack proposed by Zhang et al. (Zhang et al., 2020a) is to maximize the distance between the perturbed policy $\pi_h$ and the clean policy $\pi$, i.e.,

$$h^{\text{MaxDiff}} \in \text{argmax}_{h \in H_\epsilon} \text{D}_{\text{TV}}[\pi_h(\cdot|s)||\pi(\cdot|s)], \forall s \in \mathcal{S} \tag{IV}$$

where TV denotes the total variance distance between two distributions. In practical implementations, the TV distance can be replaced by the KL-divergence, as $\text{D}_{\text{TV}}[\pi_h(\cdot|s)||\pi(\cdot|s)] \leq (\text{D}_{\text{KL}}[\pi_h(\cdot|s)||\pi(\cdot|s)])^2$. This type of attack is inspired by the fact that if two policies select actions with similar action distributions on all the states, then the value of the two policies is also small (see Theorem 5 in Zhang et al. (2020a)).

**Lemma 28** (**Optimality of MaxDiff**). *Denote the set of optimal solutions to objective* (IV) *as* $H^{MaxDiff}$. *There exist an MDP* $\mathcal{M}$ *and an agent policy* $\pi$, *such that* $H^{MaxDiff}$ *contains a non-optimal adversary* $h^*$, *i.e.,* $H^{MaxDiff} \not\subset H_\epsilon^*$.

*Proof of Lemma 28.* The proof follows from the proof of lemma 25. In the MDP we constructed, $\pi' = \beta_1 - \epsilon_1, \pi_{h^{\text{MinBest}}}(a_1|s_2) = \beta_2 - \epsilon_2$ is one of the policies that has the maximum KL divergence from the victim policy within Adv-policy-set. However, as we proved in 25, this is not the optimally perturbed policy. Therefore, MaxDiff attacker may not be optimal.

$\square$

# E ADDITIONAL EXPERIMENT DETAILS AND RESULTS

In this section, we provide details of our experimental settings and present additional experimental results. Section E.1 describes our implementation details and hyperparameter settings for Atari and MuJoCo experiments. Section E.2 provide additional experimental results, including experiments with varying budgets ($\epsilon$) in Section E.2.1, more comparison between SA-RL and PA-AD in terms of convergence rate and sensitivity to hyperparameter settings as in Section E.2.3, robust training in MuJoCo games with fewer training steps in Section E.2.4, attacking performance on robust models in Atari games in Section E.2.5, as well as robust training results in Atari games in Section E.2.6.

## E.1 IMPLEMENTATION DETAILS

### E.1.1 ATARI EXPERIMENTS

In this section we report the configurations and hyperparameters we use for DQN, A2C and ACKTR in Atari environments. We use *GeForce RTX 2080 Ti* GPUs for all the experiments.

**DQN Victim** We compare PA-AD algorithm with other attacking algorithms on 7 Atari games. For DQN, we take the softmax of the Q values $Q(s, \cdot)$ as the victim policy $\pi(\cdot|s)$ as in prior works (Huang et al., 2017). For these environments, we use the wrappers provided by stable-baselines (Hill et al., 2018), where we clip the environment rewards to be $-1$ and $1$ during training and stack the last 4 frames as the input observation to the DQN agent. For the victim agent, we implement Double Q learning (Hado Van Hasselt, 2016) and prioritized experience replay (Tom Schaul & Silver, 2016). The clean DQN agents are trained for 6 million frames, with a learning rate 0.00001 and the same network architecture and hyperparameters as the ones used in Mnih et al. (2015). In addition, we use a replay buffer of size $5 \times 10^5$. Prioritized replay buffer sampling is used with $\alpha = 0.6$ and $\beta$ increases from 0.4 to 1 linearly during training. During evaluation, we execute the agent's policy without epsilon greedy exploration for 1000 episodes.

**A2C Victim** For the A2C victim agent, we also use the same preprocessing techniques and convolutional layers as the one used in Mnih et al. (2015). Besides, values and policy network share the same CNN layers and a fully-connected layer with 512 hidden units. The output layer is a categorical distribution over the discrete action space. We use 0.0007 as the initial learning rate and apply linear learning rate decay, and we train the victim A2C agent for 10 million frames. During evaluation, the A2C victim executes a stochastic policy (for every state, the action is sampled from the categorical distribution generated by the policy network). Our implementation of A2C is mostly based on an open-source implementation by Kostrikov Kostrikov (2018).

**ACKTR Adversary** To train the director of PA-AD and the adversary in SA-RL, we use ACKTR (Wu et al., 2017) with the same network architecture as A2C. We train the adversaries of PA-AD and SA-RL for the same number of steps for a fair comparison. For the DQN victim, we use a learning rate 0.0001 and train the adversaries for 5 million frames. For the A2C victim, we use a learning rate 0.0007 and train the adversaries for 10 million frames. Our implementation of ACKTR is mostly based on an open-source implementation by Kostrikov Kostrikov (2018).

**Heuristic Attackers** For the MinBest attacker, we following the algorithm proposed by Huang et al. (2017) which uses FGSM to compute adversarial state perturbations. The MinBest + Mo-

mentum attacker is implemented according to the algorithm proposed by Korkmaz (2020), and we set the number of iterations to be 10, the decaying factor $\mu$ to be 0.5 (we tested $0.01, 0.1, 0.5, 0.9$ and found 0.5 is relatively better while the difference is minor). Our implementation of the MinQ attacker follows the gradient-based attack by Pattanaik et al. (2018), and we also set the number of iterations to be 10. For the MaxDiff attacker, we refer to Algorithm 3 in Zhang et al. (2020a) with the number of iterations equal to 10. In addition, we implement a random attacker which perturbs state $s$ to $\tilde{s} = s + \epsilon \text{sign}(\mu)$, where $\mu$ is sampled from a standard multivariate Gaussian distribution with the same dimension as $s$.

### E.1.2 MuJoCo Experiments

For four OpenAI Gym MuJoCo continuous control environments, we use PPO with the original fully connected (MLP) structure as the policy network to train the victim policy. For robustness evaluations, the victim and adversary are both trained using PPO with independent value and policy optimizers. We complete all the experiments on MuJoCo using *32GB Tesla V100*.

**PPO Victim**   We directly use the well-trained victim model provided by Zhang et al. (2020a).

**PPO Adversary**   Our PA-AD adversary is trained by PPO and we use a grid search of a part of adversary hyperparameters (including learning rates of the adversary policy network and policy network, the entropy regularization parameter and the ratio clip $\epsilon$ for PPO) to train the adversary as powerful as possible. The reported optimal attack result is from the strongest adversary among all 50 trained adversaries.

**Other Attackers**   For Robust Sarsa (RS) attack, we use the implementation and the optimal RS hyperparameters from Zhang et al. (2020a) to train the robust value function to attack the victim. The reported RS attack performance is the best one over the 30 trained robust value functions.

For MaxDiff attack, the maximal action difference attacker is implemented referring to Zhang et al. (2020a).

For SA-RL attacker, following Zhang et al. (2021), the hyperparameters is the same as the optimal hyperparameters of vanilla PPO from a grid search. And the training steps are set for different environments. For the strength of SA-PPO regularization $\kappa$, we choose from $1 \times 10^{-6}$ to 1 and report the worst-case reward.

**Robust Training**   For ATLA Zhang et al. (2021), the hyperparameters for both victim policy and adversary remain the same as those in vanilla PPO training. To ensure sufficient exploration, we run a small-scale grid search for the entropy bonus coefficient for agent and adversary. The experiment results show that a larger entropy bonus coefficient allows the agent to learn a better policy for the continual-improving adversary. In robust training experiments, we use larger training steps in all the MuJoCo environments to guarantee policy convergence. We train 5 million steps in Hopper, Walker, and HalfCheetah environments and 10 million steps for Ant. For reproducibility, the final results we reported are the experimental performance of the agent with medium robustness from 21 agents training with the same hyperparameter set.

### E.2 Additional Experiment Results

### E.2.1 Attacking Performance with Various Budgets

In Table 1, we report the performance of our PA-AD attacker under a chosen epsilon across different environments. To see how PA-AD algorithm performs across different values of $\epsilon$'s, here we select three Atari environments each for DQN and A2C victim agents and plot the performance of PA-AD under various $\epsilon$'s compared with the baseline attackers in Figure 10. We can see from the figures that our PA-AD universally outperforms baseline attackers concerning various $\epsilon$'s.

In Table 2, we provide the evaluation results of PA-AD under a commonly unused epsilon in four MuJoCo experiments (Zhang et al. (2020a; 2021)) to show that PA-AD attacker also has the best attacking performance compared with other attackers under different $\epsilon$'s in Figure 11.

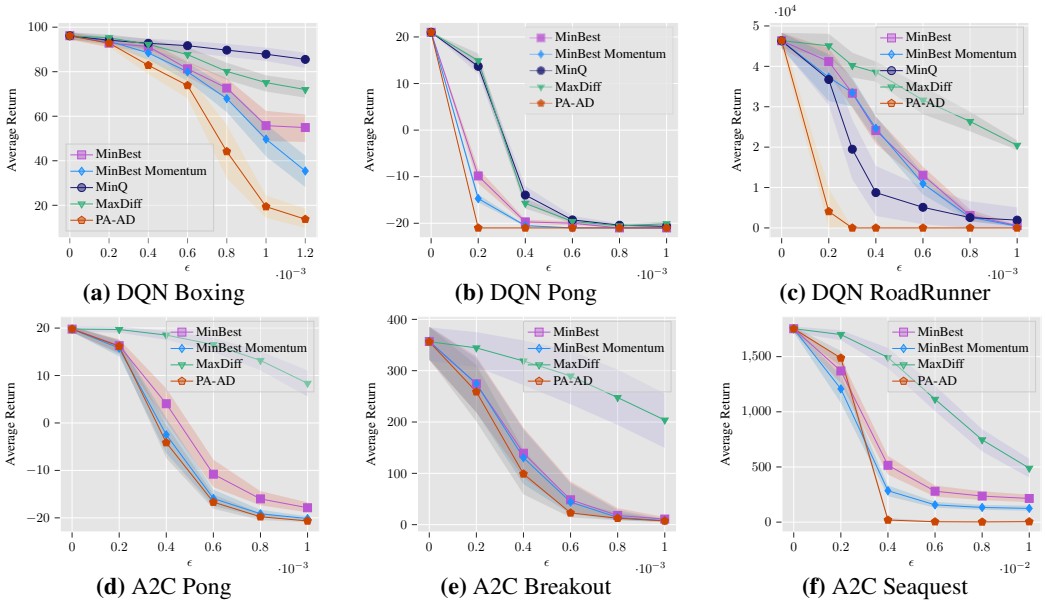

**Figure 10:** Comparison of different attack methods against DQN and A2C victims in Atari w.r.t. different budget $\epsilon$'s.

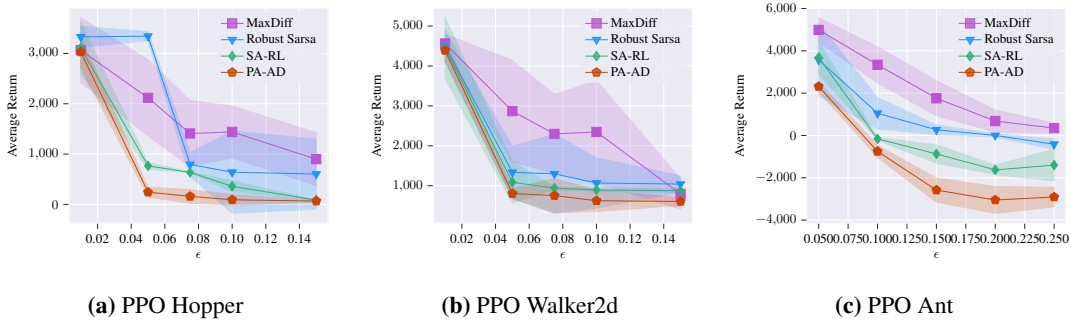

**Figure 11:** Comparison of different attack methods against PPO victims in MuJoCo w.r.t. different budget $\epsilon$'s.

### E.2.2 Hyperparameter Test

In our Actor-Director Framework, solving an optimal actor is a constraint optimization problem. Thus, in our algorithm, we instead use Lagrangian relaxation for the actor's constraint optimization. In this section, we report the effects of different choices of the relaxation hyperparameter $\lambda$ on the final performance of our algorithm. Although we set $\lambda$ by default to be 1 and keep it fixed throughout all of the other experiments, here we find that in fact, difference choice of $\lambda$ has only minor impact on the performance of the attacker. This result demonstrates that our PA-AD algorithm is robust to different choices of relaxation hyperparameters.

**Table 4:** Performance of PA-AD across difference choices of the relaxation hyperparameter $\lambda$

|  | Pong | Boxing |
|---|---|---|
| Nature Reward | $21 \pm 0$ | $96 \pm 4$ |
| $\lambda = 0.2$ | $-19 \pm 2$ | $16 \pm 12$ |
| $\lambda = 0.4$ | $-18 \pm 2$ | $17 \pm 12$ |
| $\lambda = 0.6$ | $-20 \pm 2$ | $19 \pm 15$ |
| $\lambda = 0.8$ | $-19 \pm 2$ | $14 \pm 12$ |
| $\lambda = 1.0$ | $-19 \pm 2$ | $15 \pm 12$ |
| $\lambda = 2.0$ | $-20 \pm 1$ | $21 \pm 15$ |
| $\lambda = 5.0$ | $-20 \pm 1$ | $19 \pm 14$ |

**(a)** Atari

|  | Ant | Walker |
|---|---|---|
| Nature Reward | $5687 \pm 758$ | $4472 \pm 635$ |
| $\lambda = 0.2$ | $-2274 \pm 632$ | $897 \pm 157$ |
| $\lambda = 0.4$ | $-2239 \pm 716$ | $923 \pm 132$ |
| $\lambda = 0.6$ | $-2456 \pm 853$ | $954 \pm 105$ |
| $\lambda = 0.8$ | $-2597 \pm 662$ | $872 \pm 162$ |
| $\lambda = 1.0$ | $-2580 \pm 872$ | $804 \pm 130$ |
| $\lambda = 2.0$ | $-2378 \pm 794$ | $795 \pm 124$ |
| $\lambda = 5.0$ | $-2425 \pm 765$ | $814 \pm 140$ |

**(b)** Mujoco

### E.2.3 Empirical Comparison between PA-AD and SA-RL

In this section, we provide more empirical comparison between PA-AD and SA-RL. Note that PA-AD and SA-RL are different in terms of their applicable scenarios: SA-RL is a black-box attack methods, while PA-AD is a white-box attack method. When the victim model is known, we can see that by a proper exploitation of the victim model, PA-AD demonstrates better attack performance, higher sample and computational efficiency, as well as higher scalability. Appendix F.2 shows detailed theoretical comparison between SA-RL and PA-AD.

**PA-AD has better convergence property than SA-RL.** In Figure 12, we plot the learning curves of SA-RL and PA-AD in the CartPole environment and the Ant environment. Compared with SA-RL attacker, PA-AD has a higher attacking strength in the beginning and converges much faster. In Figure 12b, we can see that PA-AD has a "warm-start" (the initial reward of the victim is already significantly reduced) compared with SA-RL attacker which starts from scratch. This is because PA-AD always tries to maximize the distance between the perturbed policy and the original victim policy in every step according to the actor function ($G$). So in the beginning of learning, PA-AD works similarly to the MaxDiff attacker, while SA-RL works similarly to a random attacker. We also note that although PA-AD algorithm is proposed particularly for environments that have state spaces much larger than action spaces, in CartPole where the state dimensions is fewer than the number of actions, PA-AD still works better than SA-RL because of the distance maximization.

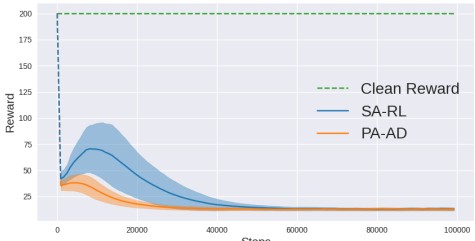

**(a)** Learning curve of SA-RL and PA-AD attacker against an A2C victim in CartPole.

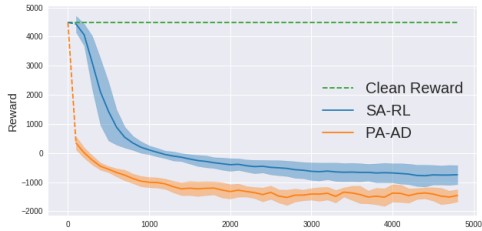

**(b)** Learning curve of SA-RL and PA-AD attacker against a PPO victim in Ant.

**Figure 12:** Comparison of convergence rate between SA-RL and PA-AD in Ant and Cartpole. Results are averaged over 10 random seeds.

**PA-AD is more computationally efficient than SA-RL.** Our experiments in Section 6 show that PA-AD converges to a better adversary than SA-RL given the same number of training steps, which verifies the sample efficiency of PA-AD. Another aspect of efficiency is based on the computational resources, including running time and required memory. For RL algorithms, the computation cost comes from the interaction with the environment (the same for SA-RL and PA-AD) and the policy/value update. If the state space $\mathcal{S}$ is higher-dimensional than the action space $\mathcal{A}$, then SA-RL requires a larger policy network than PA-AD since SA-RL has a higher-dimensional output, and thus SA-RL has more network parameters than PA-AD, which require more memory cost and more computation operations. On the other hand, PA-AD requires to solve an additional optimization problem defined by the actor objective ($G$) or ($G_D$). In our implementation, we use FGSM which only requires one-step gradient computation and is thus efficient. But if more advanced optimization algorithms (e.g. PGD) are used, more computations may be needed. In summary, if $\mathcal{S}$ is much larger than $\mathcal{A}$, PA-AD is more computational efficient than SA-RL; if $\mathcal{A}$ is much larger than $\mathcal{S}$, SA-RL is more efficient than PA-AD; if the sizes of $\mathcal{S}$ and $\mathcal{A}$ are similar, PA-AD may be slightly more expensive than SA-RL, depending on the optimization methods selected for the actor.

To verify the above analysis, we compare computational training time for training SA-RL and PA-AD attackers, which shows that PA-AD is more computationally efficient. Especially on the environment with high-dimensional states like Ant, PA-AD takes significantly less training time than SA-RL (and finds a better adversary than SA-RL), which quantifies the efficiency of our algorithm in empirical experiments.

| Method | Hopper | Walker2d | HalfCheetah | Ant |
|--------|--------|----------|-------------|-----|
| **SA-RL** | 1.80 | 1.92 | 1.76 | 4.88 |
| **PA-AD** | 1.43 | 1.46 | 1.40 | 3.76 |

**Table 5:** Average training time (in hours) of SA-RL and PA-AD in MuJoCo environments, using GeForce RTX 2080 Ti GPUs. For Hopper, Walker2d and HalfCheetah, SA-RL and PA-AD are both trained for 2 million steps; for Ant, SA-RL and PA-AD are both trained for 5 million steps

**PA-AD is less sensitive to hyperparameters settings than SA-RL.** In addition to better final attacking results and convergence property, we also observe that PA-AD is much less sensitive to hyperparameter settings compared to SA-RL. On the Walker environment, we run a grid search over 216 different configurations of hyperparameters, including actor learning rate, critic learning rate, entropy regularization coefficient, and clipping threshold in PPO. Here for comparison we plot two histograms of the agent's final attacked results across different hyperparameter configurations.

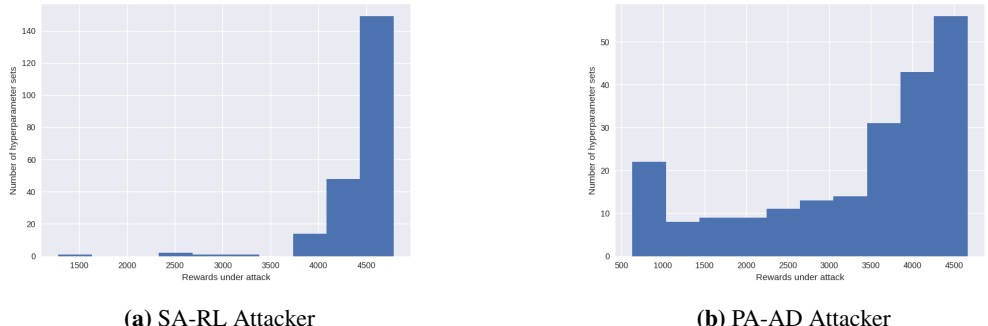

(a) SA-RL Attacker        (b) PA-AD Attacker

**Figure 13:** Histograms of victim rewards under different hyperparameter settings of SA-RL and PA-AD on Walker.

The perturbation radius is set to be 0.05, for which the mean reward reported by Zhang et al. (2020a) is 1086. However, as we can see from this histogram, only one out of the 216 configurations of SA-RL achieves an attacking reward within the range 1000-2000, while in most hyperparameter settings, the mean attacked return lies in the range 4000-4500. In contrast, about $10\%$ hyperparameter settings

of PA-AD algorithm are able to reduce the reward to 500-1000, and another $10\%$ settings could reduce the reward to 1000-2000. Therefore, the performance of PA-AD attacker is generally better and more robust across different hyperparameter configurations than SA-RL.

### E.2.4 ROBUST TRAINING EFFICIENCY ON MUJOCO BY PA-ATLA

In the ATLA process proposed by Zhang et al. (2021), one alternately trains an agent and an adversary. As a result, the agent policy may learn to adapt to the specific type of attacker it encounters during training. In Table 3, we present the performance of our robust training method PA-ATLA-PPO compared with ATLA-PPO under different types of attacks during testing. ATLA-PPO uses SA-RL to train the adversary, while PA-ATLA-PPO uses PA-AD to train the adversary during alternating training. As a result, we can see that ATLA-PPO models perform better under the SA-RL attack, and PA-ATLA-PPO performs better under the PA-AD attack. However, the advantage of ATLA-PPO over PA-ATLA-PPO against SA-RL attack is much smaller than the advantage of PA-ATLA-PPO over ATLA-PPO against PA-AD attack. In addition, our PA-ATLA-PPO models significantly outperform ATLA-PPO models against other heuristic attack methods, and achieve higher average rewards across all attack methods. Therefore, PA-ATLA-PPO is generally more robust than ATLA-PPO.

Furthermore, the efficiency of training an adversary could be the bottleneck in the ATLA Zhang et al. (2021) process for practical usage. Appendix E.2.3 suggests that our PA-AD generally converges faster than SA-RL. Therefore, when the computation resources are limited, PA-ATLA-PPO can train robust agents faster than ATLA-PPO. We conduct experiments on continuous control environments to empirically show the efficiency comparison between PA-ATLA-PPO and ATLA-PPO. In Table 6, we show the robustness performance of two ATLA methods with 2 million training steps for Hopper, Walker and Halfcheetah and 5 million steps for Ant (Compared with results in Table 3, we have reduced training steps by half or more). It can be seen that our PA-ATLA-PPO models still significantly outperform the original ATLA-PPO models under different types of attacks. More importantly, our PA-ATLA-PPO achieves higher robustness under SA-RL attacks in Walker and Ant, suggesting the efficiency and effectiveness of our method.

| Environment | $\epsilon$ | step(million) | Model | Natural Reward | RS Zhang et al. (2020a) | SA-RL Zhang et al. (2021) | PA-AD (ours) | Average reward across attacks |
|---|---|---|---|---|---|---|---|---|
| Hopper | 0.075 | 2 | ATLA-PPO | $1763 \pm 818$ | $1349 \pm 174$ | $1172 \pm 344$ | $\mathbf{477 \pm 30}$ | 999.3 |
| | | | **PA-ATLA-PPO** | $2164 \pm 121$ | $1720 \pm 490$ | $1119 \pm 123$ | $\mathbf{1024 \pm 188}$ | 1287.7 |
| Walker | 0.05 | 2 | ATLA-PPO | $3183 \pm 842$ | $2405 \pm 529$ | $2170 \pm 1032$ | $\mathbf{516 \pm 47}$ | 1697.0 |
| | | | **PA-ATLA-PPO** | $3206 \pm 445$ | $2749 \pm 106$ | $2332 \pm 198$ | $\mathbf{1072 \pm 247}$ | 2051.0 |
| Halfcheetah | 0.15 | 2 | ATLA-PPO | $4871 \pm 112$ | $3781 \pm 645$ | $3493 \pm 372$ | $\mathbf{856 \pm 118}$ | 2710.0 |
| | | | **PA-ATLA-PPO** | $5257 \pm 94$ | $4012 \pm 290$ | $3329 \pm 183$ | $\mathbf{1670 \pm 149}$ | 3003.7 |
| Ant | 0.15 | 5 | ATLA-PPO | $3267 \pm 51$ | $3062 \pm 149$ | $2208 \pm 56$ | $\mathbf{-18 \pm 100}$ | 1750.7 |
| | | | **PA-ATLA-PPO** | $3991 \pm 71$ | $3364 \pm 254$ | $2685 \pm 41$ | $\mathbf{2403 \pm 82}$ | 2817.3 |

**Table 6:** Average episode rewards $\pm$ standard deviation of robust models with fewer training steps under different evasion attack methods. Results are averaged over 50 episodes. We bold the strongest attack in each row. The gray cells are the most robust agents with the highest average rewards across all attacks.

### E.2.5 ATTACKING ROBUSTLY TRAINED AGENTS ON ATARI

In this section, we show the attack performance of our proposed algorithm PA-AD against DRL agents that are trained to be robust by prior works (Zhang et al., 2020a; Oikarinen et al., 2020) in Atari games.

Zhang et al. (2020a) propose **SA-DQN**, which minimizes the action change under possible state perturbations within $\ell_p$ norm ball, i.e., to minimize the extra loss

$$\mathcal{R}_{\text{DQN}}(\theta) := \sum_s \max\left\{ \max_{\hat{s} \in B(s)} \max_{a \neq a^*} Q_\theta(\hat{s}, a) - Q_\theta\left(\hat{s}, a^*(s)\right), -c\right\} \tag{37}$$

where $\theta$ refers to the Q network parameters, $a^*(s) = \text{argmax}_a Q_\theta(a|s)$, and $c$ is a small constant. Zhang et al. (2020a) solve the above optimization problem by a convex relaxation of the Q network, which achieves $100\%$ action certification (i.e. the rate that action changes with a constrained state perturbation) in Pong and Freeway, over $98\%$ certification in BankHeist and over $47\%$ certification in RoadRunner under attack budget $\epsilon = 1/255$.

| | Environment | Natural Reward | $\epsilon$ | Random | MinBest Huang et al. (2017) | MinBest + Momentum Korkmaz (2020) | MinQ Pattanaik et al. (2018) | MaxDiff Zhang et al. (2020a) | PA-AD (ours) |
|---|---|---|---|---|---|---|---|---|---|
| **SA-DQN** | **RoadRunner** | $46440 \pm 5797$ | $\frac{1}{255}$ | $45032 \pm 7125$ | $40422 \pm 8301$ | $43856 \pm 5445$ | $42790 \pm 8456$ | $45946 \pm 8499$ | $\mathbf{38652 \pm 6550}$ |
| | **BankHeist** | $1237 \pm 11$ | $\frac{1}{255}$ | $1236 \pm 12$ | $1235 \pm 15$ | $1233 \pm 17$ | $1237 \pm 14$ | $1236 \pm 13$ | $1237 \pm 14$ |
| **RADIAL -DQN** | **RoadRunner** | $39102 \pm 13727$ | $\frac{1}{255}$ | $41584 \pm 8351$ | $41824 \pm 7858$ | $42330 \pm 8925$ | $40572 \pm 9988$ | $42014 \pm 8337$ | $\mathbf{38214 \pm 9119}$ |
| | | | $\frac{3}{255}$ | $23766 \pm 6129$ | $9808 \pm 4345$ | $35598 \pm 8191$ | $39866 \pm 6001$ | $18994 \pm 6451$ | $\mathbf{1366 \pm 3354}$ |
| | **BankHeist** | $1060 \pm 95$ | $\frac{1}{255}$ | $1037 \pm 103$ | $991 \pm 105$ | $\mathbf{988 \pm 102}$ | $1021 \pm 96$ | $1042 \pm 112$ | $999 \pm 100$ |
| | | | $\frac{3}{255}$ | $1011 \pm 130$ | $801 \pm 114$ | $460 \pm 310$ | $842 \pm 33$ | $1023 \pm 110$ | $\mathbf{397 \pm 172}$ |
| **RADIAL -A3C** | **RoadRunner** | $30854 \pm 7281$ | $\frac{1}{255}$ | $30828 \pm 7297$ | $31296 \pm 7095$ | $31132 \pm 6861$ | $30838 \pm 5743$ | $32038 \pm 6898$ | $\mathbf{30550 \pm 7182}$ |
| | | | $\frac{3}{255}$ | $30690 \pm 7006$ | $30198 \pm 6075$ | $29936 \pm 5388$ | $29988 \pm 6340$ | $31170 \pm 7453$ | $\mathbf{29768 \pm 5892}$ |
| | **BankHeist** | $847 \pm 31$ | $\frac{1}{255}$ | $847 \pm 31$ | $847 \pm 33$ | $848 \pm 31$ | $848 \pm 31$ | $848 \pm 31$ | $848 \pm 31$ |
| | | | $\frac{3}{255}$ | $848 \pm 31$ | $644 \pm 158$ | $822 \pm 11$ | $842 \pm 33$ | $834 \pm 30$ | $\mathbf{620 \pm 168}$ |

**Table 7:** Average episode rewards $\pm$ standard deviation of SA-DQN, RADIAL-DQN, RADIAL-A3C robust agents under different evasion attack methods in Atari environments RoadRunner and BankHeist. All attack methods use 30-step PGD to compute adversarial state perturbations. Results are averaged over 50 episodes. In each row, we bold the strongest attack, except for the rows where none of the attacker reduces the reward significantly (which suggests that the corresponding agent is relatively robust).)

Oikarinen et al. (2020) propose another robust training method named RADIAL-RL. By adding a adversarial loss to the classical loss of the RL agents, and solving the adversarial loss with interval bound propagation, the proposed **RADIAL-DQN** and **RADIAL-A3C** achieve high rewards in Pong, Freeway, BankHeist and RoadRunner under attack budget $\epsilon = 1/255$ and $\epsilon = 3/255$.

**Implementation of the Robust Agents and Environments.** We directly use the trained SA-DQN agents provided by Zhang et al. (2020a), as well as RADIAL-DQN and RADIAL-A3C agents provided by Oikarinen et al. (2020). During test time, the agents take actions deterministically. In order to reproduce the results in these papers, we use the same environment configurations as in Zhang et al. (2020a) and Oikarinen et al. (2020), respectively. But note that the environment configurations of SA-DQN and RADIAL-RL are simpler versions of the traditional Atari configurations we use (described in Appendix E.1.1). Both SA-DQN and RADIAL-RL use a single frame instead of the stacking as 4 frames. Moreover, SA-DQN restricts the number of actions as 6 (4 for Pong) in each environment, although the original environments have 18 actions (6 for Pong). The above simplifications in environments can make robust training easier since the dimensionality of the input space is much smaller, and the number of possible outputs is restricted.

**Attack Methods** In experiments, we find that the robust agents are much harder to attack than vanilla agents in Atari games, as claimed by the robust training papers (Zhang et al., 2020a; Oikarinen et al., 2020). A reason is that Atari games have discrete action spaces, and leading an agent to make a different decision at a state with a limited perturbation could be difficult. Therefore, we use a 30-step Projected Gradient Descent for all attack methods (with step size $\epsilon/10$), including MinBest (Huang et al., 2017) and our PA-AD which use FGSM for attacking vanilla models. Note that the PGD attacks used by Zhang et al. (2020a) and Oikarinen et al. (2020) in their experiments are the same as the MinBest-PGD attack we use. For our PA-AD, we use PPO to train the adversary since PPO is relatively stable. The learning rate is set to be $5e - 4$, and the clip threshold is 0.1. Note that SA-DQN, RADIAL-DQN and RADIAL-A3C agents all take deterministic actions, so we use the deterministic formulation of PA-AD as described in Appendix C.1. In our implementation, we simply use a CrossEntropy loss for the actor as in Equation (38).

$$g_D(\widehat{a}, s) = \operatorname{argmin}_{s' \in B_\epsilon(s)} \mathsf{CrossEntropy}(\pi(s'), \widehat{a}). \tag{38}$$

**Experiment Results** In Table 7, we reproduce the results reported by Zhang et al. (2020a) and Oikarinen et al. (2020), and demonstrate the average rewards gained by these robust agents under different attacks in RoadRunner and BankHeist. Note that SA-DQN is claimed to be robust to attacks with budget $\epsilon = 1/255$, and RADIAL-DQN and RADIAL-A3C are claimed to be relatively robust against up to $\epsilon = 3/255$ attacks. ($\ell_\infty$ is used in both papers.) So we use the same $\epsilon$'s for these agents in our experiments.

It can be seen that compared with vanilla agents in Table 1, SA-DQN, RADIAL-DQN and RADIAL-A3C are more robust due to the robust training processes. However, in some environments, PA-AD can still decrease the rewards of the agent significantly. For example, in RoadRunner with $\epsilon = $

3/255, RADIAL-DQN gets 1k+ reward against our PA-AD attack, although RADIAL-DQN under other attacks can get 10k+ reward as reported by Oikarinen et al. (2020). In contrast, we find that RADIAL-A3C is relatively robust, although the natural rewards gained by RADIAL-A3C are not as high as RADIAL-DQN and SA-DQN. Also, as SA-DQN achieves over $98\%$ action certification in BankHeist, none of the attackers is able to noticeably reduce its reward with $\epsilon = 1/255$.

Therefore, our PA-AD can approximately evaluate the worst-case performance of an RL agent under attacks with fixed constraints, i.e., *PA-AD can serve as a "detector" for the robustness of RL agents.* For agents that perform well under other attacks, PA-AD may still find flaws in the models and decrease their rewards; for agents that achieve high performance under PA-AD attack, they are very likely to be robust against other attack methods.

### E.2.6 Improving Robustness on Atari by PA-ATLA

Note that different from SA-DQN (Zhang et al., 2020a) and RADIAL-RL (Oikarinen et al., 2020) discussed in Appendix E.2.5, we use the traditional Atari configurations (Mnih et al., 2015) without any simplification (e.g. disabling frame stacking, or restricting action numbers). We aim to improve the robustness of the agents in original Atari environments, as in real-world applications, the environments could be complex and unchangeable.

**Baselines** We propose PA-ATLA-A2C by combining our PA-AD and the ATLA framework proposed by Zhang et al. (2021). We implement baselines including vanilla A2C, adversarially trained A2C (with MinBest (Huang et al., 2017) and MaxDiff (Zhang et al., 2020a) adversaries attacking 50 frames). SA-A2C (Zhang et al., 2020a) is implemented using SGLD and convex relaxations in Atari environments.

In Table 6, naive adversarial training methods have unreliable performance under most strong attacks and SA-A2C is ineffective under PA-AD strongest attack. To provide evaluation using different $\epsilon$, we provide the attack rewards of all robust models with different attack budgets $\epsilon$. Under all attacks with different $\epsilon$ value, PA-ATLA-A2C models outperform all other robust models and achieve consistently better average rewards across attacks. We can observe that our PA-ATLA-A2C training method can considerably enhance the robustness in Atari environments.

| Model | Natural Reward | $\epsilon$ | Random | MinBest Huang et al. (2017) | MaxDiff Zhang et al. (2020a) | SA-RL Zhang et al. (2021) | PA-AD (ours) | Average reward across attacks |
|---|---|---|---|---|---|---|---|---|
| A2C vanilla | $1228 \pm 93$ | 1/255 | $1223 \pm 77$ | $972 \pm 99$ | $1095 \pm 107$ | $1132 \pm 30$ | $\mathbf{436 \pm 74}$ | 971.6 |
| | | 3/255 | $1064 \pm 129$ | $697 \pm 153$ | $913 \pm 164$ | $928 \pm 124$ | $\mathbf{284 \pm 116}$ | 777.2 |
| A2C (adv: MinBest Huang et al. (2017)) | $948 \pm 94$ | 1/255 | $932 \pm 69$ | $927 \pm 30$ | $936 \pm 11$ | $940 \pm 103$ | $\mathbf{704 \pm 19}$ | 887.8 |
| | | 3/255 | $874 \pm 51$ | $813 \pm 32$ | $829 \pm 27$ | $843 \pm 126$ | $\mathbf{521 \pm 72}$ | 774.2 |
| A2C (adv: MaxDiff Zhang et al. (2020a)) | $743 \pm 29$ | 1/255 | $756 \pm 42$ | $702 \pm 89$ | $752 \pm 79$ | $749 \pm 85$ | $\mathbf{529 \pm 45}$ | 697.6 |
| | | 3/255 | $712 \pm 109$ | $638 \pm 133$ | $694 \pm 115$ | $686 \pm 110$ | $\mathbf{403 \pm 101}$ | 626.6 |
| SA-A2C Zhang et al. (2021) | $1029 \pm 152$ | 1/255 | $1054 \pm 31$ | $902 \pm 89$ | $1070 \pm 42$ | $1067 \pm 18$ | $\mathbf{836 \pm 70}$ | 985.8 |
| | | 3/255 | $985 \pm 47$ | $786 \pm 52$ | $923 \pm 52$ | $972 \pm 126$ | $\mathbf{644 \pm 153}$ | 862.0 |
| PA-ATLA-A2C (ours) | $1076 \pm 56$ | 1/255 | $1055 \pm 204$ | $957 \pm 78$ | $1069 \pm 94$ | $1045 \pm 143$ | $\mathbf{862 \pm 106}$ | 997.6 |
| | | 3/255 | $1026 \pm 78$ | $842 \pm 154$ | $967 \pm 82$ | $976 \pm 159$ | $\mathbf{757 \pm 132}$ | 913.6 |

**Table 8:** Average episode rewards $\pm$ standard deviation over 50 episodes of A2C, A2C with adv. training, SA-A2C and our PA-ATLA-A2C robust models under different evasion attack methods in Atari environment BankHeist. In each row, we bold the strongest attack. The gray cells are the most robust agents with the highest average rewards across all attacks.

## F Additional Discussion of Our Algorithm

### F.1 Optimality of Our Relaxed Objective for Stochastic Victims

**Proof of Concept: Optimality Evaluation in A Small MDP**

We implemented and tested heuristic attacks and our PA-AD in the 2-state MDP example used in Appendix B.6, and visualize the results in Figure 14. For simplicity, assume the adversaries can perturb only perturb $\pi$ at $s_1$ within a $\ell_2$ norm ball of radius 0.2. And we let all adversaries perturb the policy directly based on their objective functions. As shown in Figure 14a, all possible $\tilde{\pi}(s_1)$'s form a disk in the policy simplex, and executing above methods, as well as our PA-AD, leads to 4 different policies on this disk. All these computed policy perturbations are on the boundary of the policy perturbation ball, justifying our Theorem 4.

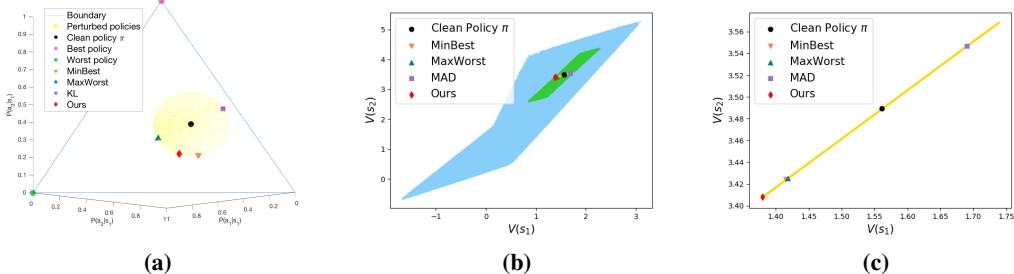

**Figure 14:** Comparison of the optimality of different adversaries. **(a)** The policy perturbation generated for $s_1$ by all attack methods. **(b)** The values of corresponding policy perturbations. **(c)** A zoomed in version of (b), where the values of all possible policy perturbations are rendered. Our method finds the policy perturbation that achieves the lowest reward among all perturbations.

As our theoretical results suggest, the resulted value vectors lie on a line segment shown in Figure 14b and a zoomed in version Figure 14c, where one can see that MinBest, MaxWorst and MAD all fail to find the optimal adversary (the policy with lowest value). On the contrary, our PA-AD finds the optimal adversary that achieves the lowest reward over all policy perturbations.

**For Continuous MDP: Optimality Evaluation in CartPole and MountainCar**

We provided a comparison between SA-RL and PA-AD in the CartPole environment in Figure 15, where we can see the SA-RL and PA-AD converge to the same result (the learned SA-RL adversary and PA-AD adversary have the same attacking performance).

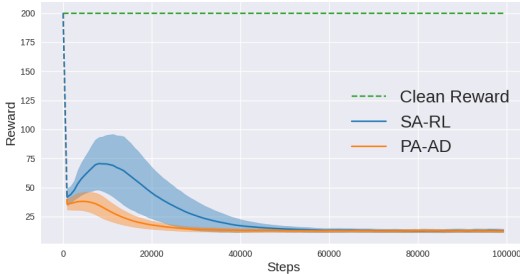

**Figure 15:** Learning curve of SA-RL and PA-AD attacker against an A2C victim in CartPole.

CartPole has a 4-dimensional state space, and contains 2 discrete actions. Therefore since SA-RL has an optimal formulation, we expect SA-RL to converge to the optimal adversary in a small MDP like CartPole. Then the result in Figure 15 suggests that our PA-AD algorithm, although with a relaxation in the actor optimization, also converges to the optimal adversary with even a faster rate than SA-RL (the reason is explained in Appendix E.2.3).

In addition to CartPole, we also run experiments in MountainCar with a 2-dimensional state space against a DQN victim. The SA-RL attacker reduces the victim reward to -128, and our PA-AD attacker reduces the victim reward to -199.45 within the same number of training steps. Note that the lowest reward in MountainCar is -200, so our PA-AD indeed converges to a near-optimal adversary, while SA-RL fails to converge to a near-optimal adversary. This is because MountainCar is an environment with relatively spare rewards. The actor in **PA-AD utilizes our Theorem 4 and only focuses on perturbations in the outermost boundary, which greatly reduces the exploration burden in solving an RL problem**. In contrast, SA-RL directly uses RL algorithms to learn the perturbation, and thus it has difficulties in converging to the optimal solution.

## F.2    MORE COMPARISON BETWEEN SA-RL AND PA-AD

We provide a more detailed comparison between SA-RL and PA-AD from the following multiple aspects to claim our contribution.

**1. Size of the Adversary MDP**
Suppose the original MDP has size $|\mathcal{S}|$, $|\mathcal{A}|$ for its state space and action space, respectively. Both PA-AD and SA-RL construct an adversary's MDP and search for the optimal policy in it. But the adversary's MDPs for PA-AD and SA-RL have different sizes.

**PA-AD**: state space is of size $|\mathcal{S}|$, action space is of size $\mathbb{R}^{|\mathcal{A}|-1}$ for a stochastic victim, or $|\mathcal{A}|$ for a deterministic victim.
**SA-RL**: state space is of size $|\mathcal{S}|$, action space is of size $|\mathcal{S}|$.

**2. Learning Complexity and Efficiency**
When the state space is larger than the action space, which is very common in RL environments, PA-AD solves a smaller MDP than SA-RL and thus more efficient. In environments with pixel-based states, SA-RL becomes computationally intractable, while PA-AD still works. It is also important to note that the actor's argmax problem in PA-AD further accelerates the convergence, as it rules out the perturbations that do not push the victim policy to its outermost boundary. Our experiment and analysis in Appendix E.2.3 verify the efficiency advantage of our PA-AD compared with SA-RL, even in environments with small state spaces.

**3. Optimality**
**PA-AD**: (1) the formulation is optimal for a deterministic victim policy; (2) for a stochastic victim policy, the original formulation is optimal, but in practical implementations, a relaxation is used which may not have optimality guarantees.
**SA-RL**: the formulation is optimal.
Note that both SA-RL and PA-AD require training an RL attacker, but the RL optimization process may not converge to the optimal solution, especially in deep RL domains. Therefore, SA-RL and PA-AD are both approximating the optimal adversary in practical implementations.

**4. Knowledge of the Victim**
**PA-AD**: needs to know the victim policy (white-box). Note that in a black-box setting, PA-AD can still be used based on the transferability of adversarial attacks in RL agents, as verified by Huang et al. (2017). But the optimality guarantee of PA-AD does not hold in the black-box setting.
**SA-RL**: does not need to know the victim policy (black-box).
It should be noted that the white-box setting is realistic and helps in robust training:
(1) The white-box assumption is common in existing heuristic methods.
(2) It is always a white-box process to evaluate and improve the robustness of a given agent, for which PA-AD is the SOTA method. As discussed in our Ethics Statement, the ultimate goal of finding the strongest attacker is to better understand and improve the robustness of RL. During the robust training process, the victim is the main actor one wants to train, so it is a white-box setting. The prior robust training art ATLA (Zhang et al., 2021) uses the black-box attacker SA-RL, despite the fact that it has white-box access to the victim actor. Since SA-RL does not utilize the knowledge of the victim policy, it usually has to deal with a more complex MDP and face converging difficulties. In contrast, if one replaces SA-RL with our PA-AD, PA-AD can make good use of the victim policy and find a stronger attacker with the same training steps as SA-RL, as verified in our Section 6 and Appendix E.2.6.

**5. Applicable Scenarios**
**SA-RL is a good choice if** (1) the action space is much larger than the state space in the original MDP, or the state space is small and discrete; (2) the attacker wants to conduct black-box attacks.
**PA-AD is a good choice if** (1) the state space is much larger than the state space in the original MDP; (2) the victim policy is known to the attacker; (3) the goal is to improve the robustness of one's own agent via adversarial training.

In summary, as we discussed in Section 4, there is a trade-off between efficiency and optimality in evasion attacks in RL. SA-RL has an optimal RL formulation, but empirical results show that *SA-RL usually do not converge to the optimal adversary in a continuous state space, even in a low-dimensional state space* (e.g. see Appendix F.1 for an experiment in MountainCar). Therefore, the difficulty of solving an adversary's MDP is the bottleneck for finding the optimal adversary. Our PA-AD, although may sacrifice the theoretical optimality in some cases, greatly reduces the size and the exploration burden of the attacker's RL problem (can also be regarded as trading some estimation bias off for lower variance). Empirical evaluation shows our PA-AD significantly outperforms SA-RL in a wide range of environments.

Though PA-AD requires to have access to the victim policy, PA-AD solves a smaller-sized RL problem than SA-RL by utilizing the victim's policy and can be applied on evaluating/improving the robustness of RL policy. It is possible to let PA-AD work in a black-box setting based on the transferability of adversarial attacks. For example, in a black-box setting, the attacker can train a proxy agent in the same environment, and use PA-AD to compute a state perturbation for the proxy agent, then apply the state perturbation to attack the real victim agent. This is out of the scope of this paper, and will be a part of our future work.

### F.3 VULNERABILITY OF RL AGENTS

It is commonly known that neural networks are vulnerable to adversarial attacks (Goodfellow et al., 2015). Therefore, it is natural that deep RL policies, which are modeled by neural networks, are also vulnerable to adversarial attacks (Huang et al., 2017). However, there are few works discussing the difference between deep supervised classifiers and DRL policies in terms of their vulnerabilities. In this section, we take a step further and investigate the vulnerability of DRL agents, through a comparison with standard adversarial attacks on supervised classifiers. Our main conclusion is that commonly used deep RL policies can be instrinsically **much more vulnerable** to small-radius adversarial attacks. The reasons are explained below.

**1. Optimization process**
Due to the different loss functions that RL and supervised learning agents are trained on, the size of robustness radius of an RL policy is much smaller than that of a vision-based classifier.
On the one hand, computer vision-based image classifiers are trained with cross-entropy loss. Therefore, the classifier is encouraged to make the output logit of the correct label to be larger than the logits of other labels to maximize the log probability of choosing the correct label. On the other hand, RL agents, in particular DQN agents, are trained to minimize the Bellman Error instead. Thus the agent is not encouraged to maximize the absolute difference between the values of different actions. Therefore, if we assume the two networks are lipschitz continuous and their lipschitz constants do not differ too much, it is clear that a supervised learning agent has a much larger perturbation radius than an RL agent.

To prove our claim empirically, we carried out a simple experiment, we compare the success rate of target attacks of a well-trained DQN agent on Pong with an image classifier trained on the CIFAR-10 dataset with similar network architecture. For a fair comparison, we use the same image preprocessing technique, which is to divide the pixel values by 255 and no further normalization is applied. On both the image-classifier and DQN model, we randomly sample a target label other than the model predicted label and run the same 100-step projected gradient descent (PGD) attack to minimize the cross-entropy loss between the model output and the predicted label. We observe that for a perturbation radius of 0.005 ($l_\infty$ norm), the success rate of a targeted attack for the image classifier is only $15\%$, whereas the success rate of a targeted attack for the DQN model is $100\%$. This verifies our claim that a common RL policy is much more vulnerable to small-radius adversarial attacks than image classifiers.

**2. Network Complexity**
In addition, we also want to point out that the restricted network complexity of those commonly used deep RL policies could play an important role here. Based on the claim by Madry et al. (2018), a neural network with greater capacity could have much better robustness, even when trained with only clean examples. But for the neural network architectures commonly used in RL applications, the capacity of the networks is very limited compared to SOTA computer vision applications. For example, the commonly used DQN architecture proposed in Mnih et al. (2015) only has 3 convolutional layers and 2 fully connected layers. But in vision tasks, a more advanced and deeper structure (e.g. ResNet has 100 layers) is used. Therefore, it is natural that the perturbation radius need for attacking an RL agent is much smaller than the common radius studied in the supervised evasion attack and adversarial learning literature.

