# OpenReview forum: "Who Is the Strongest Enemy? Towards Optimal and Efficient Evasion Attacks in Deep RL"
_ICLR.cc/2022/Conference — ICLR 2022 Poster_

### Official Review · Reviewer_73ny · 2021-10-31

**Correctness:** 3
**Technical Novelty And Significance:** 3
**Empirical Novelty And Significance:** 3
**Recommendation:** 8
**Confidence:** 4

**Main Review:**

The paper's novelty lies in the fact that the attacks are crafted based on the mapping from the state space to the action space, which is more efficient than the previously proposed methods. And the technique is provably efficient and generates relatively stronger attacks than its alternatives. Studying adversarial attacks on RL is an important subject, and the paper is original and novel.

However, some of the claims made in the paper are not accurate.

First, on page 1, "many existing attacks (Huang et al., Zhang et al., 2020) are based on heuristics...". To the best of my knowledge, the attacks studied in (Zhang et al., 2020) are not based on heuristics, which are actually the results of the SA-MDP.

Second, on page 2, "summary of contributions," the authors say they introduce a "compact" policy adversarial MDP. Can the authors explain "compact" in the context? Does it mean the policy space is compact?

Third, in definition 6, when the authors define the action space, \hat{\mathcal{A}}, the sum of d_i should be one if I understand it correctly.

Fourth,in fig 4, since the victim observes \tilde{s}, should the policy in the figures be written as \pi(\tilde{s})?

Here are some other comments:

1. In definition 2, there are assumptions on the MDP and \pi that one needs to make sure the admissible adversarial policy set is connected and compact. Would the authors mind listing them in the main body of the paper? This is because not every MDP problem and \pi satisfy these assumptions. People who want to apply your method better get notified of these conditions.
2. The authors mention several times that their approach is more efficient because the action space is much smaller than the state space in most deep RL applications. However, the authors did not demonstrate and quantify such efficiency in the experiments, especially not in the tables presented. In figure 12, the authors did show that PA-AD converges faster. But it didn't show that the PA-AD approach is more computationally tractable.
2. The authors list an incomplete list of "related works to adversarial RL." Adversarial (deep?) RL has attracted much attention in recent years.  People studied adversarial RL for both robustness and security purposes. If the authors only want to talk about works that are developed for robustness purposes. The papers listed in the manuscript are almost complete. Otherwise, below are some works that I deem relevant. The authors can consider including them when discussing related works:
      * Zhang, Xuezhou, et al. "Adaptive reward-poisoning attacks against reinforcement learning." International Conference on Machine Learning. PMLR, 2020.
     * Rakhsha, Amin, et al. "Policy teaching via environment poisoning: Training-time adversarial attacks against reinforcement learning." International Conference on Machine Learning. PMLR, 2020.
     * Huang, Yunhan, and Quanyan Zhu. "Deceptive reinforcement learning under adversarial manipulations on cost signals." International Conference on Decision and Game Theory for Security. Springer, Cham, 2019.
    *  Russo, Alessio, and Alexandre Proutiere. "Towards optimal attacks on reinforcement learning policies." 2021 American Control Conference (ACC). IEEE, 2021.






**Summary Of The Paper:**

The paper proposes a method to craft stronger and more efficient attacks on state observation of the RL agent.  Here, the 'strongest' attacks refer to the attacks that minimize the reward the most. An 'efficient' attacker can craft such attacks using the least computational resources. In the work of (Zhang et al., 2021), the attacking strategy is described by a mapping from the state space of the underlying MDP problem to the same space. The agent observes the falsified states rather than the actual ones and takes actions based on the observed states. This misinformation eventually tricks the agent into taking a different action. In this paper, the authors argue that instead of finding the mapping from the state space to the same space, the attacker can first find the mapping from the state space to the action space that generates the least accumulated rewards to the agent. The 'strongest' attacks on the state observation can then be crafted based on the mapping found. Since in many deep RL applications, the state space is much larger than the action space. Hence, finding the mapping from the state space to the action space is more efficient than finding the one from the state space to the state space.

**Summary Of The Review:**

In general, the paper is well-structured, and the problem studied here leads to a robust RL agent, which fuels the applicability of RL in real-world situations. However, some claims in the paper are not accurate. But these claims do not affect the main results. Also, there is still room to improve the writing.

---

> ### Author Response · Authors · 2021-11-16
> **[Response 1/2] Clarifications on our claims**
>
> We thank Reviewer 73ny for the detailed and insightful feedback. We are encouraged that Reviewer 73ny finds our paper original and novel. We have updated our manuscript and made the claims more clear as suggested by Reviewer 73ny. Below we address Reviewer 73ny's concerns in detail.
>
> > Q1: on page 1, "many existing attacks (Huang et al., Zhang et al., 2020) are based on heuristics...". To the best of my knowledge, the attacks studied in (Zhang et al., 2020) are not based on heuristics, which are actually the results of the SA-MDP.
>
> Here we are referring to the Robust Sarsa (RS) attack and the Maximal Action Difference (MAD) proposed by Zhang et al. 2020, which are based on heuristics. But we understand that this may lead to confusion. We have changed this citation on page 1.
>
> > Q2: on page 2, "summary of contributions," the authors say they introduce a "compact" policy adversarial MDP. Can the authors explain "compact" in the context? Does it mean the policy space is compact?
>
> Here we mean that the PAMDP is in general **smaller** than the adversary MDP proposed by Zhang et al. 2020 and Zhang et al. 2021. It is not relevant to the concept of compact in topology. We apologize for the inappropriate wording. We have removed the word "compact" in the introduction.
>
> > Q3: in definition 6, when the authors define the action space, $\hat{\mathcal{A}}$, the sum of $d_i$ should be one if I understand it correctly.
>
> This is not a typo, since $\widehat{a}$ is designed to be a "perturbing direction" in the policy space. That is, $\forall \widehat{a}\in\widehat{A}$,
> - there exists a constant $\theta_0 \geq 0$ such that $\forall \theta\leq \theta_0, \pi(\cdot|s) + \theta \frac{\widehat{a}}{\|\widehat{a}\|}$ belongs to the simplex $\Delta(A)$ (the perturbed policy is a valid probability vector);
> - there exists a constant $\theta_1 \geq 0$ such that $\forall \theta\leq \theta_1, \pi(\cdot|s) + \theta \frac{\widehat{a}}{\|\widehat{a}\|}$ belongs to the Adv-policy-set $B^H_\epsilon(\pi)$ (can be generated by an admissible state adversary).
>
> The goal of the actor is then to perturb $\pi(s)$ towards the direction of $\widehat{a}$ as far as possible under the given attack budget, which is described by problem (G).
>
> Here we use a concrete example to illustrate how $\widehat{a}$ works. In state $s$, suppose the victim policy takes action $a_1$ with probability 0.6, and action $a_2$ with probability 0.4, which can be denoted as $\pi(\cdot|s)=[0.6, 0.4]$; if the perturbing direction is $\widehat{a}=[-0.1, 0.1]$ (elements sum to 0), then, probability vectors $[0.5, 0.5]$, $[0.4, 0.6]$, $[0.35, 0.65]$ are all on the perturbing direction of $\widehat{a}$.
>
> We thank the reviewer for raising this question. We have updated our manuscript and added more interpretations to avoid confusion (page 6).
>
>
> > Q4: in fig 4, since the victim observes $\tilde{s}$, should the policy in the figures be written as $\pi(\tilde{s})$?
>
> This is indeed a typo. Thank Reviewer 73ny for pointing it out. We have fixed it.
>
>
> > Q5: In definition 2, there are assumptions on the MDP and $\pi$ that one needs to make sure the admissible adversarial policy set is connected and compact. Would the authors mind listing them in the main body of the paper?
>
> Thank Reviewer 73ny for the advice. We apologize for putting the detailed analysis of $B^H_\epsilon$ in the Appendix due to the space limit. Now we have added more explanations in the main paper (page 3).
>
> In short, the only condition we need for $B^H_\epsilon(\pi)$ being connected and compact is that $\pi$ is a continuous function, which is satisfied by most function approximators including neural networks. Therefore, our algorithm is applicable in most deep RL problems as verified in our experiments.

---

> > ### Author Response · Authors · 2021-11-16
> > **[Response 2/2] Clarifications on our claims**
> >
> > > Q6: The authors mention several times that their approach is more efficient ... But it didn't show that the PA-AD approach is more computationally tractable.
> >
> > Thank Reviewer 73ny for the advice. PA-AD is theoretically more efficient than SA-RL as we explain in the last paragraph of page 6. In the paragraph above Table 3, page 9, we also mentioned the empirical efficiency of PA-AD compared to SA-RL. But we admit that we did not explicitly compare the empirical efficiency of SA-RL and PA-AD in experiments due to the space limit. Now we have made the efficiency comparison more clear in the paper (page 8 and page 9 for sample efficiency) and provided more experiment results (page 33 for computational efficiency). Below we explain our claims in detail.
> >
> > **1. Comparison of Sample Complexity.**
> >
> > We emphasize that since both SA-RL and PA-AD have optimal formulations, and we use the same training steps for SA-RL in PA-AD in experiments, **the reason that PA-AD outperforms SA-RL in experiments is exactly its sample efficiency.**
> > - In Table 1, we can see that SA-RL works similarly to a random attacker in Atari games, although it is trained with the same number of steps/samples as PA-AD. This is because in Atari games, SA-RL learns a policy with both pixel inputs and pixel outputs, which is hard and expensive.
> > - In Table 2, we can see that PA-AD outperforms SA-RL in most cases, especially in Ant where the state space is relatively large. This also justifies the sample efficiency of PA-AD.
> > - Figure 12 in Appendix E.2.3 shows the convergence comparison, which justifies that the sample efficiency of PA-AD is higher than SA-RL. (PA-AD requires fewer samples than SA-RL to learn a near-optimal adversary.)
> > - In Appendix E.2.4 and Table 5, we also show that with fewer training steps, PA-AD outperforms SA-RL more significantly, which implies the sample efficiency of PA-AD compared to SA-RL.
> >
> > **2. Comparison of Computational Complexity.**
> >
> > Another aspect of efficiency is based on the computational resources, including running time and required memory. We are not sure whether Reviewer 73ny's question is regarding this type of efficiency. If so, we provide the following analysis and additional results below.
> >
> > *(1) Theoretical analysis:* suppose we use the same deep RL algorithm to learn both SA-RL and PA-AD attacks with the same number of training steps.
> > - If the state space $\mathcal{S}$ is higher-dimensional than the action space $\mathcal{A}$, then SA-RL requires a larger policy network than PA-AD since SA-RL has a higher-dimensional output. Therefore, SA-RL has more network parameters than PA-AD, and thus requires more memory cost and more computation operations.
> > - On the other hand, PA-AD requires solving an additional optimization problem defined by the actor objective $(G)$ or $(G_D)$. In our implementation, we use FGSM which only requires one-step gradient computation and is thus efficient. But if more advanced optimization algorithms (e.g. PGD) are used, more computations may be needed.
> >
> > In summary, if $\mathcal{S}$ is much larger than $\mathcal{A}$ which is common in practice, PA-AD is more computational efficient than SA-RL; if $\mathcal{A}$ is much larger than $\mathcal{S}$, SA-RL is more efficient than PA-AD; if the sizes of $\mathcal{S}$ and $\mathcal{A}$ are similar, PA-AD may be slightly more expensive than SA-RL, depending on the optimization methods selected for the actor.
> >
> > *(2) Empirical comparison:* We have added Table 5 to Appendix E.2.3 (page 33) to show the running time of SA-RL and PA-AD in MuJoCo games. (For Atari games, the running time comparison does not make much sense since SA-RL does not find a good adversary as Table 1 shows.) We can see that **PA-AD generally takes less running time than SA-RL,** as these classic environments have larger state spaces than action spaces.
> >
> > > Q7: The authors list an incomplete list of "related works to adversarial RL."
> >
> > Thank Reviewer 73ny for providing these related works. We were aware of these relevant and insightful adversarial RL papers. The reason why we did not cite the first 3 papers is that they study poisoning (training-time) attacks, different from the evasion (test-time) attack we focus on. But Reviewer 73ny is correct that they are important works in adversarial RL. We have updated our paper and cited them in the related work section.
> >
> > For the 4th paper by Russo et al., we have already cited it (page 12) and discussed it (the 2nd paragraph in related work). Please let us know if our understanding of the work is inaccurate.
> >
> > ---
> > We greatly appreciate Reviewer 73ny for the positive comments and constructive suggestions. We have updated our paper to address all the points raised by the reviewer. Please let us know if there are other questions or concerns. Thank you!

---

### Official Review · Reviewer_RTbW · 2021-11-01

**Correctness:** 2
**Technical Novelty And Significance:** 2
**Empirical Novelty And Significance:** 2
**Recommendation:** 3
**Confidence:** 2

**Main Review:**

The definition of admissible perturbations relies on that the attacks would be hard to be perceived. How can one determine a reasonable epsilon?

The attacker not knowing the victim policy yet not knowing the environment dynamics is relatively contrived.

There is an optimality argument in the paper for the problem posed in the paper. Yet, it is not clear whether this problem as posed is a realistic or even a relevant one. For example, the set B_epsilon^H, which is used in the problem statement itself, is hard obtain.

Additionally, the importance of the theoretical result even in this contrived setting is unclear because there is no guarantee that the algorithm devised in the paper will be able to compute that optimal.

The claims made in the paper are likely unjustifiably strong. For example, how can a few empirical examples be used to conclude that a method "universally" outperforms another one? What is even the universe?

----------------

Comments after the author response:

The fact that there are other papers (even if they are popular papers) is not a justification of the importance of a problem. In an adversarial setting, if your adversary knows the policy, which itself can be a complicated object, you are implementing, then the problem you are studying is the least of your issue. You are your strongest enemy. Maybe consider changing the policy, first.

You are going to try to justify that this is the worst-case analysis and you are empowering the adversary. Then, why does the adversary not know anything about the environment? So, it is really not a worst-case analysis either. Also, isn't it odd that the adversary does not understand *anything* about the environment but understands a policy used in that environment?

**Summary Of The Paper:**

The paper proposes a method for computing the strongest adversarial perturbation on state observations of an RL agent from a specified set of perturbations. It relates perturbations on states to perturbations on policies. It poses a specific problem to this end, develops a solution, and establishes its optimality.

**Summary Of The Review:**

The paper is possibly on an interesting topic yet the problem studied in the paper is contrived and the results are of limited interest.

---

> ### Author Response · Authors · 2021-11-10
> **We are solving a common and important problem in adversarial RL, not a contrived problem**
>
> We thank Reviewer RTbW for reviewing our paper and providing feedback.
>
> We think that Reviewer RTbW might have some misunderstandings on our problem setup, because Reviewer RTbW summarized our work as
>
> > Q0: It poses a specific problem to this end, develops a solution, and establishes its optimality.
>
> However, we do not "pose and address a specific problem". Instead, we focus on a common and important problem: *finding the optimal adversary to attack an RL agent with a given attack radius.* This problem is studied and well-motivated by many adversarial RL papers[1,2]. We re-formulate this common problem as solving a PAMDP (Definition 6). As commented by other reviewers, our paper provides solid contributions to the line of work on adversarial attacks.
>
> Therefore, we are not sure why Reviewer RTbW thinks we pose and address a "specific problem". We believe that **our problem setup is not different from existing adversarial RL papers**[1,2,3,4]. We would like to hear more from the reviewer if they have different opinions.
>
>
> ---
> Below we address Reviewer RTbW's other concerns in detail.
>
> > Q1: The definition of admissible perturbations ... How can one determine a reasonable epsilon?
>
>
> First, the goal of our paper is not to determine a reasonable $\epsilon$, but to find the strongest attack for any given $\epsilon$. We do not require that the attacks are hard to be perceived. Our algorithm and theory work for any $\epsilon>0$.
>
> Second, we consider $l_p$ attack ($\| s-\tilde{s} \|_p \leq \epsilon$) which is the most commonly used threat model in adversarial RL[1,2,3,4] and the broader area of adversarial learning[5,6]. As in related works[1-5], $\epsilon$ is a hyper-parameter specified by the attacker, controlling how large a perturbation the attacker can incur. In practice, an attacker usually desires smaller $\epsilon$ in order to avoid being detected.
>
>
> > Q2: The attacker not knowing the victim ... is relatively contrived.
>
> We guess Reviewer RTbW wanted to say "The attacker knowing the victim", since this is what we do in the paper. We believe that this setting is common and is not contrived due to the following reasons.
>
> - Not knowing the environment dynamics is a reasonable setup: the dynamics of real-life environments (e.g. video games, auto-driving systems) are usually complicated and unknown. So, it is a typical setting that neither the agent nor the attacker knows the dynamics in the adversarial RL literature[1,2,3,4].
> - Knowing the victim is also a reasonable setup: this is called a white-box attack model which is considered in many adversarial learning papers[2,3,5,6].
>
> We hope we have convinced the reviewer that our setting is relevant and not contrived. If not, we would appreciate any constructive suggestions about a more reasonable setting.
>
> > Q3: There is an optimality argument in the paper ... is hard obtain.
>
> First, as stated in our answers above, we focus on a common evasion RL setup, not a special setting.
>
> Second, $B_\epsilon^H$ is a concept used to rigorously define the optimal attack, but we **do not need to obtain** $B_\epsilon^H$. We instead search for the optimal perturbing direction based on Theorem 4, which is not hard and is a novel contribution of this paper. Please see the paragraph above Theorem 4 for details.
>
>
> > Q4: Additionally, the importance of ... compute that optimal.
>
> First, as we answered above, our setting is not contrived.
>
> Second, as our contribution (2) on page 2 states, we convert the optimal evasion attack problem (a common and well-motivated problem[1,2]) to the problem of solving a new MDP (Definition 6). Theorem 7 proves the optimality of our formulation. Regarding the implementation of the algorithm, we use off-the-shelf deep RL algorithms to solve this PAMDP. The final results depend on these deep RL algorithms, whose optimality is beyond the scope of our paper. Please refer to Appendix F.1. for empirical justifications on the optimality of PA-AD.
>
>
> > Q5: The claims ... What is even the universe?
>
> In every sentence with the word "universally", there is a phrase illustrating the conditions, e.g. "our proposed PA-AD universally outperforms state-of-the-art attacking methods **in various Atari and MuJoCo environments**".
>
> Therefore, the word "universally" is with regard to the set of environments that we evaluate the algorithms on. If Reviewer RTbW has different opinions on this claim, we will appreciate it if Reviewer RTbW can provide more suggestions.
>
> Finally, we would like to emphasize that we use a wide range of environments and various attack budgets, and our algorithm significantly outperforms all baselines. We believe that this is a solid contribution to the robust RL community.
>
> ---
>
> Thank you again for your time and effort in reviewing our paper. Please let us know if the above explanation does not address your concerns. We are happy to answer any further questions.
>
> (References are in the next reply due to the space limit)

---

> > ### Author Response · Authors · 2021-11-10
> > **References**
> >
> > [1] Zhang et al. Robust reinforcement learning on state observations with learned optimal adversary. ICLR 2021.
> >
> > [2] Pattanaik et al. Robust deep reinforcement learning with adversarial attacks. AAMAS 2018.
> >
> > [3] Huang et al. Adversarial attacks on neural network policies.
> >
> > [4] Zhang et al. Robust deep reinforcement learning against adversarial perturbations on observations. NeurIPS 2020.
> >
> > [5] Goodfellow et al. Explaining and harnessing adversarial examples. ICLR 2015.
> >
> > [6] Moosavi-Dezfooli et al. DeepFool: a simple and accurate method to fool deep neural network. CVPR 2016.

---

> ### Author Response · Authors · 2021-11-29
> **Further Clarifications**
>
> We thank Reviewer RTbW for updating the review, and we appreciate your comments very much. We would like to further clarify our contributions and the importances of the studied problem below.
>
> ### About the importance of the problem
>
> Even if the reviewer does not agree that the problem of finding the strongest enemy itself is important, please note an important usage of our proposed method: **to improve the robustness of RL policies.** The robustness of deep learning models is widely studied in the field of deep learning, and is believed to be an important topic. Therefore, the study of the strongest enemy and our proposed method is very important to understanding and improving the robustness of deep RL policies (we achieve SOTA robustness in many environments).
>
>
> ### Response to Proposed Concerns
>
> > Concern 1: The fact that there are other papers (even if they are popular papers) is not a justification of the importance of a problem. In an adversarial setting, if your adversary knows the policy, which itself can be a complicated object, you are implementing, then the problem you are studying is the least of your issue. You are your strongest enemy. Maybe consider changing the policy, first.
>
> There might be some misunderstanding about how adversarial attacks work in practice. The reviewer suggests the agent to change the policy, which is a good defending strategy. But in practice, **the agent does not necessarily know when it is being hacked and attacked.** Therefore, it is important to **"prepare for the worst",** i.e. to be aware the worst-case performance of the agent and to improve the robustness against any possible attacker.
>
> Please note that the white-box attack is commonly studied in both supervised learning and reinforcement learning as we cited in our previous reply. In this line of research, the key is to understand the vulnerability of deep neural networks to adversarial attacks in the test time. As motivated in these papers, the study of the strongest attack is an important step towards understanding and improving robustness of deep learning models.
>
> > Concern 2: You are going to try to justify that this is the worst-case analysis and you are empowering the adversary. Then, why does the adversary not know anything about the environment? So, it is really not a worst-case analysis either. Also, isn't it odd that the adversary does not understand anything about the environment but understands a policy used in that environment?
>
> We would like to emphasize that this is a common setting in literature and a realistic setting in practice.
> 1. **Why the adversary does not know the environment dynamics.**
> We would like to clarify that we do not say "the adversary does not know anything of the environment". Instead, the adversary does not know the exact transition probabilities and the reward functions of the environment. Please note that we consider a common deep RL problem, where it is often unrealistic to know the environment dynamics. For example, an Atari game has a large state space, and it is hard to measure the transition probabilities of all states. Moreover, in a real-world application like auto-driving, the environment dynamics are determined by real-world physics, that are often not known to any learner. An analogy of not knowing the environment dynamics in supervised learning is that the underlying data distribution is unknown, which is a common and realistic setting.
>
> 2. **Why the adversary knows the victim policy.**
> Different from the environment, the victim policy is just a function (e.g. a neural network) implemented by programs, which is relatively easy to obtain. For example, the attacker can be an insider or a hacker who cracks the program of the agent policy. In some real-world applications, the policy network may even be carried by distributed devices that could be hacked.
>
> Therefore, it is reasonable that the adversary does not know the environment dynamics but knows the victim policy in an environment. This is a common setting in literature -- we are not aware of any adversary deep RL papers that assume knowledge of the environment dynamics, and most existing attack methods in RL assume the knowledge of the victim policy, to the best of our knowledge.
>
>
> > Concern 3: it is really not a worst-case analysis either if the adversary does not know the environment.
>
> Since the victim is interacting with the environment, the adversary can eavesdrop the interaction samples of the victim. Therefore, the adversary is **learning** from the interaction samples using RL methods with exploration and exploitation. Our claim is that the adversary should learn an optimal solution to the proposed PAMDP, which is theoretically shown to be equivalent to an optimal adversary.
>
>
> ---
>
> We greatly appreciate your questions. We sincerely hope that our clarifications can address your concerns. We are happy to discuss more if there are further questions.

---

> > ### Author Response · Authors · 2021-11-30
> > **To Reviewer RTbW: evaluating and improving the robustness of RL policies are important**
> >
> > Dear Reviewer RTbW,
> >
> > As the discussion period is going to end, we would like to kindly ask you to consider our clarifications. We would like to again emphasize the importance of the studied problem, which seems to be the major concern of the reviewer. In particular, the reviewer said
> >
> > > The fact that there are other papers (even if they are popular papers) is not a justification of the importance of a problem.
> >
> > We are sorry if our previous explanation did not explain the importance well for the reviewer. We did not claim that the problem is important because a lot of papers are studying it. We would like to point out that the problem is important as **verified** by many papers, both in theory and in practice.
> >
> > In theory, as pointed out by many fundamental papers in adversarial training[1], the central objective is a **minimax problem, and the inner maximization is exactly to find the strongest attacker.** Therefore, our paper focuses on the problem of finding the strongest attacker and is thus very important for robust learning.
> >
> > In practice, [2] has proposed to study the "strongest attack" against an RL policy, and justified that such an attack leads to much more robust agents than prior robust training methods. We believe that there is no need to stress the importance of robustness in deep learning, as it is one of the most important topics nowadays in the deep learning community.
> >
> > In summary, **identifying the strongest attacker is significantly helpful for evaluating and improving the robustness of RL policies.** As we show in the experiments, the vulnerability of a trained policy can be revealed by our PA-AD attack, and the robustness can be improved by our PA-AD attack with state-of-the-art performance. Therefore, what we study is an important problem.
> >
> > We will appreciate it a lot if the reviewer could consider our explanations and consider raising the score. We are happy to discuss more if there are any other questions.
> >
> > Best regards,
> >
> > Paper4077 Authors
> >
> > ---
> > Refs:
> >
> > [1] Madry, Aleksander, et al. "Towards deep learning models resistant to adversarial attacks."
> >
> > [2] Zhang et al. Robust reinforcement learning on state observations with learned optimal adversary. ICLR 2021.

---

### Official Review · Reviewer_pYuF · 2021-11-02

**Correctness:** 3
**Technical Novelty And Significance:** 3
**Empirical Novelty And Significance:** 3
**Recommendation:** 8
**Confidence:** 3

**Main Review:**

Overall, I enjoyed reading the paper, and in my opinion, this paper contains interesting results and provides solid contributions to the line of work on adversarial attacks. The paper both formally justifies its approach and provides concrete experimental evidence that the approaches improves over the state-of-the-art methods. Admittedly, compared to prior work, some of the modeling assumptions might be restrictive - more detailed remarks about the paper (i.e., its strengths and weaknesses) are listed below.

- *Novelty*: While the problem of designing optimal evasion attacks in deep RL has been studied by prior work, the key idea that the paper introduces seems rather novel. To my knowledge, this type of attack method has not been considered in the recent literature.

- *Methodology*: The paper provides justifications grounded in theory. Utilizing the results from e.g. Dadashi et al. 2019, it shows that the novel framework admits an optimal solution, despite the fact that the proposed method searches over policies rather than over the state space. While this fact seems intuitive, I still find this result interesting and non-trivial.

- *Modeling assumptions*: One drawback of the proposed approach is that it relies on the assumption that the attacker has access to the victim's policy. This assumptions is somewhat restrictive, corresponding to a white-box attack, and it seems to be critical for the formal analysis. Some of the SOTA method do not seem to require this assumption (SA-RL).

- *Experiments*: The paper conducts extensive experiments, and empirical results show that the proposed approach outperforms the SOTA attack methods under the studied setting, and significantly so.

- *Clarity*: The paper is clearly written and it provides intuitions behind the most important technical aspects. Minor comments: The paper contains a few typos, the most important one might be in Definition 6: why is the sum $\sum_i d_i$ equal to  $0$? Is this supposed to be $\sum_i d_i = 1$? Moreover, some of the references could be updated, e.g., Zhang et al. 2021 is referenced as an arXiv paper, but this paper seems to have appeared at ICLR2021.


**Summary Of The Paper:**

The paper studies evasion attacks in deep reinforcement learning (RL). More specifically, the paper considers a novel approach to performing evasion attacks based on two-component design --- director and actor modules, where the latter perturbs a given state based on the policy direction that the former specifies. Effectively, the search for an optimal attack is performed in the policy space, and since the policy space is typically more compact than the state space, the search is more efficient. The paper formally justifies its design choices, and experimentally validates the efficacy of the propose approach, showing that it yields significant improvements compared to the state-of-the-art methods.


**Summary Of The Review:**

This paper provides interesting contributions to the line of work on adversarial attacks. The paper contributes a novel attack method and shows that it outperforms the SOTA baselines.

---

> ### Author Response · Authors · 2021-11-16
> **More discussion on the white-box setting and clarifications regarding the notations**
>
> We thank Reviewer pYuF for the positive comment and valuable suggestions. We are encouraged that Reviewer pYuF thinks our paper provides solid contributions to the line of work on adversarial attacks. We have updated the paper according to Reviewer pYuF's advice, and we address Reviewer pYuF's questions and concerns below.
>
> > Q1: One drawback of the proposed approach is that it relies on the assumption that the attacker has access to the victim's policy ... Some of the SOTA method do not seem to require this assumption (SA-RL).
>
> We admit that due to the space limit, we didn’t emphasize the difference between PA-AD and SA-RL in terms of the white-box v.s. black-box assumption, although we provided the comparison in Appendix F.2. Now we have updated the main paper and explained it explicitly (page 5 and page 9).
>
> Compared to SA-RL which treats the victim and the environment together as a black-box, PA-AD exploits the victim model and reduces the complexity of learning the adversary. We would like to emphasize the following points.
>
> 1. White-box attacks are common in literature and useful in practice. Many existing evasion attack works[1] assume white-box access to the model parameters. Existing evasion RL attackers usually computes perturbations w.r.t. a known policy model, e.g. MinBest[3], MinQ[2] and MaxDiff[5], etc.
>
> 2. In adversarial training where the user has white-box access to the model, PA-AD improves the robustness of RL agents more than SA-RL. (See Table 3 and Table 7.) This is a practical and important application of PA-AD.
>
> 3. PA-AD can be extended to a black-box setting based on the transferability of adversarial attacks, although it is out of the scope of our paper. Following prior works[3,4], we can use PA-AD to attack a trained proxy victim model and transfer the attack to the real victim, although the optimality guarantee may not hold in this case.
>
> 4. The black-box attack SA-RL, although having an optimal formulation, can not naturally scale up to a large state-space (e.g. images). Our Table 1 shows that SA-RL works similarly as a random attacker in Atari games, with the same number of training steps as PA-AD.
>
> In summary, PA-AD and SA-RL can be applied to different scenarios. When the action space is larger than the state space, or when the victim policy is unknown and the state space is not too large, SA-RL can be a good choice. On the contrary, PA-AD should be used when the state space is large, or when the victim policy is known (e.g. in adversarial training). Please see our Appendix F.2. for a detailed comparison between SA-RL and PA-AD based on their complexity, optimality, assumption, and applicable scenarios. More empirical comparisons between SA-RL and PA-AD are in Appendix E.2.3.
>
> > Q2: In Definition 6: why is the sum $\sum_i d_i$ equal to 0? Is this supposed to be $\sum_i d_i=1$?
>
> This is not a typo, since $\widehat{a}$ is designed to be a "perturbing direction" in the policy space. That is, $\forall \widehat{a}\in\widehat{A}$,
> - there exists a constant $\theta_0 \geq 0$ such that $\forall \theta\leq \theta_0, \pi(\cdot|s) + \theta \frac{\widehat{a}}{\|\widehat{a}\|}$ belongs to the simplex $\Delta(A)$ (the perturbed policy is a valid probability vector);
> - there exists a constant $\theta_1 \geq 0$ such that $\forall \theta\leq \theta_1, \pi(\cdot|s) + \theta \frac{\widehat{a}}{\|\widehat{a}\|}$ belongs to the Adv-policy-set $B^H_\epsilon(\pi)$ (can be generated by an admissible state adversary).
>
> The goal of the actor is then to perturb $\pi(s)$ towards the direction of $\widehat{a}$ as far as possible under the given attack budget, which is described by problem (G).
>
> Here we use a concrete example to illustrate how $\widehat{a}$ works. In state $s$, suppose the victim policy takes action $a_1$ with probability 0.6, and action $a_2$ with probability 0.4, which can be denoted as $\pi(\cdot|s)=[0.6, 0.4]$; if the perturbing direction is $\widehat{a}=[-0.1, 0.1]$ (elements sum to 0), then, probability vectors $[0.5, 0.5]$, $[0.4, 0.6]$, $[0.35, 0.65]$ are all on the perturbing direction of $\widehat{a}$.
>
> We thank the reviewer for raising this question. We have updated our manuscript and added more interpretations to avoid confusion (page 6).
>
> > Q3: Some of the references could be updated.
>
> We have updated the references as suggested. Thank Reviewer pYuF for pointing them out.
>
> ---
> We greatly appreciate Reviewer pYuF's valuable feedback on our paper. Please let us know if there are any additional questions.
>
> ---
> Refs:
>
> [1] Chakraborty, Anirban, et al. Adversarial attacks and defences: A survey.
>
> [2] Pattanaik et al. Robust deep reinforcement learning with adversarial attacks. AAMAS 2018.
>
> [3] Huang et al. Adversarial attacks on neural network policies.
>
> [4] Inkawhich, Matthew, Yiran Chen, and Hai Li. Snooping attacks on deep reinforcement learning. AAMAS 2020.
>
> [5] Zhang et al. Robust deep reinforcement learning against adversarial perturbations on observations. NeurIPS 2020.

---

> > ### Comment · Reviewer_pYuF · 2021-11-21
> > **Thank you for the clarifications**
> >
> > Thank you for providing further clarifications. My opinion about the paper has not considerably changed.

---

### Official Review · Reviewer_mGyi · 2021-11-03

**Correctness:** 4
**Technical Novelty And Significance:** 3
**Empirical Novelty And Significance:** 3
**Recommendation:** 6
**Confidence:** 3

**Main Review:**

I find the paper very well-written and easy to follow. The idea is novel and the proposed method gives new insights about the evasion attacks. My only issue is that many significant details are left to the appendix while mentioning them in a short sentence could give the reader a better understanding of the limitations. For example

- The algorithm is assuming access to the learner's policy. I don't mind this assumption since white-box attacks have their own purposes and scenarios, but the authors could be more direct about this. Especially considering the amount of comparison done with SA-RL which is a black-box attack. Seems like the advantages of PA-AD stem from this additional knowledge. I still find the method to utilize this knowledge is a respectable contribution.

- The details of solving optimization problem (G) could be more explained in the main text. It is a challenging problem to solving without relaxation and "can be implemented in various ways" is to vague.

Even though the above points leave room for improvement, I think the paper is a solid contribution.

**Summary Of The Paper:**

The paper introduces a novel algorithm to find the optimal evasion attack against RL in scenarios where the agent's policy is known. The key idea is to only learn the aspect of problem depending on the environment and is unknown through a clever decomposition. The algorithm shows promising empirical results.

**Summary Of The Review:**

The paper has some limitation on the knowledge assumptions and need for relaxation, but the contribution is enough in my opinion.

---

> ### Author Response · Authors · 2021-11-16
> **We have moved more details from the appendix to the main paper and added more discussion**
>
> We thank Reviewer mGyi's valuable feedback and constructive suggestions. We are encouraged that Reviewer mGyi found our method novel and our results promising.
>
> Reviewer mGyi's main concern is that many details are put in the Appendix. We appreciate Reviewer mGyi's suggestion and have updated our manuscript to include more details. For the points raised by the reviewer, we provide more explanations below.
>
> - **Additional Explanation on the White-box Access:**
>
> 1. White-box attacks are common in literature and useful in practice. Many existing evasion attack works[1] assume white-box access to the model parameters. Existing evasion RL attackers usually computes perturbations w.r.t. a known policy model, e.g. MinBest[3], MinQ[2] and MaxDiff[5], etc.
>
>
> 2. In adversarial training where the user has white-box access to the model, PA-AD improves the robustness of RL agents more than SA-RL. (See Table 3 and Table 7.) This is a practical and important application of PA-AD.
>
> 3. PA-AD can still be used in a black-box setting, based on the transferability of adversarial attacks. Following prior works[3,4], we can use PA-AD to attack a trained proxy victim model and transfer the attack to the real victim, although the optimality guarantee may not hold in this case.
>
> 4. The black-box attack SA-RL, although having an optimal formulation, can not naturally scale up to a large state-space (e.g. images). Our Table 1 shows that SA-RL works similarly as a random attacker in Atari games, with the same number of training steps as PA-AD.
>
> - **Discussion about PA-AD and SA-RL:**
>
> We admit that due to the space limit, we didn’t emphasize the difference between PA-AD and SA-RL in terms of the white-box v.s. black-box assumption, although we provided the comparison in Appendix F.2. Now we have updated the main paper and explained it explicitly (page 5 and page 9).
>
> SA-RL treats the environment and the victim together as a black-box environment. Although the SA-RL attacker does not need to know the policy network, the complexity of learning the black-box environment becomes too high for a large state space. Our finding is, when the victim parameter is known, the large black-box environment can be simplified by a proper disentanglement of the attack process (Figure 3 in paper). Therefore, Reviewer mGyi is right that the efficiency of PA-AD comes from the knowledge of the victim. However, please note that SA-RL does not explicitly use the victim model even if the model is known. For example, in the alternate training algorithm[6], the agent and the attacker are trained together, but the attacker does not exploit the agent's model. On the contrary, PA-AD effectively exploits the agent's model and **finally leads to more robust RL models.** (See Table 3 and Table 7.)
>
> In addition, we have a detailed comparison between SA-RL and PA-AD based on their complexity, optimality, assumption, and applicable scenarios in our Appendix F.2., and an empirical comparison in Appendix E.2.3.
>
>
> - **About Solving Optimization Problem (G):**
>
> We thank Reviewer mGyi for the advice. The implementation details of the optimization problem are provided in Appendix C.2. in our original submission. Due to the space limit, we are not able to show all details in the main paper. But following reviewer's advice, we add more discussion in the revised paper (page 7) and make it precise. Please let us know if there are more suggestions.
>
> ---
> Since our paper proposes a new evasion RL method with both theoretical insights and empirical results, it is hard to put all details and results in a 9-page paper. Therefore, we have to put some solid theoretical analysis, complementary experimental results and comprehensive discussions in the Appendix. We made sure that the results in Appendix are well-indexed and cited in the main paper, so that readers who are interested can easily navigate to them. Please let us know if anything else needs further clarification.
>
> We greatly appreciate Reviewer mGyi's time and effort in reviewing our paper. We hope that our modification in the paper and explanation above address the questions of Reviewer mGyi. We are happy to answer any further questions.
>
>
> ---
> Refs:
>
> [1] Chakraborty, Anirban, et al. Adversarial attacks and defences: A survey.
>
> [2] Pattanaik et al. Robust deep reinforcement learning with adversarial attacks. AAMAS 2018.
>
> [3] Huang et al. Adversarial attacks on neural network policies.
>
> [4] Inkawhich, Matthew, Yiran Chen, and Hai Li. Snooping attacks on deep reinforcement learning. AAMAS 2020.
>
> [5] Zhang et al. Robust deep reinforcement learning against adversarial perturbations on observations. NeurIPS 2020.
>
> [6] Zhang et al. Robust reinforcement learning on state observations with learned optimal adversary. ICLR 2021.

---

### Public Comment · ~Ezgi_Korkmaz2 · 2021-11-15
**Some Questions on Details**

Thank you for the interesting work. I have a couple of questions relating to some of the details provided in the submission.

The results in Table 6 of the proposed attack are within one standard deviation of natural rewards in half of the games, and in the other half of the games they are equal. What should we conclude from these results?

In Table 7 the prior work and the proposed algorithm seem like they are within one standard deviation. Again what can we conclude from these results?
For adversarial training experiments why did the authors choose A2C? All the prior work on adversarial training in ALE is with DQN. Why are the results not reported with DQN?

How are the hyperparameters (e.g. $\mu$) for the Nesterov Momentum attack decided? It could be nice if the authors can provide the range of $\mu$. I am not sure 0.5 is the optimal choice for $\mu$ here.

Why did the authors exclude Boxing from the A2C experiment in Table 1?

Are the authors sure that DQN training is conducted with exactly the same network architecture and hyperparameters with [1,2] with Double Q-learning as is claimed in their submission? Because even though the authors trained for significantly fewer frames the scores of the trained agents are way higher in this submission (the authors of the submission train for 6 million frames, the original paper trains for 200 million frames).

|      Games       | Submission |  Original paper |
|--------------------|------------------|---------------------|
| Tutankhamun  |       227        |         33.6           |
| Alien                |     1623       |         900.5         |
| Boxing             |        96         |          68.6          |



[1] Ioannis Antonoglou Tom Schaul, John Quan and David Silver. Prioritized experience replay. In International Conference on Learning Representations, 2016.

[2] Volodymyr Mnih, Koray Kavukcuoglu, David Silver, Andrei A Rusu, Joel Veness, Marc G Bellemare, Alex Graves, Martin Riedmiller, Andreas K Fidjeland, Georg Ostrovski, et al. Human-level control through deep reinforcement learning. Nature, 518(7540):529, 2015.

---

> ### Author Response · Authors · 2021-11-15
> **Answers to the proposed questions**
>
> Thank you for the questions! Here are our answers.
>
> > Q1: The results in Table 6 of the proposed attack are within one standard deviation of natural rewards in half of the games, and in the other half of the games they are equal. What should we conclude from these results?
>
> Table 6 shows the results of attacking existing robustly trained models in RoadRunner and BankHeist. We can see that in many cases, the robust models still achieve good performance under the proposed attack and other attacks. This suggests that SA-DQN, RADIAL-DQN and RADIAL-A3C are indeed robust under current attack methods, including our PA-AD attack. But in some scenarios, e.g. RADIAL-DQN in RoadRunner with $\epsilon=3/255$, PA-AD reduces the reward much more than previous attacks. Therefore, our claim is that PA-AD, as a strong attack method, helps with evaluating the worst-case performance of a well-trained agent.
>
> > Q2: In Table 7 the prior work and the proposed algorithm seem like they are within one standard deviation. Again what can we conclude from these results? For adversarial training experiments why did the authors choose A2C? All the prior work on adversarial training in ALE is with DQN. Why are the results not reported with DQN?
>
> Similar to our answer above, the evaluated models are trained to be robust, so it is reasonable to see that current attacking methods can not reduce their reward by too much.
>
> We chose A2C mainly because it is fast. DQN usually requires longer training time to converge than A2C. Since the ATLA framework requires alternately training an agent and an attacker, it can be relatively expensive if the agent is trained with DQN. More advanced and efficient adversarial training methods can be adapted to train robust DQN models, but it is out of the scope of this paper. We will be happy to see if someone can combine PA-AD and other adversarial RL training techniques to produce robust models.
>
> > Q3: How are the hyperparameters (e.g. $\mu$) for the Nesterov Momentum attack decided? It could be nice if the authors can provide the range of $\mu$. I am not sure 0.5 is the optimal choice for $\mu$ here.
>
> In our initial test, we experimented with several different $\mu$'s (e.g. 0.01, 0.1, 0.5, 0.9) as shown in the table below. We found that the difference between the final results are minor. Therefore, we report the result of Nesterov Momentum Attack Method with $\mu=0.5$, which achieves relatively good performance among all $\mu$'s we tested.
>
> |                | Natural reward | $\mu=0.01$ | $\mu=0.1$ | $\mu=0.5$ | $\mu=0.9$ | ours |
> |----------------|----------------|------------|-----------|-----------|-----------|------|
> | DQN-Pong       | 21             | -14        | -12       | -14       | -14       | -21  |
> | DQN-Boxing     | 96             | 52         | 52        | 52        | 56        | 19   |
> | A2C-Alien      | 1615           | 956        | 911       | 940       | 940       | 507  |
> | A2C-Tutankham  | 258            | 132        | 134       | 134       |     127      | 71   |
>
> Any suggestions on how to set $\mu$ will be greatly appreciated. We will test them and update the results in the paper.
>
> > Q4: Why did the authors exclude Boxing from the A2C experiment in Table 1?
>
> This is because we find out that in our experiments, the victim A2C learner cannot achieve a reasonably good performance on Boxing, same as the observation reported in the original A3C paper [5]. Therefore, we did not carry out any further attacking experiments for A2C policies on Boxing.
>
>
> > Q5: Are the authors sure that DQN training is conducted with exactly the same network architecture and hyperparameters with [1,2] with Double Q-learning as is claimed in their submission? Because even though the authors trained for significantly fewer frames the scores of the trained agents are way higher in this submission (the authors of the submission train for 6 million frames, the original paper trains for 200 million frames).
>
> In our experiment, we reproduce the implementation of Double Q-learning [2] and Prioritized Experience Replay [3] with the exact same network architecture and hyperparameters. The difference between our reported clean reward and the reward cited in the comment is due to the different evaluation methods. As explained in the paper [1,2,3], the clean rewards cited in the comment are based on a "human start" evaluation method, where the initial state is sampled from a human expert's trajectory. In comparison, because we do not have access to such human experts' trajectories, we use the "no-op" evaluation method instead. It is also explained in detail in the paper [2], where we execute a special no-op action that does not affect the environment up to 30 times, to provide different starting points for the dqn agent. In our final implementation, we use the Atari environment wrapper provided by the stable_baselines [4] codebase.
>
> (References are in the next response)

---

> > ### Author Response · Authors · 2021-11-15
> > **References**
> >
> > [1] Mnih et al. Human-level control through deep reinforcement learning. *Nature*, 518(7540):529, 2015.
> >
> > [2] Hasselt et al. Deep Reinforcement Learning with Double Q-learning. AAAI 2016.
> >
> > [3] Schaul et al. Priortized Experince Replay. Neurips 2016.
> >
> > [4] Hill et al. Stable Baselines. 2019
> >
> > [5] Minh et al. Asynchronous Methods for Deep Reinforcement Learning. ICML 2016.

---

### Author Response · Authors · 2021-11-17
**General Response: Summary of Paper Updates**

We thank all reviewers for their insightful questions and valuable feedback. We are particularly encouraged that they consider the tackled problem important (*73ny, pYuF*), the proposed method novel (*mGyi, pYuF, 73ny*), the results significant (*mGyi, pYuF*) and our paper well-written (*mGyi, pYuF*).

Here we briefly outline the updates to the revised submission based on the reviews. We address individual questions of reviewers in separate responses.

### Paper Updates:

- **[Section 1]** (1) We highlighted the motivation and the importance of the proposed question. (*RTbW*) (2) We improved the writing and updated the citations. (*73ny*)
- **[Section 2]** We described the threat model more precisely. (*RTbW*)
- **[Section 3]** We have expanded the remarks of Definition 2 in greater details, including the conditions for $B^H_\epsilon$ being connected and compact. (*73ny*)
- **[Section 4]** (1) We have added explanations about the difference between SA-RL and PA-AD in terms of the assumptions. (*mGyi, pYuF*) (2) We explained the design of $\widehat{A}$ more clearly. (*pYuF,73ny*) (3) We improved the writing about the algorithm description, and added more details in solving the actor's optimization problem. (*mGyi*)
- **[Section 5]** We cited more papers on poisoning attacks. (*73ny*)
- **[Section 6]** (1) We have added more efficiency comparison between SA-RL and PA-AD (*73ny*). (2) We emphasized the different applicable scenarios of SA-RL and PA-AD. (*mGyi, pYuF*) (3) We improved the writing and made the description more precise. (*73ny*)
- **[References]** We updated the publication venues of some references. (*pYuF*)
- **[Appendix E.2.3]** (1) We emphasized the difference between the knowledge of SA-RL and PA-AD. (*mGyi, pYuF*) (2) We added the running time comparison between SA-RL and PA-AD. (*73ny*)
- **[Appendix F.2]** We listed the applicable scenarios for SA-RL and PA-AD, respectively. (*mGyi, pYuF*)

We greatly appreciate all reviewers' suggestions. We hope that our paper updates and responses have addressed reviewers' questions and concerns. Please let us know if there are further questions.

---

### Decision · Program_Chairs · 2022-01-20

**Decision:**

Accept (Poster)

**Comment:**

Overall the paper makes good contributions to the area of robust deep reinforcement learning. The presentation needs to be improved to avoid any confusions. Please take all the reviews into account and revise the paper accordingly.